# Competitive processes shape multi-synapse plasticity along dendritic segments

Thomas E. Chater[1,3,6], Maximilian F. Eggl [2,4,6], Yukiko Goda [1,5,7] ✉ & Tatjana Tchumatchenko [2,7] ✉

Neurons receive thousands of inputs onto their dendritic arbour, where individual synapses undergo activity-dependent plasticity. Long-lasting changes in postsynaptic strengths correlate with changes in spine head volume. The magnitude and direction of such structural plasticity - potentiation (sLTP) and depression (sLTD) - depend upon the number and spatial distribution of stimulated synapses. However, how neurons allocate resources to implement synaptic strength changes across space and time amongst neighbouring synapses remains unclear. Here we combined experimental and modelling approaches to explore the elementary processes underlying multi-spine plasticity. We used glutamate uncaging to induce sLTP at varying number of synapses sharing the same dendritic branch, and we built a model incorporating a dual role $Ca^{2+}$-dependent component that induces spine growth or shrinkage. Our results suggest that competition among spines for molecular resources is a key driver of multi-spine plasticity and that spatial distance between simultaneously stimulated spines impacts the resulting spine dynamics.

A typical neuron in the mammalian brain receives thousands of synaptic inputs across its dendritic arbour. The pattern of presynaptic input can drive changes in the size and molecular composition of corresponding postsynaptic dendritic spines and induce synaptic potentiation or depression. In classical Hebbian synaptic plasticity, changes are thought to be confined to active synapses and independent of other nearby inputs. However, there is growing evidence that stimulated spines can be affected by the cross-talk of stimulated and unstimulated spines, giving rise to complex multi-spine plasticity patterns[1–3]. Consequently, when two or more plasticity induction events nearly coincide in space and time, predicting the outcome is critical for understanding future patterns of neuronal activity that drive circuit functions[4–6]. Various roles for multi-site plasticity have been proposed, from homeostatic function that supports

homosynaptic change at the expense of compensatory, heterosynaptic changes (e.g., refs. 7,8) to synchronisation of co-active inputs across brain regions[2,9–11]. It remains an open question what the mechanisms are that determine the variety of plasticity outcomes across different spatio-temporal multi-spine stimulation protocols[12]. In part, this is a combinatorial problem that needs to be studied using defined spatial and temporal distances between stimulated synapses across several to tens of minutes, and this investigation would benefit from a model framework to explain the observed effects.

At the postsynaptic spine, synaptic plasticity is triggered by a flux of $Ca^{2+}$ ions that enter the spine through NMDARs, which then engage a wide range of downstream $Ca^{2+}$-dependent signalling events in the postsynaptic neuron. Under certain conditions, the spread of signalling molecules to neighbouring spines can mediate further plasticity

[1]Laboratory for Synaptic Plasticity and Connectivity, RIKEN Center for Brain Science, Wako-shi, Saitama, Japan. [2]Institute of Experimental Epileptology and Cognition Research, University of Bonn Medical Center, Venusberg-Campus 1, 53127 Bonn, Germany. [3]Present address: Department of Physiology, Keio University School of Medicine, Tokyo, Japan. [4]Present address: Institute of Neuroscience, CSIC-UMH, Alicante, Spain. [5]Present address: Synapse Biology Unit, Okinawa Institute of Science and Technology Graduate University, Onna-son, Kunigami-gun, Okinawa, Japan. [6]These authors contributed equally: Thomas E. Chater, Maximilian F. Eggl. [7]These authors jointly supervised this work: Yukiko Goda, Tatjana Tchumatchenko. ✉e-mail: yukiko.goda@oist.jp; tatjana.tchumatchenko@uni-bonn.de

events or change the threshold for future synaptic plasticity[13,14]. Electrical activity can also contribute to multi-spine plasticity, including NMDA spikes and back-propagating action potentials (e.g.[15–18]). Secondary events, such as activity-dependent local dendritic translation, add an additional layer of regulation. Proposed mechanistic frameworks include synaptic tag and capture (STC)[19] and the clustered plasticity model where the influence of plasticity is confined to individual dendritic branches[20].

Many candidate molecules have been identified as mediators of inter-synaptic signalling, such as the $Ca^{2+}$/calmodulin-dependent protein phosphatase calcineurin[21,22], the $Ca^{2+}$/calmodulin-dependent protein kinase CaMKII[23,24] (but see ref. [25]), the small GTPases Ras[13] and RhoA[26], and the diffusible gas nitric oxide (NO)[22] amongst others (reviewed in[27]). Following synaptic activity, several of these molecules have been shown to diffuse away from the activated spine, spread along the dendrite, and enter nearby synapses. Other components such as Cdc42 and CaMKII are likely confined to the activated spine[25,26,28]. Functionally, CaMKII and calcineurin appear to act in parallel to readout frequency and strength of the stimulus[29] where CaMKII plays a role in LTP/sLTP (e.g., [30,31]) and calcineurin is required for LTD/sLTD[21,32]. Hence, the amount and distribution of active protein kinases and phosphatases following synaptic activity is likely to be a critical determinant for activating and allocating essential components to neighbouring spines for the downstream expression of plasticity.

Overall, the current knowledge of synaptic plasticity highlights the existence of cooperative mechanisms that can jointly upregulate the strengths of multiple synapses (e.g., spreading of plasticity factors and de novo protein synthesis) and the contribution of counter-forces that dampen the synaptic response (e.g., competition for shared protein resources). To clarify how synapses jointly coordinate their synaptic strengths following multiple plasticity-triggering events, we sought to quantitatively predict the spatio-temporal footprint of synaptic plasticity resulting from the activity of a specific set of synapses based on the known features of molecular players of plasticity. To this end, we combined experimental and modelling approaches and explored the minimal principle components that control this plasticity. Using glutamate uncaging, we systematically elicited sLTP[13,33] at variable numbers of target spines sharing the same dendrite and monitored the spine structural dynamics over time while also testing their sensitivity to pharmacological perturbation of candidate synaptic signalling molecules. In parallel, we built a mathematical model in which the action of fast-diffusing $Ca^{2+}$ ions (and/or their related molecules) and molecular dynamics within the dendrite results in sLTP or sLTD at synaptic sites depending on the context in which activity is imposed.

Our mathematical approach was motivated by a number of previously proposed models. Following the classic Bienenstock-Cooper-Monroe (BCM) sliding threshold model for LTP and LTD induction[34], Lisman[35] put forward an influential $Ca^{2+}$ threshold hypothesis that was based on experimental data (reviewed in refs. [36, 37]), which stated that the synapses undergo LTP or LTD according to the availability of $Ca^{2+}$. Several models have since extended the original concept. Notably[38], and[39] considered the Michaelis-Menten kinetics of protein kinases and protein phosphatases that are directly or indirectly regulated by intracellular $Ca^{2+}$ and promote differing LTP and LTD responses[39]. also considered the role of NMDARs as sources of $Ca^{2+}$ for triggering LTD and LTP. However, these $Ca^{2+}$-driven models typically operate on the timescale of milliseconds with an emphasis on the initial events of plasticity induction, precluding sufficient time for inter-spine protein communication via diffusion. Therefore, we adapted these model principles to match the timeframe of our experimental data. Capitalising on the $Ca^{2+}$ threshold hypothesis models, we introduced a mathematical model that coupled the fast kinetics of $Ca^{2+}$ and related molecules to slow protein dynamics. Interestingly, experimental evidence pointed to the existence not only of $Ca^{2+}$-dependent LTP and

LTD but to a third type of outcome, referred to as "no-man's land"[40,41]. This regime has escaped existing modelling approaches because it yields neither clear LTP nor LTD but falls between the two, depending potentially not just on $Ca^{2+}$ but on additional parameters. Taking into account the full spectrum of possible LTP and LTD outcomes and this third regime, we introduced a model that avoided an abrupt LTP/LTD border and instead continuously transitioned between LTP and LTD and back. Specifically, we pursued a model with a minimal number of parameters that could reproduce experimental plasticity results on the timescale of tens of minutes, along with molecular dynamics that operated on faster timescales. To this end, we considered in our model two distinct sets of molecules: a fast-diffusing set of $Ca^{2+}$-binding molecules (C) that enact sLTP and a slower-diffusing set of molecules (P) activated by $Ca^{2+}$-binding molecules that lead to an sLTD or sLTP response based on the size of the spine prior to the triggering of plasticity.

Our experimental results revealed that the number and spatial arrangement of the stimulated spines strongly affected the spine structural plasticity response. A weaker average spine potentiation was observed upon increasing the number of stimulated spines within the multi-spine paradigm, whereby the size of the average spine growth was inversely proportional to the distance to its nearest stimulated neighbours. Notably, our mathematical model could reproduce the key features of spine dynamics observed in our experiments. Collectively, our experimental and modelling results provide insights into competitive mechanisms over minutes to tens of minutes timescales, that allocate plasticity resources across spines as a function of temporal and spatial distance from the induction sites, thereby directing the global plasticity response at multiple spines sharing a dendrite.

## Results

To investigate the molecular mechanisms underlying multi-spine plasticity (that is induced by simultaneous stimulation of multiple spines), we combined experimental results from GFP-expressing CA1 neurons in hippocampal organotypic slices with a mathematical model that is based on the temporal and spatial evolution of synaptic plasticity in response to a number of different simultaneous stimulation events.

We began by imaging short stretches of apical oblique dendrites, eliciting sLTP at clusters of dendritic spines, and following the resulting structural plasticity over time. To potentiate specific dendritic spines on the same dendritic branch, we employed glutamate uncaging and systematically varied the number of stimulated spines (1, 3, 7, and 15). As reported previously, short trains of glutamate uncaging in $0Mg^{2+}$-containing artificial cerebrospinal fluid (aCSF) (e.g., [21,28,33]), resulted in robust spine structural enlargement that lasted for several tens of minutes (see Fig. 1a, b). When targeting more than one spine (i.e. for groups of 3, 7, or 15 spines), stimulation was performed quasi-simultaneously by uncaging glutamate at each spine for 4 msec and then moving to the next spine within 3 msec. This way, stimulation of a group of 7 spines, for example, was completed within 50 msec; this was then repeated at 1 Hz for 1 min, which led to robust sLTP. Details on the N numbers of animals, experiments, stimulated spines for each experimental paradigm are shown in Table 1, while the source data can be found in ref. [42].

Observing the stimulus-triggered structural dynamics of spines sharing the dendritic branch allowed us to introduce our mathematical model. At its core, the model depends on two distinct classes of molecules: C, which acts on faster timescales, and P, which is activated by C and evolves on the scale of minutes to hours. We hypothesised that sharing of these molecular resources amongst spines along a stretch of dendritic could produce differences in plasticity responses when more than a single spine was potentiated. Thus, in our model, C consists of a shared initial dendritic population, $C_d$, and a spine-specific amount, $C_s$, which is only available to the spine itself and unlocked by

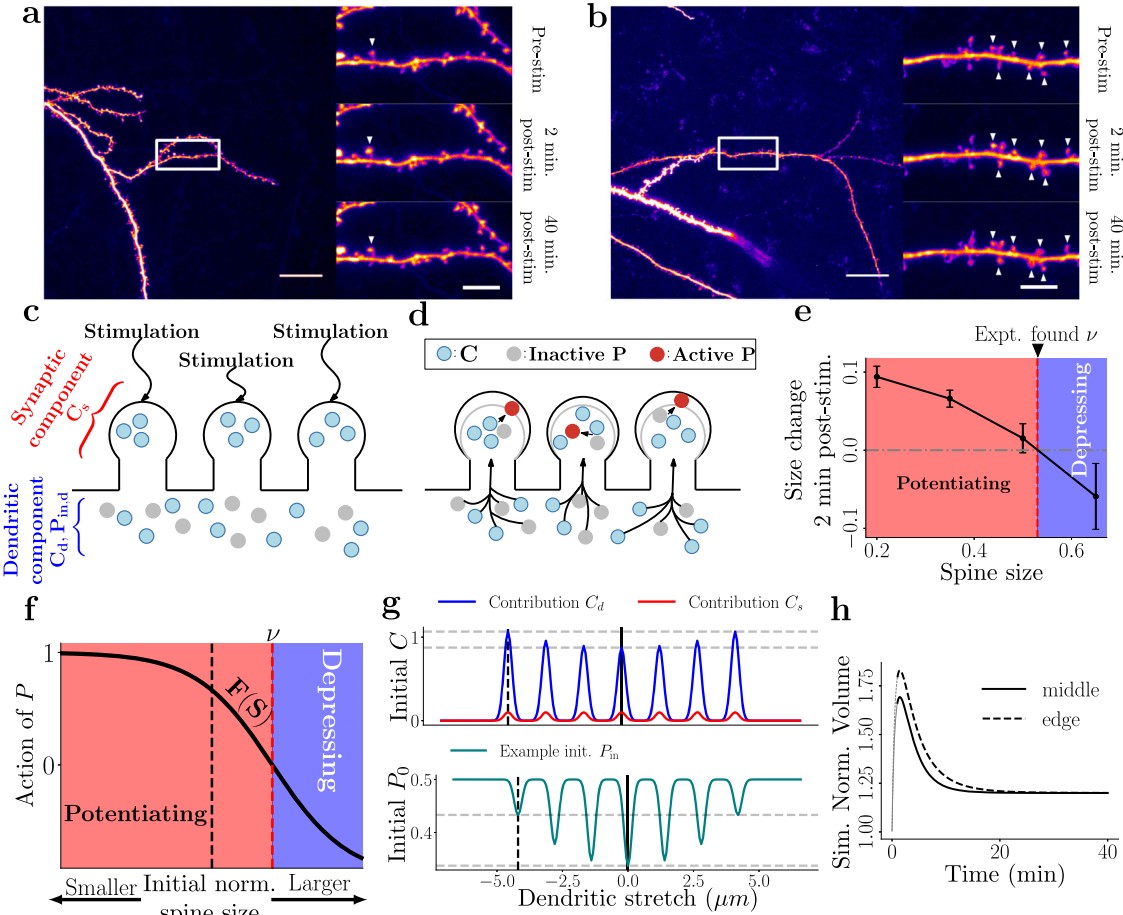

**Fig. 1 | Studying spine dynamics following sLTP induction. a, b** Example GFP-expressing CA1 neuron used for targeting 1 spine (**a**) and 7 spines (**b**). Left: Dendrite targeted for stimulation. Scale bar = 20 μm. Right: Time-lapse frames of boxed region showing the stimulated spine(s) (white arrowheads). Scale bar = 5 μm. **c** Before sLTP induction, *C* (blue circles) and inactivated *P* (grey circles) are available at spines (red bracket) and dendrite (blue bracket). **d** Upon sLTP induction, spine-specific *C* is released, while inactive *P* and dendritic *C* move towards the stimulation sites, leading to resource competition. *C* activates *P* in spines, which transitions at rate $\beta_2$ from inactive $P_{in}$ to its active form (red circles). The activation of *P* and *C* leads to spine size changes. **e** Spine size change depends on current size. *v* represents the point where the changes flip from growth to shrinkage. Data adapted from the 15 spine stimulation paradigm, Fig. 5. Error bars represent ± s.e.m. **f** Based on results seen in **e**, we model the effect of *P* (potentiating or depressing) using the activation function *F* and the spine size *S* (normalised to pre-stimulation). When the

normalised spine size is below the threshold *v* (here for illustration at *v* = 1.4), *P* induces growth (*F(S)* > 0 in the red region). However, when *S* > *v*, *F(S)* < 0 (blue region), *P* is depressing, providing a potential mechanism to avoid uncontrolled growth. **g** Top: Simulated initial distribution of *C* when simultaneously stimulating 7 spines. Red line · *C* in synaptic stores, blue line · shared dendritic component. The combination of both represents the total *C* available to the spine at stimulation. Horizontal dashed lines highlight the minimum and maximum amount of *C* (middle and edge spines, respectively). Vertical black lines denote spine locations shown in h. *bottom)* Simulated initial distribution of $P_{in}$ when simultaneously stimulating 7 spines. **h** Example simulated spine dynamics arising from the initial condition in g. Distinct plasticity behaviours emerge in middle and edge spines early after plasticity induction. Dashed grey lines at *t* ≈ 0 represent the stimulation region (not modelled).

the stimulation event. Figure 1c, d schematically summarise the model mechanisms. The spine and dendritic distribution of *C* and inactive *P* in basal conditions (Fig. 1c) changes upon the simultaneous induction of sLTP at multiple spines. The concentration of *C* increases in each stimulated spine along with the activation of *P* (expressed as an increase in *P*; red circles in Fig. 1d). The motivation of $C_s$ comes from the spine calcium transients seen in response to synaptic activity.

To numerically simulate this sudden and local increase of *C* at the stimulated spines, we used a sum of narrow Gaussians centred around each stimulation site (cf. Fig. 1g). Mathematically, this can be described as follows: given *N* stimulation events at positions $x_i$ (where *i* refers to the $i^{th}$ stimulation), the *C* distribution immediately after stimulation is then

$$C(x, 0) = \sum_{i=1}^{N} (C_s + C_d D_i) \exp(g(x - x_i)^2). \quad (1)$$

The width of the Gaussian kernel, *g*, was set to be −1000 in our study, leading to a tractable computational space while still allowing us to simulate our experimental domain. This width was then scaled to 1 μm (see supplemental Fig. S1a).

As $C_d$ represents the shared dendritic component of *C*, competition will naturally occur if multiple spines draw on this resource. To model this competition, we have introduced the term $D_i$, a factor that modifies the amount of available $C_d$ at spine *i*. The form of $D_i$ is defined by the following three model assumptions: *i)* the total number of stimulation sites, *N*, will decrease the amount of $C_d$ at each stimulation site by a factor of 1/*N*, *ii)* the distance between the stimulations has a fundamental effect on competition such that the closer the stimulation events are in space, the stronger the contribution of competition and the 1/*N* term, whereas as the distance between stimulations increases, the effect of competition should decrease, and *iii)* the number of spines that are not directly stimulated, $\hat{N}$, but are sufficiently close to multiple stimulation sites (which we have defined to be within 2 μm in

## Table 1 | Details of the experimental dataset

| Experiment | Reference figures | # of animals | # of expts | Stimulated spines |
|---|---|---|---|---|
| Three spine | Figs. 2 and 6 | 4 | 7 | 21 |
| Three spine (sham) | Fig. 2 | 6 | 10 | 30 |
| Single spine | Figs. 3 and 6 | 8 | 13 | 13 |
| Single spine (sham) | Fig. 3 | 6 | 13 | 13 |
| Single spine (AIP) | Figs. 3 and 6 | 4 | 6 | 6 |
| Single spine (FK506) | Figs. 3 and 6 | 5 | 6 | 6 |
| 7 spine | Figs. 4 and 6 | 9 | 25 | 174 |
| 7 spine (sham) | Figs. 4 and 6 | 4 | 7 | 49 |
| 7 spine (Distributed) | Fig. 4 | 4 | 10 | 68 |
| 15 spine | Figs. 5 and 6 | 6 | 13 | 181 |
| 15 spine (sham) | Fig. 5 | 5 | 7 | 102 |

this study), will also compete for the resource $C_d$. Specifically, we can formulate $D_i$ as

$$D_i = \frac{1+d_i}{\underbrace{N}_{\text{Competition among stim. spines}} + \underbrace{\hat{N}}_{\text{Nearby unstim. spines}}}, \qquad (2)$$

The first term in the denominator represents the competition among stimulated spines. However, this amount is modified by the parameter $d_i$, where $d_i$ represents the spatial effect of stimulation arrangement and is defined by the normalised distances (between 0 and 1), which indicates how close each of the other stimulated spines is to the stimulation point $i$. When the distances are small, $d_i$ will be close to 0, and we will tend towards the $1/N$ competitive rule. If the distances are large, we expect $d_i$ to tend towards $N-1$ to negate competition. Therefore, to define $d_i$, we used a sum of normalised absolute difference metrics:

$$d_i = \sum_{j=1, j\neq i}^{N} \left( \frac{|x_i - x_j|}{1 + |x_i - x_j|} \right)^\lambda, \qquad (3)$$

where $\lambda$ controls the shape or behaviour of each of the terms in $d_i$ according to the differences between $x_i$ and $x_j$. Smaller $\lambda$ will strengthen the effect of competition over large distances, while larger $\lambda$ will prioritise spatially close stimulation events.

The second term in the denominator of eq. (2) represents the heterosynaptic unstimulated spines that will also compete for resources. This term was motivated by experimental observations (e.g., [21]) in which neighbouring spines underwent plasticity when stimulated spines surrounded unstimulated spines and, presumably, resources needed for the expression of plasticity overlapped from multiple nearby stimulation sites (for a model simulation, see supplementary Fig. S1b).

In summary, as sLTP is induced at an increasing number of sites, more spines (both homosynaptic and heterosynaptic) will compete for the resources required for potentiation. When sufficiently many spines are stimulated, a saturation point is reached by the depletion of the store of $C_d$, and the uncontested synaptic component, $C_s$, becomes the primary source of $C$. Additionally, as the distance between $x_i$ and the other stimulation points increases, $D_i$ will tend towards 1; that is, when stimulations are far apart in space, spines will not compete for the same dendritic resources, and thus, the dynamics will decouple.

Additionally, there will be minimal overlap in stimulations such that heterosynaptic competition will also be minimal. An example of how this competitive rule leads to the initial $C$ distribution is depicted in Fig. 1g.

Finally, we introduce the dynamical equation describing $C$

$$\frac{\partial C}{\partial t} = \alpha_1 \frac{\partial^2 C}{\partial x^2} - \alpha_2 C, \qquad (4)$$

where the dynamics of $C$ are driven by a diffusive component at rate $\alpha_1$ and, simultaneously, degraded at a rate $\alpha_2$. The diffusion equation indicates that $C$, introduced at $x = x_i$, diffuses in the local vicinity, leading to plasticity events at neighbouring synapses.

So far, we have considered only the fast $C$ molecules and their distribution across spines. Next, we introduce the slow molecule resource, $P$, whose production is promoted by the local accumulation of $C$ and the resulting interaction with the inactive form, $P_{in}$. In our model, $C$ always has a net potentiating effect, however, there is a wealth of experimental results that see both potentiation and depression[3,21,22]. Therefore, $P$ should, in some form, elicit depression. Inspired by the calcium threshold hypothesis[35,37], we define $P$ as a dual-role molecule(s) that can, under certain circumstances, lead to potentiation and, in others, cause depression. As shown in Fig. 1e, (adapted from[43]), given that small spines tend to grow and large spines tend to shrink, we link the dual nature of $P$ to the basal spine size prior to stimulation: if the spine is smaller than a certain threshold $v$, then $P$ will have a potentiating effect to encourage growth in response to the stimulation. Conversely, if the size of the spine is above $v$, then $P$ will have a net depressing effect, introducing a feedback mechanism that prevents spines from growing too large.

By design, we assume that there is no initially available $P$ and that this is only generated post-stimulation through the interaction of $P_{in}$ and $C$. Therefore, we define only a baseline amount of $P_{in}$, $\rho$, at the time of stimulation. Initially, $\rho$ was fixed across experimental paradigms. However, as we induced more simultaneous stimulation events, the subsequent decrease in spine size immediately after the stimulation was more gradual. This implied that either less $P$ was available, or the effect of $P$ was tempered. As the latter option would imply that the spines had some inherent knowledge of the stimulation paradigm and would thus alter their physical characteristics (i.e., less translation of $P$ or faster degradation), the former option of initially less $P$ appeared more reasonable. Therefore, we hypothesise that $P_{in}$ is initially depleted by the stimulation event, leading to a smaller local and global amount of $P$ as more spines are stimulated. In other words, we expect a store of $\rho$ throughout the dendrite, and at the stimulation sites, it is depleted inversely to $C_d$. An example plot is shown in Fig. 1g. Mathematically, we express this as:

$$P_{in}(x, 0) = \rho(1 - \sum_{i=1}^{N}(1 - D_i) \exp\left(g(x - x_i)^2\right). \qquad (5)$$

The evolution of $P$ and $P_{in}$ is given in the form of a reaction-diffusion model

$$\frac{\partial P_{in}}{\partial t} = \beta_1 \frac{\partial^2 P_{in}}{\partial x^2} - \beta_2 P_{in} C, \qquad (6)$$

$$\frac{\partial P}{\partial t} = \beta_1 \frac{\partial^2 P}{\partial x^2} + \beta_2 C P_{in} - \gamma P, \qquad (7)$$

where $\beta_1$ is the rate of diffusion of $P_{in}$ and $P$, $\beta_2$ defines the rate at which $P_{in}$ is transformed into $P$ and $P$ is degraded at rate $\gamma$. We discuss possible biological candidates of $P$ in the supplemental text and their diffusion coefficients in supplemental Fig. S2.

In our model, the concentration of $P$ grows as long as the accumulation rate, $\beta_2$, is larger than the diffusion and degradation rate. Whether a spine grows or shrinks depends on the concentrations of both $C$ and $P$ in line with experimental reports (e.g.,[44]).

Early models often showed characteristically discrete states with instantaneous switching between the potentiating and depressing states. However, such a mechanism does not account for the presence of "no-man's land" regime[36], where an intermediate region that shows neither LTP nor LTD occurs. Thus, we introduce a decision variable $F$, which ranges between $+1$ and $-1$ and depends on the current spine size (normalised by the baseline):

$$F(S) = -\tanh(\phi(S - \nu)), \tag{8}$$

where $\phi$ refers to the strength of the switch from potentiation to depression (the higher $\phi$, the shorter the transition phase), and $\nu$ is the threshold where the switch occurs. This function was inspired by Fig. 1e, where a direct dependence between the spine size and its subsequent change due to stimulation was observed. For completeness, let us note that in the experimental data we normalised the initial spine size $S$ by the pre-stimulation size because we model average plasticity response post-stimulation. We capture the functional form observed in Fig. 1e using the function $F(S)$, an example of which, with the threshold $\nu$ set at 1.4, is shown in Fig. 1f. Next, we introduce the final component of the model, the normalised spine size $S$, which is determined by $C$, $P$, and $F$:

$$\frac{\partial S}{\partial t} = \zeta_1 C + \zeta_2 P \cdot F(S). \tag{9}$$

Here, $\zeta_1$ and $\zeta_2$ are parameters determining the biological susceptibility of the spines to $C$ and $P$, respectively. Illustration of spine dynamics given the initial conditions in Fig. 1g is shown in Fig. 1h. The mechanics of $C$ and $P$, as a function of space, and the corresponding effects on heterosynaptic spines in and outside clusters of stimulated spines is shown in supplemental Fig. S1a. We note that $C$ is highly localised in comparison to $P$ and degrades at a much faster rate, which is in line with our experimentally recorded $Ca^{2+}$ dynamics (supplemental Fig. S3).

In summary, the model makes several experimentally testable predictions.

1. Given the nature of the initial condition of $C$ and $P_{in}$, we predict that stimulated spines at the edge of the cluster, which only compete with other stimulated spines on one side, might exhibit a stronger extent of potentiation than spines in the centre of the cluster, which compete with stimulated spines on both sides.
2. Single spines, which do not compete for resources, would be expected to exhibit a stronger initial potentiation response but a more rapid decline back to baseline due to the lack of competition for both $C$ and $P$.
3. When proteins involved in sLTP are (pharmacologically) inhibited, changing either the threshold $\nu$ or the effect of $P$ on the spines, $S$, is expected to reproduce the resulting plasticity dynamics.
4. Relative to stimulating a small number of spines, when a sufficiently large number of spines are stimulated simultaneously, the initial potentiation would decrease on average, but the extent of subsequent spine shrinkage would also decrease due to reduced availability of $P$. This mechanism might facilitate sLTP, as $P$ engages in a negative feedback mechanism that is only deactivated if sufficient spines are stimulated and thus compete for $P$.
5. When the distance between spines increases, the sLTP dynamics of individual spines should be decoupled from one another and revert to single-spine dynamics.

## Dynamics of sLTP in clusters of stimulated dendritic spines

To understand how groups of potentiated spines interact, we first chose to stimulate three dendritic spines sharing a short region of a dendrite (Fig. 2a–c; white arrowheads indicate stimulated spines, average distance of $\approx 2\,\mu m$). A plot of the normalised spine sizes and a resulting fit of the model parameters as a function of time shows that glutamate uncaging at three spines leads to robust short-term spine growth followed by a decline to a spine size that stabilises above the baseline level, which is consistent with sLTP (Fig. 2d). The fits shown suggest that the dynamics of the average experimental spine are well reproduced by the average dynamics of the model spines. This is further supported by both the normalised mean square error (NMSE: the average of the mean square error of each time point divided by that time point's variance) and weighted $R^2$ ($R_w^2$: a variation of the standard $R^2$ to study the goodness-of-fit for a non-linear predictor) being close to unity (see the methods section, under statistical definitions for further details). A separate set of sham-stimulated experiments was performed as controls where MNI-glutamate was omitted from the aCSF. The sham-stimulated spines did not undergo sLTP, although they remained stable over the recording period, indicating a lack of adverse effects of uncaging laser pulses.

Using this simple 3-spine stimulation scenario, we investigated whether spines on the edge and middle of clusters interact as they compete for plasticity components. We specifically examined whether spines on the edge of a cluster potentiate more strongly than those in the middle, as suggested by the model.

To understand how the different cluster sizes and the distances between spines may affect the plasticity dynamics, we studied the statistics of the spine-to-spine distances in our 3-spine stimulation paradigm and calculated the predicted model dynamics for the minimal (2.2 µm) and the maximal (4.2 µm) experimentally observed cluster size. Fig. 2e shows the model prediction for the edge (green) and middle (blue) spines for the mean cluster size (3.2 µm; bold line), in which cluster sizes were obtained experimentally, along with the model predictions for the minimal and the maximal cluster sizes for edge (green) and middle (blue) spines, represented by a shaded area. The plot illustrates a way of setting limits to the variability expected in our modelled spine dynamics. Notably, there is a substantial overlap between the middle and edge spines, which suggests that our present experimental conditions for the 3-spine stimulation are not likely to promote differences between the plasticity behaviour amongst the stimulated spines. Consistently, our experimental data do not show substantial differences between edge and middle spines (Fig. 2f–h). Altogether, these results indicate that when 3 spines are stimulated, the spine-spine competition for the available resources is not sufficient to cause significant differences between the plasticity of edge and middle spines. This prompted us to study the effects of further increasing the number of stimulated spines on spine plasticity dynamics, to which we will return later.

## The single spine response is driven primarily by the slower molecular dynamics of $P$

Our model suggests that single spines, in the absence of competition imposed by near simultaneous activation of other nearby spines, will exhibit dynamics that are dominated by rapid growth followed by a similarly rapid decline, distinct from the dynamics of multi-spine plasticity. To explore this point, we next elicited sLTP by targeting only single spines (see supplemental Fig. S4 for specificity of spine stimulation). Glutamate uncaging of single spines resulted in robust sLTP (Fig. 3a, top row, pooled data plotted in Fig. 3d, blue line). To gain further insight into the underlying mechanisms, we turned to calcium and calmodulin-dependent enzymes, CaMKII and calcineurin, which previously have been shown to regulate LTP and LTD, respectively[22–24]. In one set of experiments, single spine glutamate uncaging was performed in the presence of a specific competitive inhibitor of CaMKII,

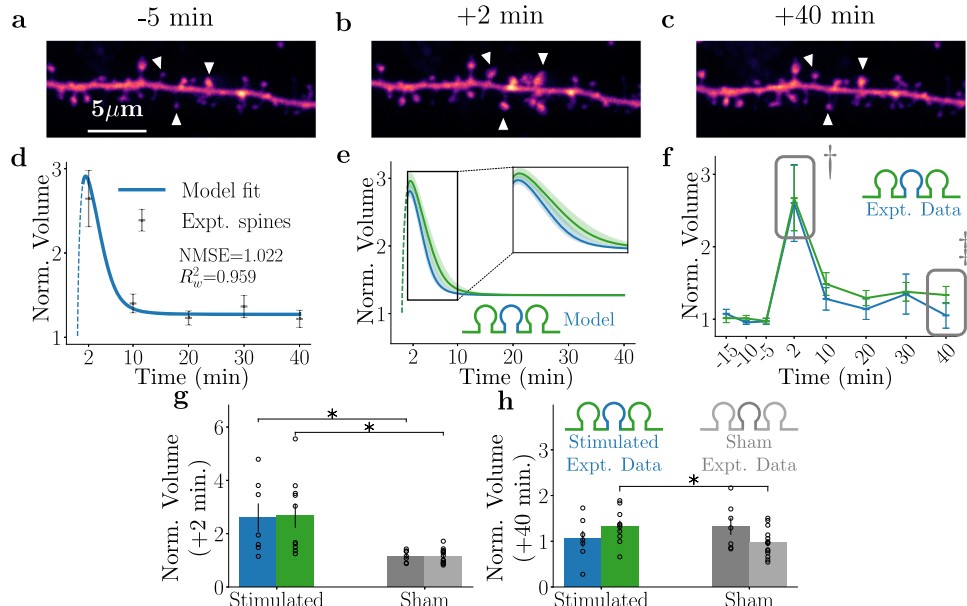

**Fig. 2 | Spine dynamics of three clustered sLTP events. a–c** Example images of a three-spine stimulation experiment showing 5 min before, 2 min after, and 40 min after sLTP induction. Stimulated spines are denoted by white arrowheads. **d** Normalised (against baseline) stimulated spine size (grey points) compared to our model fit (blue line). The model parameters obtained in this fit are kept constant across all subsequent experiments with varying number of stimulated spines. Thin dashed line refers to the stimulation region where the model predictions only have limited validity. **e** A plot of spine size dynamics predicted by the model for the edge (green) and the middle (blue) spines for the mean cluster size (3.2 μm; bold lines) along with the range of spine size dynamics for the minimal (2.2 μm) and the maximal (4.2 μm) cluster sizes, represented by the shaded area. The minimal and maximal cluster sizes associated with the experimental data help set limits to the variability expected in our modelled spine dynamics. Inset: the boxed section of the

plot at an expanded timescale. **f** Stimulated spine volume changes for edge (green) and middle (blue) spines are shown separately. Data in grey boxes marked (†, ‡) in (**f**) are displayed for detailed comparison in (**g**) and (**h**), respectively. **g** Normalised growth of middle (blue) and edge spines (green) at $t = 2$ min compared against sham (dark grey for middle and light grey for edge spines). $p$ values for edge and middle spines are $= 0.029$ and $0.003$, respectively. **h** Normalised growth at $t = 40$ min, with the same colour code as in (**g**). Middle spines are not significantly different compared to the sham experiment. $p$ values for edge and middle spines are $= 0.0314$ and $0.3$, respectively. In all figures above, * refers to $p < 0.05$ of a two-sided $t$ test, accounting for multiple comparisons and error bars represent ±s.e.m. Additionally, $N = 7$ middle spines and 14 edge spines for the control, and $N = 10$ middle and 20 edge spines for the sham for all above figures.

myristoylated autocamtide-inhibitory peptide (AIP, 5 μm[45]) to test if inhibiting CaMKII produced an effect that was similar to enhancing the effect of P. In a separate set of experiments, we inhibited calcineurin using FK506 (2 μm[46]). As calcineurin activity was previously reported to promote sLTD, inhibiting the action of calcineurin could mimic the effect of lessening the contribution of $P$ in our model.

Figure 3a–c shows an example dendritic segment containing a stimulated spine (white arrowhead) at 5 min before (a), 2 min after (b), and 40 min after sLTP induction (c). The spine size changes for all conditions (control, AIP, FK506) across the entire experiment are summarised in Fig. 3d. At 2 min post-stimulation, both control and calcineurin-inhibited spines show significant enlargement, which is lessened (resulting in loss of significance) in the CaMKII-inhibited and absent for sham spines (Fig. 3e). Spines in which calcineurin was inhibited are still able to undergo sLTP and show different decay dynamics compared to control stimulated spines (Fig. 3d).

Figure 3g illustrates that the model mostly recovers the trend in spine dynamics of the single spine stimulation experiment in the control condition. In comparison to the 3-spine experiment, our goodness-of-fit metrics are worse (in particular $R_w^2 = 0.74$). Nonetheless, the dynamics align well with one of the model's predictions, that single spines rapidly grow (as there is no competition for $C$) but then equally rapidly shrink back to a lower value of approximately 1.3 (as there is no competition for $P$). This leads to the following implication: under conditions of low activity, $P$ could act as a negative feedback mechanism that counteracts LTP. Only when a sufficient number of spines is potentiated, which leads to increased competition and depletion of $P$, would the inhibitory effect of $P$ on LTP be relieved to achieve robust potentiation. These robust LTP dynamics will have

less potentiation (due to the additional competition for $C$) but subsequently less decay in the long term.

Next, we modified the ratio of $\zeta_1/\zeta_2$ (the relative strength of $C$ and $P$) and the threshold, $v$, to model the effect of the inhibitors on spine dynamics. By increasing the ratio of $\zeta_1$ and $\zeta_2$ and thus decreasing the effect of $P$ on the dynamics, we recovered the experimentally measured data for calcineurin inhibition in our model (Fig. 3h). Our prediction for this situation performs slightly better in terms of prediction accuracy (NMSE) in comparison to the control (higher NMSE), while sacrificing the ability to account for the variance of the data (lower $R_w^2$). Nonetheless, a reasonable goodness-of-fit measurement could be obtained without altering the potentiation/depression threshold of $P$. This suggests that calcineurin might not significantly affect the initial and final states of spine dynamics; instead, calcineurin could contribute primarily to the decay of spine volume increase back to the baseline after stimulation, and its inhibition slows this decay rate.

In modelling the spine plasticity responses with inhibited CaMKII activity, we considered as a first step the constant solution corresponding to $C = 0$ and $P = 0$ (Fig. 3i, dashed light green line). However, the relatively high NMSE value of 1.143 and negative $R_w^2$ suggested that this model approximation was not able to capture the mean dynamics nor the up and down trends contained in the data. As a next step, we decreased the ratio of $\zeta_1$ and $\zeta_2$, which increases the effect of $P$ in the model and also lowered the $P$ threshold in eq. (8). With these changes we could improve the approximation of the experimental dynamics (NMSE = 1.049, $R_w^2 = 0.549$) from the previous constant model. This hints at the possibility that blocking CaMKII not only removes the potentiating component of $P$ but could also increase its depressing effect.

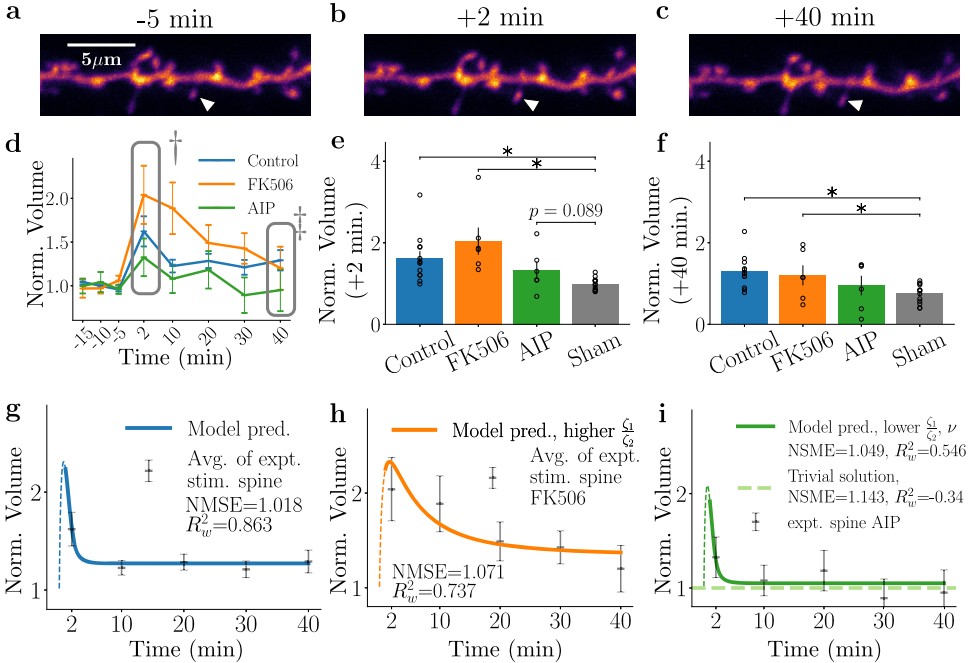

**Fig. 3 | Model fits to the single spine sLTP in control and in the presence of pharmacological inhibitors. a–c** Example images of a single spine experiment showing 5 min before, 2 min after, and 40 min after inducing sLTP. The stimulated spine is marked with a white arrowhead. **d** Temporal dynamics of sLTP in control condition (blue) and in the presence of the CaMKII (green) and calcineurin (orange) blockers, AIP and FK506. The data in the grey boxes marked with (†, ‡) are displayed for detailed comparison in e and f, respectively. **e** Normalised spine volume change relative to the baseline of all experimental conditions at $t = 2$ min. *$p$ values = 0.0036 and 0.0015 after multiple-pairwise correction for the Control and FK506 conditions, respectively. **f** Normalised spine volume change relative to the baseline of all experimental conditions at $t = 40$ min, as shown for (**e**). *$p$ values for control and FK506 are 0.03 and 0.0004, respectively. **g, h** Model prediction of spine dynamics superimposed on the experimental data in control conditions (**g**) and in

the presence of FK506 (**h**). Error bars represent ± s.e.m. Experimentally observed spine dynamics in FK506, particularly the slow decrease of the initial potentiation is reproduced by the model upon decreasing $\zeta_2$ that serves to lower the effect of $P$. **i** Trivial solution (i.e., no dynamics in dashed light green) and model spine dynamics superimposed on the experimental spine dynamics in AIP (dark green line). By increasing $\zeta_2$ to enhance the influence of $P$ on the spine size and by decreasing the threshold, $\nu$, the model captures the overall trend of spine dynamics observed in experiments when CaMKII is blocked. In all figures above, * refers to $p < 0.05$ of a two-sided $t$ test, accounting for multiple comparisons, and error bars represent ± s.e.m. $N = 13$ control, 13 sham spines, 6 AIP, and 6 FK506 spines, and the thin dashed lines refer to the stimulation region, which is outside of the temporal range of the model validity for all the above figures.

## Increasing cluster size increases competition for synaptic resources

So far, both our experimental observations and modelling results suggest that spines within a cluster could compete, as supported by i) the kinetics of their potentiation following stimulation being distinct from potentiation of single spines in isolation, which decays faster than clustered potentiated spines, and ii) the fact that effects of inhibiting key proteins involved in plasticity on the magnitude and dynamics of sLTP are reproducible in the model by modifying $P$, one of the resource factors needed for plasticity. To further test the interactions between neighbouring spines in the process of sLTP, we increased the number of stimulated spines from groups of 3 spines to groups of 7 spines sharing the same dendritic segment. Our model predicts that simultaneous stimulation events for larger clusters (both in size and number) will affect the plasticity dynamics. Figure 4a–c (top) shows example images from a 7-spine experiment. White arrowheads indicate spines targeted for glutamate uncaging. Notably, non-stimulated spines are present between the stimulated spines, which would further decrease the availability of $C$ and $P$ through competition (see equations (1) and (5)). In line with the fourth prediction (see above), this spatial arrangement of the 7-spine stimulation, when compared to the 3-spine experiment, should lead to a lower initial potentiation but a slower subsequent decline following the initial potentiation.

The experimentally measured spine dynamics and the model prediction (here, the average number of close heterosynaptic spines was estimated to be ≈7; Fig. 4d) show a trend of spine dynamics that is different from the 3-spine experiment (Fig. 2d). Robust growth of

targeted spines was observed, and even 40 min after the stimulation the spine volume increase remained significant compared to the control sham experiment (Fig. 4e).

Following the approaches used to model the 3-spine stimulation experiment (cf. Fig. 2), we studied the plasticity response of the edge and middle spines in the experiment for the 7-spine stimulation experiment. Figure 4f displays the plasticity response of the three inner spines (middle spines: blue) alongside the plasticity response of the four outermost spines (edge spines: green). To compare model predictions with recorded data, we first assessed the distribution of minimal spine-to-spine distance (cf. Fig. 4g) and then calculated the response dynamics 25th percentile (5.6 μm) and 75th percentile (14.1 μm) cluster size for the 7-spine experiment, with the central line defining the average cluster size (≈10.6 μm). The normalized spine volume increase between the edge and the middle spines showed significant overlap in the early stages of the predicted dynamics (2 and 10 min), whereas, at later time points, the potentiation of the edge and the middle spines became more distinct. We then compared the modelling result to the experimental data. In alignment with the model prediction, the edge spines trended larger than the middle spines from +20 min onwards, and significant differences were observed at the +30 and +40 min timepoints (Fig. 4f). This plasticity difference between the edge and the middle spines that was not discernible in the three-spine experiment, implies that imposing a condition that drives an increased competition for resources by increasing the number of stimulated spines, not only has global effects but impacts local availability for resources. In other words, an abundance of $P$ and a lack of $C$ become

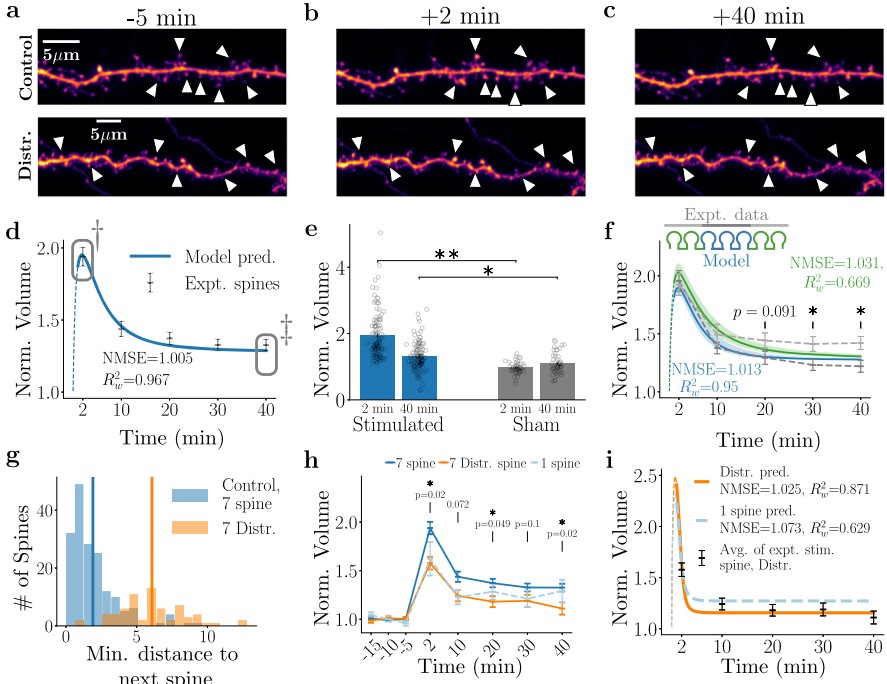

**Fig. 4 | Spine dynamics following simultaneous induction of sLTP at 7 spines.**
**a–c** Example images of a 7-spine experiment at 5 min before, 2 min after, and
40 min after stimulation. Stimulated spines are marked with white arrowheads.
Top; clustered, bottom; distributed. **d** Normalised spine changes of clustered sti-
mulated spines (grey points). By reducing the amount of initially available $C$ and $P$,
the model can largely reproduce the experimental dynamics. Data in the grey boxes
(†, ‡) in (**d**) are displayed in (**e**). **e** Normalised spine change at $t = +2$ min and
$t = +40$ min for clustered stimulation experiments compared against a 7-spine sham
experiments. **$p = 1 \times 10^{-16}$ and *$p = 0.001$ **f** When the edge (green) and the middle
(blue) spines are separated, the model predicts a significant difference for time-
points $> 10$ min. Experimental data show a trend differentiating edge (light grey)
and middle (dark grey) spines at $t = 20$ min, which becomes more significant by
$t = 30$ and 40 min (*$p = 0.04$ and 0.03, respectively). **g** Distribution of minimum
inter-spine distance between stimulated spines for clustered (blue) and distributed
(orange) experiments. Vertical lines represent the mean **h** Normalised spine sizes
for the seven clustered (dark blue), seven distributed spines (orange), and single
spine (light blue, dashed) experiments. Increasing the distance between stimulated
spines recovers the plasticity response observed in 1-spine experiments. *$p < 0.05$
between clustered and distributed 7 spines. There is no significant difference
between the distributed and single-spine dynamics. **i** The predicted dynamics of
the distributed seven-spine case (orange line) and the single-spine case (dashed
blue line) are similar. In all figures above, the p-value refers to the result of a two-
sided $t$ test, accounting for multiple comparisons, error bars represent ± s.e.m, and
the thin dashed lines refer to the stimulation region, which is outside of the tem-
poral range of the model validity. For all figures $N = 74$ middle and 100 edge spines
for the control, 21 middle and 28 spines for the sham and $N = 68$ distributed spines.

more pronounced at certain spines compared to others, which is in line
with the first model prediction (see above and Supplemental Fig. S5).

Finally, to address our last model prediction regarding the
decoupling of spines that are sufficiently spatially separated, we per-
formed 7-spine stimulation experiments where we doubled the size of
our imaging region and stimulated groups of 7 spines sharing the same
dendritic branch but now separated by several microns instead of
being clustered (see Fig. 4a–c: lower panels, "Distr."). We call this the
distributed 7-spine condition. These experiments stimulated 7 spines
quasi-simultaneously as with the control clustered experiments,
although the average inter-spine distance was three times larger than
the control clustered 7-spine experiment (Fig. 4g). The effect of spatial
confinement of stimulated spines is summarised in Fig. 4h. Overall, the
potentiation for seven distributed spines was substantially weaker
compared to the potentiation of control clustered spines, where the
changes in the spine volume following stimulation were more closely
aligned to the dynamics observed for the single spine stimulation.
Therefore, the spatial arrangement of stimulated spines is a key factor
for the plasticity outcome.

Altogether, the temporally and spatially clustered triggering of
sLTP at spines along a dendritic branch increases the size of early
potentiation to a degree that is consistent with the model prediction,
but only if they occur close in space. Moreover, as observed both
experimentally and in the model, increasing the distance between
stimulated spines recapitulates the spine plasticity dynamics of the
single-spine stimulation model, and not the clustered multi-spine

model. Nonetheless, as indicated in Fig. 4i, the model dynamics pre-
dicted for the single spines (fully decoupled) achieves lower goodness-
of-fit values (both for NMSE and $R_w^2$) in comparison with the prediction
of the distributed 7 spine model, especially for points after the decay
post-growth. This observation hints at the possible presence of fast
spine-to-spine interactions that could persist over slightly larger dis-
tances represented in the distributed spine experiments (Fig. 4h) that
are absent in single spine stimulation.

## Spine plasticity responses are altered by a reduction in shared resources

Thus far, with an increase in the number of stimulated spines from one,
to three, to seven target spines, we observed that the competitive
element increased in line with the model predictions, leading to a
lower initial potentiation but a more robust potentiation over a longer
time span of tens of minutes. In order to assess the behaviour of spines
as resources become scarce, we next potentiated fifteen closely loca-
ted spines sharing the same dendritic segment. Figure 5a–c illustrates a
representative example of the fifteen spine experiments. The nor-
malised spine volumes for the stimulated spines across time are shown
in Fig. 5d. The continuous line represents a model prediction and
indicates that it is in close agreement with the experimental data. This
suggests that upon stimulating a large number of spines over a limited
stretch of a dendritic branch, each spine increasingly relies on its
synaptic stores of $C_s$ to potentiate and, at the same time, is not hin-
dered by $P$ as it becomes increasingly scarce. The resulting robust

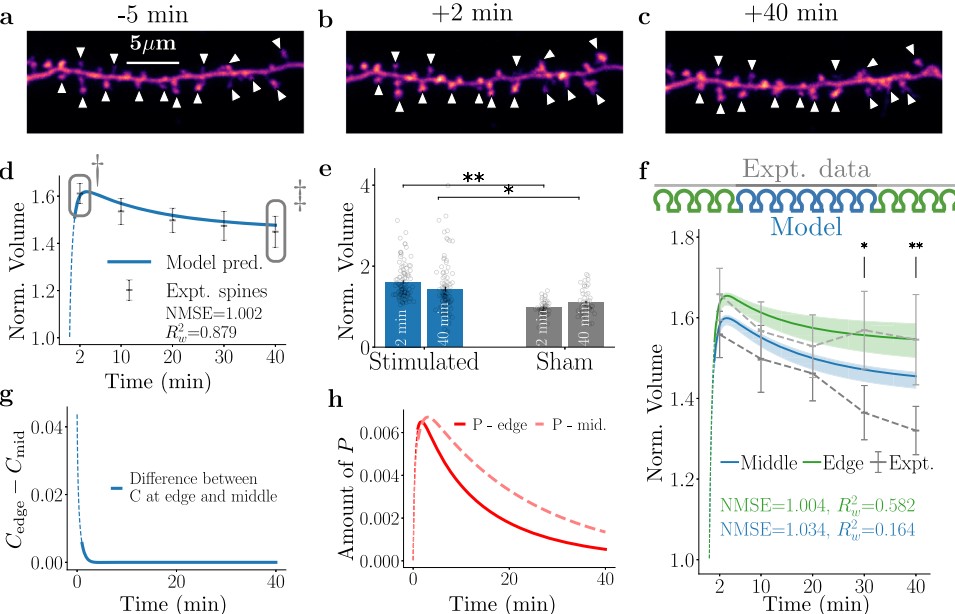

**Fig. 5 | Simultaneous induction of sLTP at 15 spines is associated with a reduced plasticity response in the early phase after stimulation. a–c** Example images showing 15 stimulated spines at 5 min before, 2 min after, and 40 min after inducing sLTP. The stimulated spines are marked by white arrowheads. **d** Normalised spine volume of stimulated spines (grey points). By reducing the amount of initially available $C$ and $P$ according to the model equations (no refitting), the model can capture the experimentally measured dynamics. Data in grey boxes (†, ‡) are shown in (**e**). **e** Normalised stimulated spine size at $t = +2$ and $t = +40$ min compared against a 15-spine sham group. **$p = 5 \times 10^{-18}$ and *$p = 0.001$ (**f**) 15 spines separated into 8 edge (green) and 7 middle spines (blue) and modelled by taking into consideration the variability in their pairwise distances. Our model prediction shows increasing differences that become more prominent for later timepoints. Experimental data also shows a significant difference between the edge (light grey) and middle (dark grey) spines for $t = 30$ min and $t = 40$ min. **$p = 3 \times 10^{-17}$ and *$p = 0.001$ **g** Difference in the amount of $C$ at the edge vs middle spines reveals that the edge have initially more $C$ (less competition) and thus potentiates more strongly. **h** Temporal dynamics of $P$ for the edge (dark red) and middle spines (light red). We note an increased pooling of $P$ at the middle spines due to slower dynamics. This additional $P$ also leads to a more rapid decrease in spine volume immediately after stimulation. In all figures above, the $p$ value refers to the result of a two-sided $t$ test, accounting for multiple comparisons, error bars represent ± s.e.m, and the thin dashed lines refer to the stimulation region, which is outside of the temporal range of the model validity. For all figures $N = 92$ for the middle and 89 edge spines for the control and $N = 49$ middle and 49 edge spines for the sham.

potentiation is maintained at 40 min after stimulation with no significant decline from the initial potentiation response (Fig. 5e). We note that this behaviour is maintained even when only considering those spines that have an initial size similar to those found in the single spine experiment (see Supplemental Fig. S6), implying that the number of co-stimulations is a much stronger effect than the individual spine configuration.

Finally, we sought to investigate whether the condition of increased demand for shared resources and heightened competition by simultaneously stimulating 15 spines, would change the plasticity dynamics. To this end, we generated a set of model predictions using the same model parameters as above and stimulating 15 spines in total whose smallest, mean, and largest cluster sizes were 20 μm, 28.3 μm, and 30.6 μm, respectively. In addition, we separated the 15 spines into 4 edge spines on either side flanking the 7 middle spines, and compared the predicted dynamics (Fig. 5f) with the experimentally observed results. We note that the model has good NMSE values in both edge and middle cases, but the $R_w^2$ value shows that the experimental variability is missed by the model. Additionally, the edge spines (4 outermost spines on each side) showed a stronger potentiation than the middle spines (7 inner spines) at +30 and +40 min post-stimulation, which was overall in agreement with the model prediction. These observations are consistent with $C$ and $P$ dynamics of the model (Fig. 5g, h). As defined by the initial condition, the edge spines start with a higher $C$ because they have slightly more neighbouring resources than the middle spines (see positive values for the difference between $C$ at edge and middle in Fig. 5g). $C$ then diffuses away rapidly, such that after 2 min the level of $C$ available to edge spines becomes comparable to the spines in the middle of the cluster. In contrast, the

behaviour of the variable $P$ is slightly different because it evolves on longer timescales (see Fig. 5h). As stated above, $C$ is initially available at the edge spines (due to less competition), and consequently, more $P$ is subsequently generated at those edge spines. However, for timepoints past 2 min, $P$ in the middle spines accumulates due to the diffusion of $P$ from spines on either side of the middle spine that are closer to the edge (in addition to the local activation of $P$) and the higher level of $P$ is coupled with the lower degradation of $P$. This pooling of $P$ amongst middle spines affects their spine plasticity dynamics over longer timescales, which leads to lower potentiation and more rapid depression back to the baseline compared plasticity of the edge spines (see Fig. 5f). Altogether, both our model predictions and experimental findings are consistent with the hypothesis that the increased competition in the 15-spine paradigm leads to larger differences in the plasticity responses within the stimulated cluster compared to plasticity responses in the 3-spine and 7-spine cases.

## Competition among stimulated spines alters plasticity response dynamics

Having demonstrated the predictive power of the model for plasticity protocols with a growing number of spines, we now consider a comparison of the different experiments. To this end, we summarised the experimental data and model dynamics for the average response of the stimulated spines in Fig. 6a, b, respectively. Notably, spines in the 3-spine paradigm show the strongest initial potentiation following stimulation, subsequently declining to stabilise at above baseline levels. In the single-spine paradigm that does not involve any competition for resources, the stimulated spine also potentiates reasonably strongly and displays the fastest decay of the

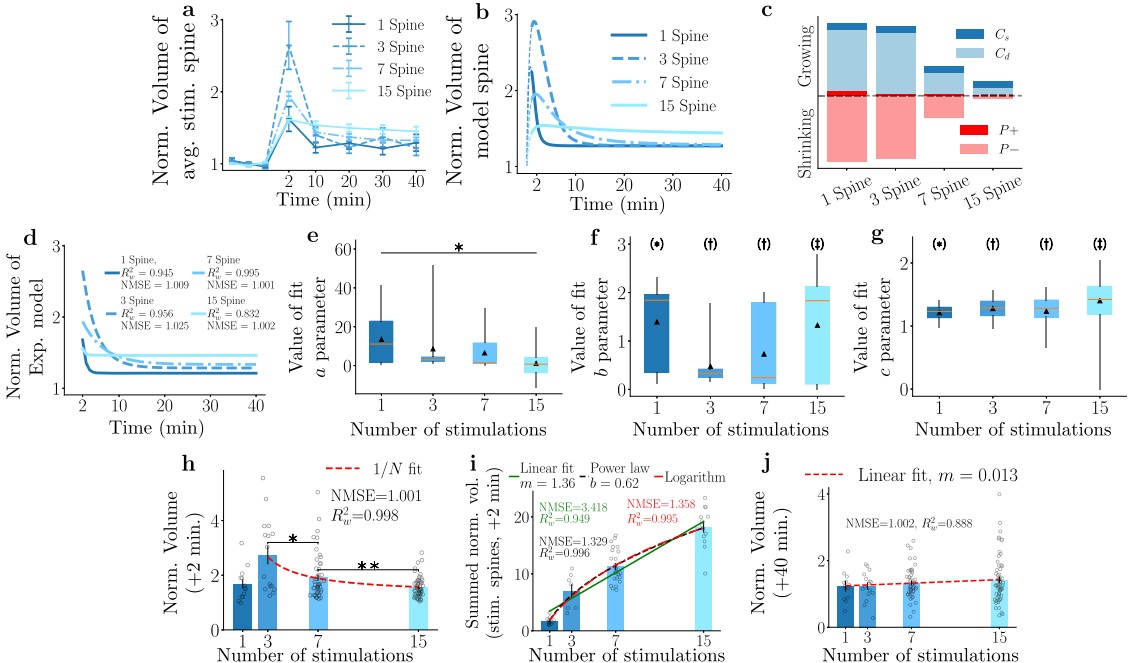

**Fig. 6 | Competition for resources alters plasticity outcomes as a function of stimulation sites. a** Experimentally measured spine sizes across experiments with 1-, 3-, 7- or 15-spine stimulations. Error bars represent s.e.m. **b** Model parameters obtained from data in Fig. 2 are used to predict the outcomes with varying number of stimulation sites. Thin dashed lines refer to the stimulation region (not modelled). **c** Comparing the dendritic and somatic $C$ and $P$ components across varying numbers of stimulation sites. Here $P_+$ and $P_-$ denote the potentiating and depressive factors. **d** Model performance vs an exponential $f(x) = a\exp(-bx) + c$ with varying parameters across experimental paradigms. **e**–**g** Statistical distributions for the exponential model parameters ($a$, $b$, and $c$ in **e**–**g**) determined by bootstrapping (see methods). * in (**e**) indicates that, (application of Kruskal–Wallis test and pairwise Dunn's test) each experiment differs significantly ($p < 0.0005$). In f and g, experiments have different symbols when statistically different ($p < 1e − 9$). Boxes represent the bootstrapped interquartile range; whiskers, the 95% confidence interval, black triangles, the mean of the data and orange lines, the median. **h** Average normalised spine response across experimental conditions at $t = 2$ min. $1/N$ relationship is observed between the number of stimulation events (experiments with >1 stimulated spine) and the initial potentiation. *$p < 0.005$ and **$p = 0.0038$. **i** Integrated spine volume increase (across stimulated spines relative to baseline) vs # of spines demonstrates an increase in spine size despite the increase of individual sLTP declines as a function of number of stimulated spines. Test fits (linear, power law, logarithms) demonstrate increasing spine growth. $N$=# of experiments (see Table 1). **j** Different experimental average normalised fluorescence intensities at $t = 40$ min with an overlaid linear fit between the number of stimulated spines and final average spine size. **h**–**j** error bars represent ±s.e.m. and $p$ value refers to the result of a two-sided t-test, accounting for multiple comparisons. **h**, **g**, $N$ numbers can be found in the corresponding figures above.

transient potentiation. Finally, spines in clustered 7-spine and 15 spine stimulations exhibit similar plasticity behaviours as the competition for $P$ and $C$ begins to dominate the dynamics with increasing number of stimulated spines.

Using the inner dynamics of the model, we can now attempt to explain the observed dynamics. For this purpose, we have dissected the total contribution of $C$ (both dendritic and synaptic) and the size-dependent potentiating $P_+$ or depressing $P_-$ contribution of $P$ to the spine dynamics in Fig. 6c. When considering stimulated spines, we propose that $P$ acts as a depressing agent because it serves the role of a size-dependent feedback mechanism which either provides additional growth when the spine is too small or restricts spine growth when the spine is already large. In particular, shortly after plasticity induction, the stimulated spines grow through the action of $C$, such that $P$ needs to provide the negative feedback mechanism to stop growth beyond a certain limit.

Additionally, the model can also provide quantitative comparisons across the different experimental paradigms. Spine dynamics could be explained by a rapid decrease of $C$ and $P$ as competition for resources increases. As more spines are stimulated, there is increasingly less $P$, which decreases the shrinkage that drives the spines back to the baseline size. This implies, on a network level, that only a sufficient amount of stimuli occurring within a certain distance and/or time (see the supplementary material for a discussion on how temporal dynamics can be implemented in the model) might allow for robust and sustained change that is associated with memory acquisition.

To gain insight into the temporal decay of the spines following the initial potentiation, we have attempted to fit an exponential decay model of the form $f(x) = a\exp(-bx) + c$ to each experimental dataset. While the simple exponential fits will not provide us with the mechanistic insight into the plasticity dynamics that our original model does, it can help quantify the decay differences via the changes in $b$.

Firstly, we fit the exponential model (fit parameters $a$, $b$, and $c$) to the mean dynamics of each experimental paradigm in Fig. 6d. In all cases, the NMSE is close to 1, which follows intuitively as this quantity is correlated with the value we minimise when performing the fitting procedure. We note that this simpler model also reproduces the general features of the dataset (e.g., order of efficacy with which spine size increase is observed at 2 min, decay rates of the initial transient potentiation, and the level of spine size increase at 40 min). We emphasise that these results are directly obtained from fitting the simple model and do not provide a prediction like the more complex model. We note that qualitative differences exist between the experimental data and the simple exponential model when comparing the 15-spine paradigm. In this setting, the stimulated spines do not show a decline in size after the initial potentiation (or do so very slowly), which leads to the insufficiency of the exponential model in describing this data (instantaneous decay to reach the saturation point, i.e. stable potentiation). In comparison, the more complex model can reproduce this slow decay, leading to better quantitative goodness-of-fit measures, possibly as a result of a more apt representation of the underlying mechanisms.

To statistically quantify the differences in the exponential decay model parameters, we performed a bootstrapping procedure, where the exponential was fit on a randomly selected subset within a stimulation paradigm, and this procedure was repeated (see the methods section for more details). This procedure led to a distribution of the model parameters that we could compare in Fig. 6e–g. Figure 6e shows the different fits of $a$ (or the scaling factor of the exponential model) with a larger $a$ implying a greater initial amplitude of the dynamics. Significant differences between the fits of this parameter for the stimulation paradigms reflect the ordering of the more complicated model we presented, albeit with the single-spine example having a larger $a$ than the 3-spine example. Next, in Fig. 6f, which depicts the degradation rate $b$, we see that the single spine decays significantly faster than the other 3 and 7 spine experiments, which agrees with our model dynamics in Fig. 6b. We note that the 15-spine example also exhibits high degradation rates; however, given the aforementioned incompatibility of the 15-spine dynamics with the exponential model, we attribute their large values to a relative difficulty fitting the dynamics to the exponential decay. Finally, we consider the saturation value $c$, which is the value that the exponential model tends towards as time advances and is shown in Fig. 6g. The 15-spine example has the highest value (significantly above the other experiments), while the single-spine experiment saturates at the lowest value. These observations agree with the experimental data, and the results of the more complicated model fits. We conclude, similar to the findings of the more complex model, that according to the exponential decay model, the 3 and 7 spine experiments fall within similar regimes, having similar decay rates and final saturation values, while the single and 15 spine experiments represent fundamentally different dynamics.

Finally, we directly compare the experimental dynamics themselves. In Fig. 6h, we plot the normalised spine sizes for the different experiments immediately after stimulation (i.e., at +2 min). We confirm that the single spine experiment exhibits significantly different behaviour than the other experiments. For experiments stimulating more than one spine, the results exhibit a $1/N$ relationship between the strength of the initial response to the stimulation and the number of spines stimulated. This is explained by the fact that short-time dynamics are dominated by $C$, for which each spine competes in a $1/N$ manner.

Nevertheless, despite this drop-off, the total spine growth across all stimulated spines of the cluster increases across the experimental paradigms (Fig. 6i). We find that the best fit (out of a linear function, power law, and logarithmic function) for these dynamics is a power law increase (followed closely by the logarithm), which indicates that as we add more spines for stimulations, the increase in the growth will slow down. This implies either that $C_s$ is sufficient to sustain potentiation across the cluster independent of how many spines are potentiated or production (e.g., protein translation) of $C$ and/or $P$, which is not taken into account in the model, is providing resources for this robust growth. We have briefly commented on this in the supplementary material.

In Fig. 6j, we consider the average fluorescence intensity of the spines at +40 min and note that these dynamics exhibit a linearly growing trend as more spines are stimulated. As more spines are potentiated, more $C_s$ is unlocked and rapidly becomes the primary source of $C$. Additionally, less is produced with more limited availability to the spines. This complicated interplay of initial competition for the dendritic component of $C$, generation of $P$, subsequent competition for $P$, and finally, the uncontested spine component of $C$ altogether serve to counteract the negative feedback loop that generates the depression back to the baseline. This effect overall is in agreement with the model predictions.

## Discussion

In this current study, we have precisely stimulated a varying number of spines sharing the same dendritic branch and generated a model to explain the allocation of molecular resources $P$ and $C$ as spines compete for these components that shape the plasticity response. We find that as the number of spines in a stimulated cluster increases, the initial magnitude of potentiation falls, but sLTP, the eventual potentiation observed over a longer term (40 min post-stimulation) increases. This can be explained within our model by the interaction between $P$ and $C$, and the availability of these two components along the dendritic stretch.

A handful of previous studies have used a similar experimental approach to study aspects of synaptic plasticity and spine-to-spine communication. However, to the best of our knowledge, this current study is the first to definitively explore the effect of cluster size and width on the plasticity outcome. Furthermore, many prior sLTP studies using glutamate uncaging have involved potentiating a single isolated spine on a dendrite, thereby omitting considerations for the interactions arising from cross-communication across synapses (for example[33]). Others have limited their approach to monitoring interactions between pairs of spines sharing the same dendrite (e.g., ref. [13]) to demonstrate the contribution of inter-spine interactions between spines that modulate plasticity.

A small group of studies to date have employed multiple spine uncaging on single dendritic segments, with a variety of outcomes. For example, in the study by ref. [21], eliciting LTP in a group of clustered spines resulted in synaptic depression of an unstimulated spine inside the cluster, which was dependent on calcineurin activity. We did not see synaptic depression inside stimulated clusters in our current study; however, this is likely due to differences in experimental design. Ref. [21] stimulated each LTP spine individually, one by one until the entire cluster had been stimulated (involving a minimum of 6 min by taking 1 min/spine to stimulate). On the other hand, our approach used quasi-simultaneous stimulation of all spines together (which involved stimulating the entire cluster within 60 sec) that likely engaged different endogenous dendritic mechanisms. In particular, single spine stimulation does not typically generate a large dendritic calcium transient, while simultaneous activity of nearby spines does. Work by[3] supported the idea that the amount of locally available $Ca^{2+}$ is controlled by clusters of simultaneously active synapses. It is worth highlighting that both simultaneous (this study) and spaced[21] stimulation patterns likely occur in vivo during behaviour and are both interesting to study and compare (e.g.[47]). Our model could explain the LTD behaviour encountered by[21] by noting that the individual $C$ peaks generated by the sequential stimulation protocol would not overlap at nearby heterosynaptic spines and instead diffuse away. Additionally, $P$, which is defined to exhibit much lower diffusion and degradation rates, would then accumulate at these non-stimulated sites and generate depression. Notably, our multi-spine stimulation paradigm could potentially promote glutamate spillover amongst spines of the targeted cluster and contribute to the spine plasticity outcome. Although the model parameters were fit using the 3-spine stimulation experiments where compound effects of spillover, if present, would have been included, future studies into the nature of $C$ and $P$ should take into consideration the effects of glutamate spillover.

Previous work by ref. [22] has demonstrated a mixture of synaptic potentiation and depression following the simultaneous potentiation of groups of spines on single dendritic branches. Again, the differences in stimulation parameters used may explain the differences in the measured outcome. Ref. [22] used whole-cell patch-clamp to depolarise the postsynaptic neuron to 0 mV for one minute in the presence of extracellular $Mg^{2+}$ to promote NMDAR activation whilst uncaging glutamate onto the homosynaptic spines to elicit structural plasticity. This likely initiated broader and more extensive signalling in the dendrite/neuron. In the current study, the experiments were, however, performed in $Mg^{2+}$-free aCSF to be consistent with other plasticity studies, such as the[21] study, without deliberate depolarisation of the postsynaptic cell membrane.

Changes in dendritic spine number and size in vivo have been shown to be tightly coupled to experience (for example, refs. 48, 49). Moreover, some studies have directly demonstrated activity-driven spine changes at synapses directly receiving active inputs and at neighbouring heterosynapses within single dendrites, for example, in hippocampal dentate granule cells ex vivo[50] and, following learning, in visual cortex in vivo[51], suggesting that homosynaptic and hetero-synaptic spine plasticity expressed at the level of dendritic branches is a common motif across the brain. Intriguingly, glutamate uncaging at single dendritic spines using more naturalistic spike patterns in brain slices in vitro appears to heighten the amount of heterosynaptic plasticity observed[52].

Neurons within a network are constantly subjected to a mixture of asynchronous intrinsic network activity, overlaid with synchronous bursts of activity arising from experience. Attempts to model this interaction (for example, ref. 53) suggest that background activity lowers the threshold for triggering dendritic spikes and action potentials. Interestingly, in a model of L5 cortical pyramidal neurons, without background activity the threshold for triggering a dendritic $Ca^{2+}$ spike required coincident stimulation of 15 synapses. With back-ground activity, the threshold fell to 6 synapses. These values closely match with the experimental range explored in this current study and suggest that the differences we have identified between the 7-spine and 15-spine dynamics could provide molecular insights into the basis by which asynchronous activity interacts with active inputs to shape dendritic integration. Further work is required to bridge the gap between knowledge gained from in vitro slice and modeling studies that have high experimental flexibility and the in vivo network activity that ultimately controls brain function.

It is unclear which protocol best represents the in vivo condition where, at least in some pyramidal neurons, it is highly likely that multiple inputs may be synchronously active on a single dendrite during behaviour and, in turn, engage the postsynaptic machinery in response to the local spatio-temporal synchrony. Regardless, our study attempted to identify the minimal components of the molecular signalling machinery that can represent the behaviour of active and inactive spines along stretches of dendrites.

With the introduction of the component P, our model aimed to retain the essence of the $Ca^{2+}$ amplitude hypothesis, which states that the peak $Ca^{2+}$ determines the direction of plasticity (either sLTP or sLTD)[35–37]. While our model did not directly include a $Ca^{2+}$ term, the model indirectly accounts for $Ca^{2+}$ changes through the C variable, which drives the P dynamics. Specifically, instead of $Ca^{2+}$, the spine size itself (inspired by refs. 33, 43) determines whether P promotes potentiation or depression in our model. Additional experimental studies motivating this hypothesis include the work by[54] showing that postsynaptic depolarisation level could affect the direction of plasti-city while the stimulation frequency was kept constant[40]. showed that directly lowering the extracellular $Ca^{2+}$ concentration turned LTP into LTD. In addition, using intracellular uncaging of $Ca^{2+}$[55], reported that a brief but high elevation of $Ca^{2+}$ resulted in LTP, whereas a longer but slight rise in $Ca^{2+}$ triggered LTD. Several models have been proposed to simulate this effect and manipulate the $Ca^{2+}$ amplitude and evaluate either frequency-based[54,56], STDP-based plasticity rules (for data, see ref. 57), or both[39,58,59]. Several important predictions have arisen from these models, including the supralinear relationship between stimu-lation frequency and calcium amplitude[39] (or ref. 60) whose model is able to fit the frequency-based plasticity curve and partly match a spike timing (STDP) based plasticity response.

Interestingly, previous studies have also demonstrated that rely-ing solely on calcium amplitude will always lead to discrepancies with experimental results. For example, ref. 61 showed that any such pure calcium-driven LTP model will always exhibit a second LTD window at long positive pre-post timing intervals. Nevertheless, several studies addressing this question did not find evidence of this phenomenon

and showed no LTD during positive pre-post intervals[62–64], while other studies did find this second LTD window[65–67]. The seemingly conflicting reports indicate that variability in the stimulation protocols, including variable timing and numbers of stimulated sites, as well as the region of interest, can lead to variable plasticity outcomes. Therefore, combin-ing the action of a slow molecular component, as we have introduced in our work, and clarifying its biological nature could help understand the mechanisms and identify the conditions that result in similar LTP/LTD outcomes. For example, our results indicate that small clusters of 3–5 spines could lead to a larger plasticity amplitude compared to scenarios where ten or more spines are clustered together or if spines are spatially further apart from each other. Future studies could clarify the precise number and cluster size along a dendrite to maximise plasticity outcomes.

Notably, in more recent models looking beyond the amplitude hypothesis[68], modelled cooperative plasticity across the dendritic tree and within single branches on a scale of milliseconds, and[69] used a finite pool of receptors to show a variety of heterosynaptic behaviours as a consequence of competitive effects.

In building our model, we implemented a fast molecular and a slower molecular component. One of the reasons we did not build directly on the models of refs. 38, 39, which also described plasticity across spines, was because these prior models focused on events over short timescales of seconds to tens of seconds. In our work, we aimed to capture spine plasticity effects evolving on the scale of minutes to tens of minutes (2–40 mins). Our model was thus moti-vated by prior models that aimed to cover relatively broad time-scales while simplifying the dynamics to two main actors. Examples of such models include the work of ref. 70, which used the concept of synaptic tagging to study early and late LTP, the study by ref. 71, which developed a three-layered model of synaptic consolidation, and the model by ref. 72, which considered discrete states to study the larger idea of how memory retention scales with the number of synapses.

In summary, we have introduced a model based on the action of fast $Ca^{2+}$ ions and their related molecules fuelling the activation of a long-lasting component P whose concentration ultimately drives the sLTP vs sLTD plasticity decision at the individual spines. Building on the previously introduced $Ca^{2+}$ levels hypothesis[35,38] and focusing on the timescale of minutes to tens of minutes (2–40 mins), we explored experimental outcomes across different spatiotemporal combinations of stimulation sites. Even though our model had only two dynamic variables and operated with an effective description of the LTP/LTD boundary, it could capture the experimentally measured plasticity and its time course across different numbers of stimulated spines. These insights will allow the model to be employed in spiking network simulations and give insight into the circuit-level consequences of the experimentally observed multi-spine plasticity rules.

## Methods
### Experimental
**Preparation of organotypic hippocampal slice culture.** Organotypic hippocampal slices were prepared as previously reported[73]. Briefly, hippocampi of postnatal day P6-7 Wistar rat pups (Nihon SLC) were isolated and cut into 350 μm-thick transverse slices on a McIlwain tis-sue chopper (Mickle Laboratory Engineering Co. and Cavey Laboratory Engineering Co.). Slices were transferred onto cell culture inserts (0.4 mm pore size, Merck Millipore) and placed in a 6-well cell culture plate filled with 1 ml/well of culture media containing 50% Minimum Essential Medium (MEM, Thermo Fisher Scientific), 23% Earle's Balanced Salt Solution, 25% horse serum (Thermo Fisher Scientific), and 36 mM D-glucose. Slices were maintained at 35 °C in 5% $CO_2$ and used for experiments at 14–18 days in vitro.

During experiments, slices were constantly perfused (1–2 ml/min) with aCSF containing (in mM) 125 NaCl, 2.5 KCl, 26 NaHCO₃, 1.25

$NaH_2PO_4$, 20 glucose, 2 $CaCl_2$, and 4 MNI-glutamate (Tocris). aCSF was continually bubbled with 95% $O_2$ and 5% $CO_2$, and experiments were carried out at room temperature. For a subset of experiments, calcineurin or CaMKII was inhibited with FK506 (2 μM, Tocris) or myristoylated Autocamtide-2-related inhibitory peptide (AIP, 5 μM, Calbiochem), respectively. All animal experiments were approved by the RIKEN Animal Experiments Committee and performed in accordance with the RIKEN rules and guidelines [Animal Experiment Plan Approval no. W2021-2-015(3)].

**Transfection and imaging of CA1 pyramidal neurons.** Organotypic slices were biolistically transfected using a Helios gene gun (Bio-Rad) and used for experiments 48–96 hours later. For structural plasticity experiments, gold particles were coated in a plasmid encoding EGFP. 50 μg of EGFP plasmid were coated onto 20–30 mg of 1.6 μm gold particles. Neurons were imaged at 910 nm on a Zeiss LSM 780 confocal laser scanning microscope, and all data were analysed offline.

**Dendritic spine imaging and glutamate photolysis.** Regions of dendrites were chosen by eye for imaging and stimulation. Regions were imaged for a brief baseline period by collecting z stacks of the dendritic arbour (512 × 512, 4× digital zoom for a final frame size of 33.7 μm). The z step was 0.5 μm. Glutamate was uncaged onto spines lying in the focal z plane using custom-written software at a distance of 0.5 μm from the spine head. Medium-sized spines with a clear spine head within the field of view were preferentially targeted for stimulation. MNI-glutamate was photolysed with a 2-photon laser source (720 nm), and each dendritic spine received a train of 60 pulses of laser light, each 4 msec long, repeated at 1 Hz. Uncaging laser power was set so that uncaging evoked excitatory postsynaptic currents (uEPSCs) matched endogenous spontaneous miniature EPSCs (mEPSCs), as measured by whole-cell patch-clamp. We did not see structural plasticity if we moved the uncaging laser away from the spine head (to 2 μm), changed the wavelength of the uncaging laser (720 nm to 880 nm), or uncaged in the presence of APV in the aCSF.

For sham experiments, MNI-glutamate was omitted from the aCSF. For groups of stimulated spines, laser pulses were delivered in a quasi-simultaneous fashion in sequence, in which the first spine received a pulse of glutamate (4 msec), which was followed by a short delay (<3 msec) as the system positioned the laser to the next spine in the sequence. This was repeated for all targeted spines in the stimulated cluster (3, 7, or 15), and the sequence was repeated at 1 Hz for 60 cycles.

**Numerical**

**Image analysis.** Estimated spine volumes were obtained from background-subtracted maximum-projected fluorescence images using the integrated pixel intensity (see refs. [74],[75]) of an octagonally shaped ROI surrounding the spine head. These values were normalised against the three observed data points immediately preceding the glutamate uncaging. The spine ROI was generated using a semi-automatic in-house Python package that took advantage of the structures of the spines (see the supplemental information for a full list and description) to generate a reproducible ROI. The manual interaction involves a simple clicking on the interior of the spines while the ROI and subsequent measurement are performed automatically. Temporal shifting was corrected using a phase cross-correlation algorithm implemented in SciPy[76]. This algorithm is designed and written in a user-friendly Python package available for download[77].

Spines that were obscured by a dendrite or other spines were omitted from the analysis. A significance test determined the success of an experiment in the form of a z-test, with <15% of experiments considered failures. All images shown and used for analysis are maximum-intensity projects of the 3D stacks.

**Statistics and reproducibility.** The fluorescence signals of spines obtained from our ROI detection algorithm were normalised against the mean of their pre-stimulation values as follows

$$\hat{J}_{i,j} = \frac{J_{i,j}}{\frac{1}{N}\sum_{j=1}^{N} J_{i,j}} \tag{10}$$

where $J_{i,j}$ is the fluorescence of spine $i$ at snapshot $j$ and $N$ is the number of pre-stimulation snapshots (in our experiment $N = 3$). Once the spines' fluorescences were normalised against their baseline, we pooled them across repetitions of the same experiment and cells. All statistics were calculated using this measure. Error bars represent ± s.e.m, and significance was set at $p = 0.05$ (studentised bootstrap). To compare different paradigms, Welch's unequal variances $t$ test was performed. To adjust for multiple comparisons, we employed the false discovery rate (FDR) method proposed by[78]. One asterisk indicates $p < 0.05$, while two represents $p < 0.005$. No statistical method was used to predetermine sample size. A significance test determined the success of an experiment in the form of a z-test, with less than 15% of experiments considered failures and the experiments were not randomised.

To provide measures for goodness-of-fit for the model predictions, we introduce two separate metrics that allow us to evaluate the quality of our model dynamics when compared against the experimental data. Firstly, we use the Normalised Mean Squared Error (NMSE). The NMSE is a normalised version of the Mean Squared Error (MSE), a common metric for evaluating regression models by quantifying the difference between a model's predicted values and the values observed from experiments. The normalisation of the MSE is done by dividing by the variance of the observed data and making the NMSE a relative measure rather than an absolute one. To extend this to temporal data, we calculate the NMSE at each available snapshot and then average across those snapshots to get a mean NMSE value across the full dynamics. Mathematically, we define this average NMSE as:

$$\text{NMSE} = \frac{1}{N}\sum_{j=1}^{N} \frac{\sum_{i=1}^{n}(S_{\text{pred},j} - S_{\text{expt},j,i})^2}{\sum_{i=1}^{n}(S_{\text{expt},j,i} - \bar{S}_{\text{expt},j})^2} \tag{11}$$

where $N$ is the number of snapshots, $n$ is the number of datapoints for snapshot $j$, $S_{\text{pred},j}$ is the prediction of the model for snapshot $j$, $S_{\text{expt},i,j}$ is the observed value of datapoint $i$ of snapshot $j$ and $\bar{S}_{\text{expt},j}$ is the average of the observed datapoints of snapshot $j$. We note that the minimum value is achieved when $S_{\text{pred},j} = \bar{S}_{\text{expt},j}$, which leads to NMSE = 1. Therefore, the closer the value is to 1, the better the fit to the mean of the data at each snapshot, i.e., the fit follows the mean dynamics. Additionally, we introduce a second measure: a variation of the commonly used $R^2$ metric, which measures the fitness of a linear fit. Here, we adapted this by using a weighted $R^2$ value ($R_w^2$), where we compare the model predictions against the mean of the data at each of the snapshots, weighted by the inverse of the variance at that point. This means that points with higher variance are less critical to the metric. Finally, we normalise this value against the total variation of the dataset, i.e., the mean of each observed snapshot against the mean of all data points. Mathematically, we describe this as:

$$R_w^2 = 1 - \frac{\sum_{i=1}^{N} w_i(S_{pred,i} - \bar{S}_{expt,i})^2}{\sum_{i=1}^{N} w_i(\bar{S}_{expt,i} - \bar{S}_{expt})^2} \tag{12}$$

$w_i$ is the inverse of the variance of the data at snapshot $i$, $\bar{S}_{expt}$ is the mean of all observed datapoints across all snapshots, and all other variables are as above. By incorporating variance as a weighting factor, this aims to account for the heteroscedasticity (varying spread) of the data across different timepoints. A higher value of $R_w^2$ (closer to 1) indicates a better fit, considering both the model's predictive accuracy

and the inherent variability of the data. More concretely, the $R^2_w$ measures how much better our model performs over a model, which has a constant horizontal line, which takes the mean of the experimental data as its height.

Using both metrics together offers a more holistic evaluation of our model. The NMSE provides a sense of the accuracy of the predictions regarding the data's variability. At the same time, the $R^2_w$ indicates how well the model explains the variance of the data, especially at key timepoints. While the NMSE might highlight the overall prediction accuracy, the $R^2_w$ can indicate whether the model captures the data's essential patterns. Thus, a model might have a low NMSE (suggesting good accuracy) but a low $R^2_w$ (indicating it's not explaining much of the variance), possibly hinting at overfitting. Conversely, a high $R^2_w$ but a high NMSE could suggest underfitting.

A bootstrapping approach was utilised to obtain the fits for the parameters of the exponential model. Half of the experiments were randomly selected for each spine stimulation paradigm (with replacement), and the exponential model was fit. This procedure was repeated 1000 times, as this value was found to lead to stable statistics for the $a$, $b$ and $c$ parameters. To test for significance, first, a Kruskal-Wallis test was performed, followed by Dunn's multiple comparison with the FDR correction factor.

**Model fitting algorithm.** The values of the model parameters $C_s$, $C_d$, $\lambda$, $\alpha_{1,2}$, $\beta_{1,2}$, $\gamma$, $\rho$, $\phi$, $\nu$, $\zeta_{1,2}$ of the model (see the equation in Fig. 1f) were obtained using a non-linear least-squares approach. The fitting routine included the introduction of a cost function that, when minimised, enforces agreement between the simulated model and observed data points and an iterative update scheme that finds parameters that best represent the experimental data. Mathematically, the cost function can be defined as

$$\sum_{t=0}^{T} (S - L_i)^2 \tag{13}$$

where $T$ is the length of the experiment (in our case, 40 minutes) and $L_i$ is the response of the stimulated spine at spatial point $i$. Points $i$ were scaled in the simulation to overlap with the average experimental values. The iterative scheme we chose to employ is a gradient-based adjoint approach (refs. 79, 80). In part, the adjoint method was chosen due to its ability to easily and efficiently handle multiple simultaneous optimisation parameters. A numerical solution supported the optimisation routine to provide feedback on the model dynamics given a set of parameters. The initial parameter fits were obtained with the values obtained from the three-spine paradigm. After that, the parameters were not altered and were solely used for predictive purposes. Only in the drug conditions (see Fig. 3), where $\zeta_{1,2}$ and the threshold $\nu$ altered to get insight into the possible model mechanisms. The sham experiments are trivially solved by setting the initially available $C$ and $P$ to 0. Given that our approach is gradient-based, and we are employing it in a non-linear setting, it is important to consider the impact of global vs. local solutions. To account for this, we initiated our parameter optimisation algorithms ten times with multiple randomly chosen starting points. In each case, the algorithm proposed similar solutions that generalised to the other test scenarios with different numbers of stimulated spines equivalent to the parameter set we used in our manuscript.

While we initially considered a spine-specific model, we followed the convention of the field and built an effective model that averages across individual spines and their variable signal-to-noise ratios.

**Reporting summary**
Further information on research design is available in the Nature Portfolio Reporting Summary linked to this article.

## Data availability
Experimental data sets included in the manuscript to generate the figures can be found in the following public github repository github.com/meggl23/MultiSpineModel with zenodo.org/doi/10.5281/zenodo.10057156[42]. Parts of this original data (primarily the sham data) were previously analysed in ref. 43 to gain insights into the spontaneous activity of the synaptic plasticity dynamics. These insights were subsequently used in the model generation of this publication. Source data are provided with this paper.

## Code availability
The Code to generate the figures of this work can be found in the following public github repository https://github.com/meggl23/MultiSpinemodel with zenodo.org/doi/10.5281/zenodo.10057156. The code used to analyse the calcium images can be found at github.com/meggl23/SpyDen and version 1.0-beta was used[77].

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

## Acknowledgements

This research was supported by the University of Bonn Medical Centre, the University of Mainz Medical Centre, the ReALity program at the Mainz Medical Centre, RIKEN Centre for Brain Science, OIST, JSPS Core-to-Core Programme (JPJSCCA20220007 to Y.G.), and by an add-on fellowship of the Joachim Herz Stiftung (M.E.). T.T. and Y.G. thank all our group members for fruitful discussions, and Janko Petkovic for feedback on an earlier version of the manuscript (T.T.). This project has received funding from the European Research Council (ERC) under the European Union's Horizon 2020 research and innovation program ("MolDynForSyn", grant agreement no. 945700 to T.T.). This work was supported by the Open Access Publication Fund of the University of Bonn.

## Author contributions

T.E.C., M.F.E., Y.G., and T.T. contributed to the design of the study; T.E.C. carried out imaging experiments; M.F.E. carried out the modeling; M.F.E. and T.E.C. performed data analysis; Y.G. and T.T. secured funding; all authors contributed to writing the manuscript. M.F.E. and T.E.C. are equally contributing first authors, and Y.G. and T.T. are co-corresponding senior authors who contributed equally.

## Funding

## Competing interests

The authors declare no competing interests.
