## [Peer Review File · Nature Communications]

REVIEWER COMMENTS

Reviewer #1 (Remarks to the Author):

Summary:

It is clear that the plasticity of a synapse can be strongly influenced by the activity of other nearby synapses. However, the patterns of neighborhood activity that drive potentiation or depression remain unclear. Previous studies (both in vitro and in vivo) have produced seemingly contradictory results. The authors develop a phenomenological model of multi-spine plasticity with only two dynamic components: a fast (few min) component and a slow (10s of minutes) component that is also has a strong dependence on spine size. The authors also conduct novel and challenging in vitro experiments (in Mg²⁺-free extracellular solution) stimulating from 1 to 15 spines and then tracking spine sizes over 40 min. The conceptual simplicity of a model with only two dynamic components – abstracted from any specific signaling pathways – is appealing. However, the model still contains a total of 10 parameters fit to the data. While the combination of experiment and modeling is commendable, it remains unclear from the presented analyses and statistics whether the data supports important predictions of the model (see detailed comments below). This is the most critical issue to address. Secondly, while the discussion of technical issues did very briefly touch on comparisons to in vivo, the implications of the results for our understanding of learning and memory could be further explored, particularly by devoting more discussion to relevant in vivo work and previous work that has simulated in vivo conditions. Such changes could make the study useful for a broader audience of neuroscientists.

Detailed Comments:

One major prediction of the model is that edge spines and middle spines will have different dynamics. However, the modeled effect size (Fig. 2f) appears very small compared to the error in the measurements. The authors interpret Fig. 2h as supporting a difference between edge and middle spines, however the analysis only shows that stimulated edge spines are significantly different from sham edge spines. The fact that middle spines are not significantly different from sham does not mean that they are different from edge, unless directly compared or a power analysis is conducted. Given the large SEM of stim and sham middle data relative to the edge spine effect size, it seems unlikely that there is sufficient power. Ns are not provided. The authors also interpret Fig. 2f as supporting a difference between edge and middle spines. Only one timepoint of the 5 measured timepoints is significantly different at $p < 0.05$. A correction for multiple comparisons or other analyses should be conducted to confirm that this result is not spurious.

Figures 2d, 3g-l, 4d, and 5d present model fits to the timecourse of spine volume changes under different conditions. No quantitative of goodness-of-fit measures are presented. Given the model has a total of 10 parameters, the significance of the qualitative similarity between the data and model fit is difficult to assess. Also, in the text comments are made about the dynamics without a summary measure of the dynamics, making it difficult to quantitatively compare conditions. It

looks like a simple exponential fit could provide time constants that could easily be compared and tested for significant differences.

In general, comparisons of this model against an alternative model would have been very helpful.

Figures 5g-i present some of the most interesting data in the manuscript. There are significant differences in volume at 2 min after 3, 7 or 15 spines are stimulated. However, the connection to the model is again qualitative. There are no goodness-of-fit measures. In Figure 5h, deviations from a linear fit look like they may be significant and in Figure 5i it is unclear if the slope is significantly greater than 0. Sample Ns are not provided.

The authors present inclusion of spine size as a critical parameter in the model as critical to the model, however this does not seem to be tested. For example, they could shuffle spine sizes to get idea of the effect of spine size on model goodness-of-fit.

Figure 1c-d: "Pool of spine-specific C", the illustration is confusing. It looks like a fused vesicle or broken barrier. Assuming this is meant to be a model abstraction, perhaps select a different manner to illustrate the nature of compartmentalization.

Figure 1e: Caption does not make clear if the data is derived from single-spine (1a) or multi-spine (1b) stimulation or some combination of both.

On page 5: "The motivation of Cs comes from the spine calcium transients seen in response to synaptic activity. To numerically simulate this sudden and local inflow of C into the stimulated spines...". Distinctions between Ca^{2+} concentration, baseline distributions of Ca^{2+} -sensitive proteins, and distributions of activated/released Ca^{2+} proteins seem critical to maintain throughout the manuscript to prevent confusion. Here, for example, "inflow" sounds like Ca^{2+} ions but I assume just meant "increase"...? Since model abstractions are intermingled with experimental data, it would be helpful if the authors were more explicit about these distinctions throughout the manuscript, even at the expense of repetition.

On page 6, the annotation with respect to time, initial states, and active vs. inactive is a bit counter-intuitive. Why use a zero subscript for the inactive protein (i.e. "P0"), which usually denotes initial state? In Figure 5f, P+ and P- show up without definition.

To place the results in a broader context, perhaps further discuss the relationship of results to key in vivo functional work (e.g., El-Boustani et al., which is only briefly mentioned in the supplement) or modeling of in vivo input patterns (e.g., Farinella et al., Plos Comp. Biol 2014).

Reviewer #2 (Remarks to the Author):

The manuscript by Chater et al. attempted to clarify the underlying mechanisms of spine structural plasticity elicited by simultaneous stimulation of multiple spines along dendritic segments. The quantitative time-lapse imaging of spine structural dynamics before and after plasticity induction by glutamate uncaging was combined with the mathematical modeling of temporal and spatial propagation of signaling molecules, which induce bidirectional changes in spine structure. The proposed model can explain differential responses of spines stimulated individually or as clusters with different numbers or relative distances, indicating the usefulness of the model based on the idea of competition among spines for the resources of signaling molecules with slow temporal kinetics.

Previous studies reported a variety of spine structural plasticity in response to multiple spine stimulation. This variation, either spine shrinkage or expansion in response to clustered stimulation of nearby spines, should be explained by a model integrating both spatiotemporal dynamics of signaling events and subsequent regulation of spine shape. As a first step toward this goal, the manuscript by Chater et al. is highly valuable for providing the quantitative model that explains the magnitude and temporal pattern of spine responses. This reviewer positively evaluates this study linking the mathematical modeling and the response of the spine population along dendritic segments. However, several points should be corrected and improved in a future manuscript.

Major points.

1. The mathematical model contains two components, C and P, which have different mobility in the cytoplasm. If the model is consistent with the spatiotemporal pattern of signal propagations, the mobility of C and P should be comparable to the experimentally measured molecular diffusion within dendrites. This point should be confirmed.
2. In the Abstract, the authors wrote “our model can reconcile disparate experimental reports of sLTP and sLTD at stimulated and non-stimulated spines.” However, the model presented in this study does not explain the spine volume change reaching a value less than the initial volume, namely sLTD. The model contains the component of $F(S)$, which can reduce the spine volume, but this implementation of the negative regulation component (a component of depotentiation) is not directly related to the manifestation of sLTD, which requires a net decrease of spine volume after stimulation.
3. The authors should clarify the experimental data that supports the title of Figure 2, “different spine dynamics for spines on the edge or middle of the clusters.” The differences in volume change between the middle and edge spines shown in Figure 2g and 2h are not large. Are these differences statistically significant? Judging from the experimental and modeling data shown in Figure 2e and f, the largest difference may exist in the slope of volume decay from the peak ($t = 2$ min) to $t = 10$ min, probably reflecting the difference in the amount of available C. Therefore, the critical evaluation of the experimental data should be the difference in the extent of spine volume decay in this temporal window. This point should be evaluated.

4. The distances between the three spines shown in Figure 2a-c are not similar, but the two spines are in close vicinity. If two spines are closer, the effect on edge spines on this side may be weak. It is desirable to incorporate this point into the analysis.

5. Information about the optimization of parameters should be provided. For example, the ratio between C_s and C_d may influence the initial spine enlargement, but how critical is this value in mathematical modeling? Similarly, does the larger parameter ϕ , which removes the transition phase, disrupt the pattern of a slow decrease in spine volume after stimulation? These points may help understand the contribution of each parameter in the temporal pattern of spine responses.

Minor points,

1. In the Results section, the authors wrote “At its core, the model depends on two distinct proteins: C, which acts on faster timescales”. It is confusing where C is postulated as a protein or a small molecule such as calcium.

2. The decay curve of C in Figure 2i is difficult to evaluate. Individual curves should be presented.

3. In the text related to Figure 3g, the authors wrote “single spines rapidly grow (as there is no competition for C) but then equally rapidly return to the baseline.” But Figure 3g shows the normalized spine volume at $t = 40$ min is about 1.3, not returning to the baseline.

4. The spine volume data is normalized against the three observed data points immediately preceding the glutamate uncaging. This strategy is adequate for comparing single spine responses, but the data shown in Figure 5g-i calculate the total spine responses, which may require incorporating differences in the initial spine volume before stimulation.

5. In the legend of Figure 3 a-c), the words “bottom row sham” should be removed. This point should be discussed.

6. In the legend of Figure 4 a-c), “Example image of the seven-spine experiment for min before” should be “ for 5 min before”.

Reviewer #3 (Remarks to the Author):

The authors explore how plasticity of dendritic spines depends on the number and spatial proximity of other nearby active synapses, focusing on the impact of local competition for resources. They use a computational model incorporating putative plasticity-regulating proteins with various spatial and temporal dynamics to predict differences in magnitude and time course of structural LTP as a function of cluster dimensions and the position of the spines in the cluster. They compare the

model predictions with experimental measurements of spine head size after inducing LTP by glutamate uncaging in pyramidal neurons in organotypic culture.

The activity-dependent factors that shape the strength of individual synapses in the local dendritic environment are far from being completely elucidated, and more experimental and theoretical work is clearly needed to better understand local synaptic plasticity rules. However, in my opinion the conclusions of the current paper have not been convincingly supported by the results provided. I have several concerns regarding the analysis and interpretation of the data and the limitations of the applied approach.

Major comments:

1) The authors make several claims that the experimental results confirm the model's predictions, which are not sufficiently supported by the presented data and analysis.

- For the 3-spine cluster (Fig 2), the model predicts that the increase of spine head volume is larger in the first 10 minutes in spines at the cluster edge than those in the middle, followed by a decline to similar levels by 20 minutes (Fig 2f). However, the data (Fig 2e, g, h) do not demonstrate a significant difference between the middle and edge spines stimulated in the same clusters in the same experiments. The only difference is reported when the 'edge' spine group is compared to independent sham experiments at 40 minutes, however this cannot substitute for the lack of significant effect in the relevant direct comparison of middle and edge spines. Overall I do not see that the measured results would agree with the model's prediction.

- The authors say that they demonstrated that sLTP of single stimulated spines decays faster than that of clustered potentiated spines. However, no statistical analysis is provided, and no evident difference in kinetics can be seen when comparing the data in Fig 2d and Fig 3d. In fact, the differences predicted by the model should be observed within the first 10 minutes, but such short-term changes cannot be evaluated with the low temporal resolution of the measurements.

- In 7-spine clusters (Fig 4), the authors claim that spines at the edge of the cluster potentiate more than spines in the middle. My understanding is that this is based on a single data point being different at 20 minutes, whereas no differences are indicated at other time points (including 2 and 40 minutes, which are analysed in the rest of the study).

The authors also say in the text that the difference between middle and edge spines is larger in 7-spine clusters than in 3-spine clusters, but this is not confirmed by quantitative analysis.

- In the 7-spines experiment, the authors say that they predict and find differences in plasticity when the synapses are more distributed spatially than in the control clustered condition. However,

no statistical comparison was made between the control and distributed 7-spine conditions. The only analysis presented is the lack of a significant difference between distributed 7 spines and single spine data, which does not directly address the original question.

- In several cases paired t-tests were performed across multiple data groups (for example Fig 3e-f, Fig 5g). Appropriate statistical tests or adjustment need to be applied to correct for the multiple comparisons.

- The numbers of experiments should be indicated throughout the paper, and exact p values reported for all statistics.

2) The experiments were done in 0 mM Mg²⁺ and in most cases multiple spines were stimulated quasi-simultaneously. It is reasonable to expect that such stimulation produces strong summated dendritic depolarisation, activates voltage dependent Ca²⁺ and other channels and possibly release from intracellular stores, and overall induces large dendritic Ca²⁺ accumulation. This biological state differs from that in the model, where synapses are individual plasticity points that do not interact other than through biochemical mechanisms. Have the authors measured the electrical and Ca²⁺ responses to the LTP stimulation, and considered to take into account the impact of excess dendritic Ca²⁺?

3) The model predicts mostly modest differences in sLTP (for example, ~10-15% difference for middle and edge spines), which requires that the experimental method is able to reliably reveal small effects. However, glutamate uncaging is difficult to standardise to such extent across spines and experiments (location of the uncaging point around and away from the spine, laser power at different depths), and it is unclear from the manuscript what attempts have been made to do so.

Another important point is that the relative volume change from baseline during sLTP depends on the initial spine head size (Matsuzaki et al 2004 Nature, and others). Therefore the only way to appropriately determine the effects of cluster size or spine position on sLTP strength would be to select spines with very similar initial head sizes, which does not seem to have been the case here.

4) The configuration of the experimentally stimulated spines does not correspond well to that used in the model.

- Fig 2a: two of the three stimulated spines emanate from the same dendritic location. Which of these spines would be 'middle' and 'edge'?

- In Fig 5 the stimulated spines are unevenly distributed forming two large groups of several small clusters of very close spines. In this scenario, can the spines be considered as a single cluster of spines competing for the same resources?

5) The image in Fig 2b shows severe swelling and distortion of the stimulated spines and the dendrite at 2 min after uncaging, which seems out of the normal. Other than the independent sham experiments, can the authors exclude that their stimulation conditions caused unphysiological side effects affecting spine size in some experiments?

6) The model seems to be quite complicated to the details, which stands in contrast with the lack of clearly plausible biological counterparts with properties similar to those of the proteins designed in the model. One cannot help but wonder that other combinations of mechanisms and features, not based on competition for resources, might be just as suitable to explain the experimental observations.

Minor comment:

7) Fig 3a-c caption says to show top and bottom rows with control and sham experiments, but only one row is displayed.

Reviewer #1 (Remarks to the Author):

Summary:

It is clear that the plasticity of a synapse can be strongly influenced by the activity of other nearby synapses. However, the patterns of neighborhood activity that drive potentiation or depression remain unclear. Previous studies (both in vitro and in vivo) have produced seemingly contradictory results. The authors develop a phenomenological model of multi-spine plasticity with only two dynamic components: a fast (few min) component and a slow (10s of minutes) component that is also has a strong dependence on spine size. The authors also conduct novel and challenging in vitro experiments (in Mg²⁺-free extracellular solution) stimulating from 1 to 15 spines and then tracking spine sizes over 40 min. The conceptual simplicity of a model with only two dynamic components – abstracted from any specific signaling pathways – is appealing.

However, the model still contains a total of 10 parameters fit to the data. While the combination of experiment and modeling is commendable, it remains unclear from the presented analyses and statistics whether the data supports important predictions of the model (see detailed comments below). This is the most critical issue to address.

Secondarily, while the discussion of technical issues did very briefly touch on comparisons to in vivo, the implications of the results for our understanding of learning and memory could be further explored, particularly by devoting more discussion to relevant in vivo work and previous work that has simulated in vivo conditions. Such changes could make the study useful for a broader audience of neuroscientists.

We thank the reviewer for the appreciation of our work and for highlighting the appeal of the simplicity of our model. We are grateful for the many helpful comments and suggestions, which we have incorporated in our revised manuscript and addressed point-by-point below.

Detailed Comments:

One major prediction of the model is that edge spines and middle spines will have different dynamics. However, the modeled effect size (Fig. 2f) appears very small compared to the error in the measurements. The authors interpret Fig. 2h as supporting a difference between edge and middle spines, however the analysis only shows that stimulated edge spines are significantly different from sham edge spines. The fact that middle spines are not significantly different from sham does not mean that they are different from edge, unless directly compared or a power analysis is conducted. Given the large SEM of stim and sham middle data relative to the edge spine effect size, it seems unlikely that there is sufficient power. Ns are not provided. The authors also interpret Fig. 2f as supporting a difference between edge and middle spines. Only one time point of the 5 measured timepoints is significantly different at $p < 0.05$. A correction for multiple comparisons or other analyses should be conducted to confirm that this result is not spurious.

We thank the reviewer for mentioning this key point and for suggesting to provide more clarity about when significant differences can be expected in the data and when the model indicates no difference between edge and middle spines. We followed the reviewer's suggestion and now provide additional analysis to quantify the effect size of our model prediction and check whether our experimental data for the 3-spine stimulation protocol is in line with our theory.

In the previous version of the manuscript, we generated model predictions using the average size of the clusters across each experimental paradigm. Then, we used this distance of average clusters to generate an equidistant set of spines corresponding to the average spine-to-spine distance in our data. While this assumption leads to a reasonable prediction of the average stimulated spine dynamics, upon re-analysis, we discovered that it was not precise enough to accurately describe the dynamics of the edge and middle spines individually. To understand how the different cluster sizes and spine-to-spine distances may affect the dynamics, we generated the set of spine-to-spine distances and then calculated the predicted model dynamics for the minimal and the maximal experimentally observed spine-to-spine distance. We now display the model prediction for the mean spine-to-spine distance as a bold line and the model predictions for the minimal and maximal distances as a shaded area of the same colour (green for edge and blue for middle). This provided us with a model-based way of quantifying the variability in our spine dynamics. The new corresponding model figure is included below; the inset shows the predicted dynamics between 0-10 min at an expanded timescale.

Caption: Using the maximal and the minimal spine-to-spine distances in our experimental data allows us to delineate the expected variability in the dynamics of the edge (green) and middle (blue) spines as well as their average predicted behaviour. The bold line represents the respective mean response. The model predicted overlap in the edge and middle spine dynamics indicates that no significant differences between these two spine types can be expected in this experimental scenario. Inset represents the boxed section of the plot at an expanded timescale.

We note that the overlap in the model variability predicted for the edge and middle spines as calculated for the spine-to-spine distance statistics implies that little statistical differences can be expected from these two groups for this particular 3-spine stimulation paradigm.

Next, we performed the same re-analysis for our 7-spine experiments and predicted the range of plasticity responses for the middle and edge spines using the maximal and minimal spine-to-spine distance within this experiment. Here, our model predicts significant overlap in the early stages of the plasticity dynamics (2 and 10 min) and starting from 20 min the model indicates that the plasticity response for the edge spines would be above the responses of the middle spines. This is in line with the experimental data. We found that the response curve for the edge spines (green) lies above that of the middle spines (blue), and these responses differ significantly at +20 min (see figure below).

Caption: When considering edge spines (green) and middle spines (blue), our model predicts a significant difference, given our experimental spine-to-spine distance, for timepoints larger than 10 min. The experimental data reveals significantly different dynamics at $t = 20$ min between the edge (light grey) and middle (dark grey) spines. $*p = 0.03$, two-sided t -test. Error bars represent the standard error of the mean.

We extended the same re-analysis for our 15-spine experiments and predicted the range of plasticity responses for the middle (7 spines) and edge spines (4 spines flanking middle spines) using the maximal and minimal spine-to-spine distance within this stimulation scenario. See Figure below.

*Caption: When distinguishing between 8 edge (green) and 7 middle spines (blue) and considering the variability in their pairwise distances, our model predicts increasing differences for later time points. This difference between the edge (light grey) and middle (dark grey) spines is statistically significant for $t=30$ min and $t=40$ min. $**p=0.005$ and $*p = 0.04$, two-sided t -test. Error bars represent the standard error of the mean.*

Therefore, supported by the model, we hypothesise that the competition in the three-spine scenario is not sufficient to cause significant differences between spine types, but as more spines are stimulated, a growing difference between the edge and middle spines is observed, in which the response of the edge spines is above that of the middle spines.

To summarise, we are grateful for this comment and have now added the explanation above to the description of the 3-spine stimulation experiment in our manuscript as follows:

“To understand how the different cluster sizes and the distances between spines may affect the plasticity dynamics, we studied the statistics of the spine-to-spine distances in our 3-spine stimulation paradigm and calculated the predicted model dynamics for the minimal ($2.2 \mu\text{m}$) and the maximal ($4.2 \mu\text{m}$) experimentally observed cluster size. The model prediction for the mean cluster size ($3.2 \mu\text{m}$) is shown as a bold line, and the space between the model predictions for the minimal and maximal distances is represented by a shaded area of the same colour (green for edge and blue for middle) in Fig. 2e. This provides a model-based way of quantifying the variability in our spine dynamics. We note that the model dynamics we obtain from our 3-spine stimulations exhibits a substantial overlap between the middle and edge spines. Given this overlap, the model predicts that edge and middle spines would exhibit a similar plasticity response. This matches our experimental data as shown in Fig 2e. Moreover, when compared to sham-stimulated spines, while both types of spines grew significantly at 2 min, only the edge spines at 40 min were significantly enlarged (Fig. 2g and Fig. 2h, respectively).”

Given these results, we hypothesise that when 3 spines are stimulated the spine-spine competition for the available resources is not sufficient to cause significant differences between edge and middle spines. “

To describe the 7 and 15 spine stimulation experiments we have added the following text:

“Following the steps we used to model the 3-spine stimulation experiment in Fig. 2, we studied the plasticity response of the edge and middle spines in the experiment where 7 spines were simultaneously stimulated. The distribution of minimal spine-to-spine distance in the experiment with 7 stimulated spines is shown in Fig. 4g. Fig. 4f displays the plasticity response of the three inner spines (middle spines in blue) alongside the plasticity response of the four outermost spines (edge spines in green). To compare model predictions with recorded data, we first calculated the response dynamics given the largest ($13.1 \mu\text{m}$) and smallest ($6 \mu\text{m}$) cluster size, with the central line defining the average cluster size ($\approx 10.6 \mu\text{m}$). We find significant overlap in the early stages of the predicted dynamics (2 and 10 min). At later time-points, the edge and middle spines become more distinct, which suggests that there may be significant differences in the dynamics between edge and middle spines of the experimental data. As predicted by the model, we find that edge spines are significantly larger at +20 min than the middle spines (see Fig. 4f). Interestingly, this difference was larger than that observed previously in the three-spine experiment (cf. Fig. 2e-f). This implies that a high number of stimulated spines, which compete for resources, not only has global effects but also impacts local resources creating an abundance of P and a lack of C which are more pronounced at certain spines compared to others (in line with the first model prediction).”

[...]

“Finally, our goal was to investigate whether increasing the competition further by stimulating more spines (15 simultaneously stimulated spines) would change the plasticity dynamics as predicted by our model. To this end, we generated a set of model

predictions using the same model parameters as above and stimulating 15 spines in total whose smallest, mean, and largest cluster sizes were 20 μm , 28.3 μm and 30.6 μm , respectively. Next, we compared the predicted dynamics (Fig. 5f) with the experimentally observed results. We note that the model has good NMSE values in both edge and middle cases, but the R2w value shows that the experimental variability is missed by the model (Reply comment: please see below for an explanation of NMSE and R2w values). Additionally, we found that the edge spines (4 outermost spines on each side) responded stronger than the middle spines (7 inner spines flanked by 4 edge spines on either side) at +30 and +40 min post stimulation, in line with the model predictions. These observations are consistent with C and P dynamics of the model (Fig. 5g-i). As defined by the initial condition, the edge spines start with a higher C (because they have slightly more neighbouring resources; see grey horizontal dashed lines). C then diffuses away rapidly, making them comparable to the spines in the middle of the cluster after 2 min. In contrast, the behaviour of the protein variable P is slightly different because it evolves on longer time scales (see Fig. 5h). As more C is initially available at the edge spines (due to less competition), more P is subsequently generated at those edge spines. However, for time points past 2 min, P in the middle spines accumulates due to the diffusion of P from those spines on either side of the middle spine that are closer to the edge (in addition to the local activation of P) and is coupled with the lower degradation of P. This pooling of P amongst middle spines affects their longer time spine plasticity dynamics, which leads to lower potentiation and more rapid depression back to the baseline compared plasticity of the edge spines (see Fig. 5f). Therefore, supported by the model, we hypothesise that the increased competition in the 15-spine paradigm leads to larger differences in the plasticity responses within the stimulated cluster compared to plasticity responses in the three-spine and seven-spine cases."

Finally, we have added p and N values to all relevant figures.

Figures 2d, 3g-l, 4d, and 5d present model fits to the timecourse of spine volume changes under different conditions. No quantitative of goodness-of-fit measures are presented. Given the model has a total of 10 parameters, the significance of the qualitative similarity between the data and model fit is difficult to assess.

Also, in the text comments are made about the dynamics without a summary measure of the dynamics, making it difficult to quantitatively compare conditions. It looks like a simple exponential fit could provide time constants that could easily be compared and tested for significant differences.

In general, comparisons of this model against an alternative model would have been very helpful.

We agree with the reviewer that each of the fits (both those in Figure 5 - previous version, now Figure 6 - and the model fits/predictions) would benefit from quantitative goodness-of-fit measures to complement the qualitative plots. Given that our fits are often non-linear and/or consist of temporal dynamics, we introduced two measures to describe the quality of the fits and allow for a quantitative comparison between experimental paradigms.

Our first fit quality measure is the Normalised Mean Squared Error (NMSE). The NMSE is a normalised version of the Mean Squared Error (MSE), a common metric for evaluating regression models by quantifying the difference between a model's predicted values and the values observed from experiments. The normalisation of the MSE is done by dividing by the variance of the observed data and making the NMSE a relative measure rather than an absolute one. To extend this to temporal data, we calculate the NMSE at each available snapshot and then average across those snapshots to get a mean NMSE value across the full dynamics. Mathematically, we define this average NMSE as:

$$\text{NMSE} = \frac{1}{N} \sum_{j=1}^N \frac{\sum_{i=1}^n (S_{\text{pred},j} - S_{\text{expt},j,i})^2}{\sum_{i=1}^n (S_{\text{expt},j,i} - \bar{S}_{\text{expt},j})^2}$$

where N is the number of snapshots, n is the number of data points for snapshot j, $S_{\text{pred},j}$ is the prediction of the model for the j snapshot, $S_{\text{expt},j,i}$ is the observed value of the i data point of snapshot j and $\bar{S}_{\text{expt},j}$ is the average of the observed data points of snapshot j. The numerator represents the standard MSE and the denominator represents the normalisation by the variance at each point. We note that the minimum value is achieved when $S_{\text{pred},j} = \bar{S}_{\text{expt},j}$, which leads to NMSE = 1. Therefore, the closer the value is to 1, the better the fit to the mean of the data at each snapshot, i.e., the fit follows the mean dynamics.

Additionally, we introduce a second fit quality measure: a variation of the commonly used R² metric, which measures the fitness of a linear fit. Here, we adapted this by using a weighted R² value (R_w²), where we compare the model predictions against the mean of the data at each of the snapshots, weighted by the inverse of the variance at that point. This means that points with higher variance are less critical to the metric. Finally, we normalize this value against the total variation of the dataset, i.e., the mean of each observed snapshot against the mean of all data points. Mathematically, we describe this as:

$$R_w^2 = 1 - \frac{\sum_{i=1}^N w_i (S_{\text{pred},i} - \bar{S}_{\text{expt},i})^2}{\sum_{i=1}^N w_i (\bar{S}_{\text{expt},i} - \bar{S}_{\text{expt}})^2}$$

By incorporating variance as a weighting factor, this aims to account for the heteroscedasticity (varying spread) of the data across different time points. A higher value of R_w² (closer to 1) indicates a better fit, considering both the model's predictive accuracy and the inherent variability of the data. More concretely, the R_w² measures how much better our model performs over a model with a constant horizontal line which takes the mean of the experimental data as its height.

Using both metrics together offers a more holistic evaluation of our model. The NMSE provides a sense of the accuracy of the predictions regarding the data's variability. At the same time, the R_w^2 indicates how well the model explains the variance of the data, especially at key time points. While the NMSE might highlight the overall prediction accuracy, the R_w^2 can indicate whether the model captures the data's essential patterns. Thus, a model might have a low NMSE (suggesting good accuracy) but a low R_w^2 (indicating it is not capturing all variance of the data), possibly indicating overfitting. Conversely, a high R_w^2 but a high NMSE could indicate underfitting.

By adding these metrics to each of the figures that contain predictive model fits, we can now conduct comparative analysis between different settings, enabling us to gain insight into the model's utility. Example analyses, which we have added to the updated manuscript, are as follows:

“This is further supported by both the normalised mean square error (NMSE) (the average of the mean square error of each time point divided by that time point's variance) and weighted R2 (R_w^2) (a variation of the standard R2 to study the goodness-of-fit for a non-linear predictor) being close to unity (more details on these quantitative fit measures can be found in the methods section, under statistical definitions).”

[...]
“For the plasticity responses with inhibited CaMKII activity, we considered as a first step the constant solution corresponding to $C=0$ and $P=0$ that visually provided an acceptable approximation of the experimental data (Fig. 3i, dashed light green line). However, the relatively high NMSE value of 1.143 and negative $R2w$ suggested that this model approximation was not able to capture the mean dynamics nor the up and down trends contained in the data. As a next step, we decreased the ratio of ζ_1 and ζ_2 (which increases the effect of P) in the model and lowered the P threshold in eq. (8) and were now able to achieve a closer approximation of the experimental dynamics (NMSE = 1.049, $R2w = 0.549$) than the previous, constant model. This hints at the possibility that blocking CaMKII not only removes the potentiating component of P but could also increase its depressing effect.

[...]
“Nonetheless, we note that the model dynamics predicted for the single spines (fully decoupled) achieves lower goodness-of-fit values (both for NMSE and R_w^2) in Fig. 4i in comparison with the prediction of the distributed 7 spine model.”

We have added the definitions and discussions of NMSE and R_w metrics to the statistical definitions section of our updated manuscript.

Finally, we agree with the reviewer that a summary of the dynamics and comparative analysis would be valuable, and we took this advice to heart. To this end, we have split the original Figure 5 into two separate figures (now Figures 5 and 6) and include these comparative metrics in the relevant captions and text sections. The new Figure 5 in the section “Spine plasticity responses are altered by a reduction in shared resources”, now focuses solely on the 15 spine dynamics and includes an additional set of figures that describe the difference between the edge and middle spines in this setting. Overall, the manuscript now includes the content we show below for convenience:

Figure 5: Simultaneous induction of sLTP at 15 spines leads to a reduced plasticity response in the first 10 min which is well predicted by our model. a)-c) Example images showing 15 stimulated spines at 5 min before, 2 min after and 40 min after inducing sLTP. The stimulated spines are marked by white arrow heads. d) Normalised synaptic changes of the average stimulated spines are shown (grey points). By reducing the amount of initially available C and P according to equations (1) and (5), the model reproduces the dynamics of the experimental results. Detailed analysis of the data in the grey boxes marked with (†, ‡) is shown in e). e) Normalised growth of the stimulated spines at $t = +2$ and $t = +40$ min compared against a 15-spine sham experiment. $ p < 0.0005$**

and $*p = 0.0007$, two-sided t -test. $N = 98$ for the middle and 94 edge spines for the control and $N = 49$ middle and 49 edge spines for the sham. f) When distinguishing between 8 edge (green) and 7 middle spines (blue) and considering the variability in their pairwise distances, our model predicts increasing differences for later time points. This difference between the edge (light grey) and middle (dark grey) spines is statistically significant for $t = 30$ min and $t = 40$ min. $**p = 0.005$ and $*p = 0.04$, two-sided t -test. Error bars represent the standard error of the mean. g) Difference in the amount of C at the edge vs middle spines reveals that the edge gets initially more C (less competition) and thus potentiates more strongly. h) Temporal dynamics of P for the edge (darker) and middle spines (lighter). We note an increased pooling of P at the middle spines due to slower dynamics. This additional P also leads to a more rapid decrease immediately after stimulation

Figure 6, which can be found in the new section “Competition among stimulated spines alters plasticity response dynamics”, contains an overview of all stimulation conditions, the corresponding experimentally measured synaptic response dynamics, our predicted model dynamics and a comparison between our model and an exponential fit approach. This new figure is depicted below.

Figure 6: As more stimulated spines compete for resources, plasticity responses are altered the most in the first 10 min after stimulation. a) Comparative summary of experimentally measured synaptic plasticity responses across stimulation paradigms with 1, 3, 7 or 15 spines. b) Using model parameters obtained from data in Fig. 2 allows for predictions of the temporal dynamics in experiments with varying numbers of stimulated spines. c) Comparison of the contribution of the dendritic and somatic C and P components across the experiments with a varying number of stimulations. Here P+ and P- refer to the amount of potentiating and depressive actors, respectively, in the spine. d) Performance of our model vs a simple exponential $f(x) = a \exp(-bx) + c$ with varying parameters across experimental paradigms along with NMSE and R^2_w values. e) A change in the exponent b of the exponential model across simulation scenarios indicates a non-linear dependence of the temporal plasticity scales. f) Comparison of the average normalised fluorescence intensity of spines across the different experimental conditions at $t = 2$ min. For the experiments stimulating more than one spine, a $1/N$ relationship is observed between the number of stimulation events and the initial potentiation. $*p = 0.006$ and $p < 0.005$ for 3 spine vs 7 spine and 7 spine vs 15 spine, respectively. N is the number of experiments per paradigm and is summarised in table 1. g) Total synaptic growth (integrated across all stimulated spines of an experiment relative to baseline) vs number of stimulated spines demonstrates a cooperative increase in synaptic strength even though the extent increase in the size of individual sLTP spines declines as a function of number of stimulated spines. Test fits (linear, power law, logarithms) have been performed to ascertain the nature of the increasing spine growth. h) Comparison of the average normalised fluorescence intensity across the different experimental conditions at $t = 40$ min. A linear relationship is observed between the number of stimulations and the final average spine size. In all figures above, error bars represent the standard error of the mean.

The first two panels of new Figure 6 are plots of the 1,3,7, and 15 spine stimulation experiments illustrating the experimental data (Fig 6a) and the model predictions (Fig 6b). We are grateful to the reviewer for suggesting the inclusion of summary plots, because it provides an overview of our experimental findings and allows for a direct comparison with the model predictions. We find, for example, that the relative order of the size of normalised spine volume increases immediately post-stimulation are well represented by the model (albeit the model prediction of the single spine paradigm has higher growth immediately preceding 2 minutes due to lack of competition) and that the final saturation is well captured by the model as well. We have added additional discussion to the text in the new section as follows:

“Having demonstrated the predictive power of the model for plasticity protocols with a growing number of spines, we now consider a comparison of the different experiments. To this end, we summarised the experimental data and model dynamics for the average

response of the stimulated spines in Fig. 6a) and b), respectively. We note that spines in the 3-spine paradigm potentiate transiently far more strongly than the other examples, subsequently declining to stabilise at above baseline levels. On the other hand, in the single-spine paradigm, which does not involve any competition for resources, the stimulated spine potentiates strongly but also displays the fastest decay of the transient potentiation. Finally, the seven and 15 spine examples exhibit similar behaviours to each other as the competition for P and C begins to dominate the dynamics. The model dynamics not only provide a framework for interpreting the underlying basis of experimental data, additionally, we can gain insights into the dynamics at time points we did not monitor experimentally. For example, the model highlights a single-spine behaviour that is largely similar to the three-spine experiment (a fast-growing component before the 2-minute mark) reaching a peak within 2 min and exhibiting a rapid decline. Despite this rapid decline, at the 2-minute mark (black dashed line), the model correctly predicts the order of relative spine volume change within the experimental data (i.e., 15 spine, 1 spine, 7 spine, and 3 spine experiments). Additionally, the model captures the final saturation rate of each experiment, underscoring the idea that the number of spines stimulated fundamentally affects the increase in the size of the spines reached over the long-term.”

Following the reviewer’s comment that the exponential model (of the form: $f(x) = a \cdot \exp(-b \cdot x + c)$) provides insights that we could use for further analysis, we now include this information in a figure. Firstly, in Fig 6d), we fit the exponential model (fitted parameters are a, b, and c) to the mean dynamics of each experimental paradigm. In all cases, the NMSE is close to 1, which follows intuitively as this quantity is correlated with the value we minimise when performing the fitting procedure. We note that this simpler model also reproduces the general features of the dataset (e.g., the extent of transient potentiation at 2 min, decay rates of transient potentiation, and final saturation value – the extent of long-term potentiation, i.e. sLTP – at 40 min). Additionally, the exponential decay describes the 1, 3, and 7 spine experiments well, albeit with different exponents. However, these results are obtained by directly fitting the exponential form to the data, contrary to the predictive capabilities of the more complex model. Moreover, qualitative differences exist between the experimental data and the simple exponential model when comparing the 15-spine paradigm. In this setting, there is very little decay following the initial potentiation (the dynamics are almost linear) and over the observed time, such that the exponential model nearly instantaneously decays to the saturation point and remains there. In comparison, our more complex model can reproduce the slow decay of the initial increase in spine size observed experimentally, leading to better quantitative goodness-of-fit measures and providing a possible explanation of the underlying mechanisms in terms of the dynamics of P and C values.

The exponential model can provide valuable insights into some aspects that the more complicated model cannot. Notably, as suggested by the reviewer, the decay rate (parameter b) is a parameter that we can compare between experiments to determine quantitative differences. To this end, we took the individual spines of each experiment and generated the exponential fit for each of these. We omitted those spines where the resulting decay rate was larger than 10, as the sparse temporal resolution coupled with the noisy biological nature of the spines led to some spines having erroneous fit results. This then led to a set of decay parameters we could compare in Fig 6e. We note differences between the seven spine and other experiments, where the spines in the seven spine experiment show a significantly slower decay from the initial transient potentiation than the other two multi-spine stimulation paradigms. We would initially assume that the 15-spine experiment would have the slowest decay. However, given the semi-linear nature of the dynamics, the fits are not so easily obtained in this scenario, a problem that our model, which predicted rather than was fit to the dynamics, did not face. For our model, the predictive power remained valid across different stimulation paradigms, showing that the internal model parameters obtained in one stimulation paradigm generalise to varying numbers of stimulated spines. For example, we found that the smaller peak potentiation and slower plasticity decay observed in the 15-spine experiment could be explained by the increased competition among the spines, leading to the lower amount of P per stimulated spine. In our model, different amounts of P give rise to the different speeds at which the spines potentiate and then depress after stimulation.

We have added this discussion to the updated manuscript as follows:

“To gain insight into the temporal decay of the spines following the initial potentiation, we have attempted to fit an exponential decay model of the form $f(x) = a \exp(-bx + c)$ to each experimental data set. While the simple exponential fits will not provide us with the mechanistic insight into the plasticity dynamics that our original model does, it can help quantify the decay differences via the changes in b.

Firstly, we fit the exponential model (fit parameters a, b, and c) to the mean dynamics of each experimental paradigm in Fig. 6d. In all cases, the NMSE is close to 1, which follows intuitively as this quantity is correlated with the value we minimise when performing the fitting procedure. We note that this simpler model also reproduces the general features of the dataset (e.g., order of efficacy with which spine size increase is observed at 2 min, decay rates of the initial transient potentiation, and final level of spine size increase at 40 min). We emphasise that these results are directly obtained from fitting the simple model and do not provide a prediction like the more complex model. We note that qualitative differences exist between the experimental data and the simple exponential model when comparing the 15-spine paradigm. In this setting, the stimulated spines do not show a decline in size after initial potentiation (or do so very slowly), which leads to the insufficiency of the exponential model in describing this data (instantaneous decay to reach the saturation point, i.e. stable potentiation). In comparison, the more complex model can reproduce this slow decay, leading to better quantitative goodness-of-fit measures, possibly as a result of more aptly representing the underlying mechanisms.

To address the differences in decay rates of the initial potentiation, we took the individual spines of each experiment and generated the exponential fit to each of these. We omitted those spines where the resulting decay rate was larger than 10, as the sparse

temporal resolution coupled with the noisy biological nature of the spines led to some spines having erroneous fit results. This procedure led to a set of decay parameters we could compare in Fig. 6e. We note differences between the seven spine and other experiments, where the seven spine experiment decays significantly slower from the initial transient potentiation than the other two paradigms. We initially assumed that the 15-spine experiment would have the slowest decay. However, given the semi-linear nature of the dynamics, the fits are not easily obtained in this scenario (as seen in the previous fit of the average dynamics). For our model, the predictive power remained valid across different stimulation paradigms showing that the internal model parameters obtained in one stimulation paradigm generalise to varying numbers of stimulated spines. For example, we found that the smaller peak potentiation and slower plasticity decay observed in the 15-spine experiment could be explained by the increased competition among the spines, leading to the lower amount of P . In our model, the different amounts of P were the reason behind the different speeds at which the spines were depressing after stimulation.”

Figures 5g-i present some of the most interesting data in the manuscript. There are significant differences in volume at 2 min after 3, 7 or 15 spines are stimulated. However, the connection to the model is again qualitative. There are no goodness-of-fit measures. In Figure 5h, deviations from a linear fit look like they may be significant and in Figure 5i it is unclear if the slope is significantly greater than 0. Sample Ns are not provided.

We are pleased that the reviewer found the panels presented in Fig. 5g-i interesting, and we agree that including quantitative fit measures would strengthen the data. To this end, we added the R^2_w and NMSE calculations (described above) to each figure to determine whether the fitted functions suitably represent the experimental data. The alterations can be seen in the figure below.

The $1/N$ fit of the normalised volume at +2 min agrees with the data (although this may be intuitive as we fit the function directly to the data). Here, we note that the R_w^2 metric is perhaps not the most apt, as the inherent variability of the data means that we may overestimate the quality of our fit. Therefore, we concentrate instead on the NMSE metric, which measures the average difference between the model and data across time points.

For panel g, we use this metric to determine the best-fit model. The reviewer correctly mentioned that a linear fit may not be the best approximation to our data. Indeed, when comparing a linear fit, a power law, and a logarithmic fit, the logarithmic function best describes the relationship between total growth and number of stimulations. Thus, as more and more spines are stimulated, their total growth increases, but more slowly, which falls in line with our model predictions regarding increasing competition. Nonetheless, as more spines are stimulated, we expect the results to tend towards a linear fit as they progressively rely on their internal C_s stores to grow.

We have added a sentence to reflect this discussion as follows:

“We find that the best fit (out of a linear function, power law and logarithmic function) for these dynamics is a logarithmic increase, which indicates that as we add more spines for stimulations, the increase in the growth will slow down.”

As suggested by the reviewer we now added the slope of the linear fit in Fig 6h ($m=0.018$). The value is far less than one, implying a very slow increase in the normalised volume as more spines are stimulated.

f) Comparison of the average normalised fluorescence intensity of spines across the different experimental conditions at $t = 2$ min. For the experiments stimulating more than one spine, a $1/N$ relationship is observed between the number of stimulation events and the initial potentiation. * $p = 0.006$ and ** $p < 0.005$. N is the number of experiments per paradigm and is summarised in table 1. g) Total synaptic growth (integrated across all stimulated spines of an experiment relative to baseline) vs number of stimulated spines demonstrates a cooperative increase in synaptic strength even though the extent increase in the size of individual sLTP spines declines as a function of number of stimulated spines. Test fits (linear, power law, logarithms) have been performed to ascertain the nature of the increasing spine growth. h) Comparison of the average normalised fluorescence intensity across the different experimental conditions at $t = 40$ min. A linear relationship is observed between the number of stimulations and the final average spine size. In all figures above, error bars represent the standard error of the mean.

Additionally, we have added a sentence to the figure caption that points the reader to the number of experiments performed:

"N is the number of experiments per paradigm and is summarised in table 1."

The authors present inclusion of spine size as a critical parameter in the model as critical to the model, however this does not seem to be tested. For example, they could shuffle spine sizes to get idea of the effect of spine size on model goodness-of-fit.

We thank the reviewer for their insightful comments and for giving us the opportunity to clarify the role of spine size in our model. We appreciate the suggestion to test initial spine size as a critical parameter for the plasticity response. In our model, spine size is indeed an important aspect, but perhaps not in the conventional sense of a direct variable. Instead, we incorporated the size effect as a threshold (see Fig. 1e below) to indicate that smaller spines (red regime in Fig 1e) tend to potentiate and bigger spines (blue area in Fig 1e) tend to depress. This concept stems from observations in our own data and is based on previous research by several labs, (for example the Ziv lab and Nedivi lab) that smaller spines tend to grow and larger spines to shrink. The particular position of the threshold is based on the area of the spines (μm^2), rather than their normalised luminosity values.

Caption: The size of the spine is negatively correlated with its subsequent size evolution within our model, i.e., small spines tend to grow and large spines tend to shrink. ν represents the point when the changes flip from growth to shrinkage. Data adapted from the 15 spine stimulation paradigm analysed in Fig. 5.

This observation served as an inspiration for the implementation of the $F(S)$ -function in Eq. 8, guiding our understanding of spine dynamics and providing a feedback mechanism that prevents spines from growing too large. It is important to clarify that when we consider the experimentally recorded spine sizes we normalise them against their baseline values before we fit the model parameters. In effect, we start our model parameter S from one and consider in the model the behaviour of the average spine post stimulation. This threshold was then subsequently fit to the 3-spine experimental data. Nonetheless, to clarify the role of size in our model, we have adapted Fig. 1f to emphasis the normalised version of spine size:

f) Based on the results in e), we model the effect of P (potentiating or depressing) using the activation function F and initial spine size (normalised by the pre-stimulation baseline). This figure represents a possible example threshold ν . When the normalised spine size is below the threshold ν (here for illustration purposes $\nu = 1.4$), P acts in a potentiating manner ($F(S) > 0$ in the red region), encouraging growth. However, when $S > \nu$, $F(S) < 0$ (blue region), the action of P is to depress, thereby providing a potential mechanism to avoid uncontrolled growth.

Since all spine sizes are standardised to a baseline value of 1, shuffling would not alter the relative changes we are analysing and thus would not impact the model's performance.

During the experiments, we attempted to select spines of similar sizes (medium sizes spines). Below we show the distribution of spine sizes for each of the control experiments:

We note that the 1, 3 and 7 spines all have similar distributions and do not differ significantly. However, the 15 spine experiments do exhibit a wider range of sizes and also a slightly higher mean size. This is a natural result of the design of the experiment: a need to select 15 relatively clustered spines considerably limits the freedom in choosing the spine size as we require a dendritic branch carrying 15 spines showing a confined spatial distribution and that are also in the same z plane of the microscope.

Despite the limitations, we note that the percentage of spines in the 7 and 15 spine experiments whose sizes fall in the range of the three spine experiment (i.e., the number of spines that are bigger than the smallest spine in the 3 spine experiment and smaller than the largest spine in the 3 spine experiment) is 86.7% percent and 88.3% percent, respectively.

Figure 1c-d: "Pool of spine-specific C", the illustration is confusing. It looks like a fused vesicle or broken barrier. Assuming this is meant to be a model abstraction, perhaps select a different manner to illustrate the nature of compartmentalization.

We thank the reviewer for this helpful comment. We have removed the lines that looked like vesicles/broken barriers and instead added brackets that denote the synaptic and dendritic stores. The new figure is shown below:

Before sLTP induction, a pool of C (light blue circles) and inactivated P (grey circles) is available at spines (top: red bracket) and dendrite (bottom: dark blue bracket).

Figure 1e: Caption does not make clear if the data is derived from single-spine (1a) or multi-spine (1b) stimulation or some combination of both.

We are grateful to the reviewer for suggesting to add this information, which we now did in the caption to Fig. 1e:

"Data adapted from the 15 spine stimulation paradigm analysed in Fig. 5"

On page 5: "The motivation of Cs comes from the spine calcium transients seen in response to synaptic activity. To numerically simulate this sudden and local inflow of C into the stimulated spines...". Distinctions between Ca²⁺ concentration, baseline distributions of Ca²⁺-sensitive proteins, and distributions of activated/released Ca²⁺ proteins seem critical to maintain throughout the manuscript to prevent confusion. Here, for example, "inflow" sounds like Ca²⁺ ions but I assume just meant "increase"...? Since model abstractions are intermingled with experimental data, it would be helpful if the authors were more explicit about these distinctions throughout the manuscript, even at the expense of repetition.

We agree with the reviewer that a more explicit treatment of C would add clarity. To this end, we have made the change as suggested (i.e. “inflow” to “increase”). We have also ensured that a consistent description of Ca²⁺ changes is used throughout the text.

On page 6, the annotation with respect to time, initial states, and active vs. inactive is a bit counter-intuitive. Why use a zero subscript for the inactive protein (i.e. “P₀”), which usually denotes initial state? In Figure 5f, P₊ and P₋ show up without definition.

Thank you for this suggestion. To clarify our definition, we have changed P₀ to P_{in}, which we believe better reflects the inactive nature. We have made this change throughout the updated manuscript.

Additionally, the reviewer is correct in pointing out that P₊ and P₋ are used without prior definition. We have amended this oversight by adding the following sentence to Fig 6c legend:

“Here P₊ and P₋ refer to the amount of potentiating and depressive actors, respectively, in the spine.”

To place the results in a broader context, perhaps further discuss the relationship of results to key in vivo functional work (e.g., El-Boustani et al., which is only briefly mentioned in the supplement) or modeling of in vivo input patterns (e.g., Farinella et al., Plos Comp. Biol 2014).

We thank the reviewer for highlighting these important studies. To better link our results to the wider literature we have added to the discussion the following sentences:

“Changes in dendritic spine number and size in vivo have been shown to be tightly coupled to experience (for example, Barnes et al., 2017; Frank et al., 2018). Moreover, some studies have directly demonstrated activity-driven spine changes at synapses directly receiving active inputs and at neighbouring heterosynapses within single dendrites in hippocampal dentate granule cells ex vivo (Jungenitz et al., 2018) and, following learning, in visual cortex in vivo (El-Boustani et al., 2018), suggesting that homosynaptic and heterosynaptic spine plasticity expressed at the level of dendritic branches is a common motif across the brain. Intriguingly, glutamate uncaging at single dendritic spines using more naturalistic spike patterns in brain slices in vitro appears to heighten the amount of heterosynaptic plasticity observed (Argunsah and Israely, 2023).

Neurons within a network are constantly subjected to a mixture of asynchronous intrinsic network activity, overlaid with synchronous bursts of activity arising from experience. Attempts to model this interaction (for example Farinella et al., 2014) suggest that background activity lowers the threshold for triggering dendritic spikes and action potentials. Interestingly, in a model of L5 cortical pyramidal neurons, without background activity the threshold for triggering a dendritic Ca²⁺ spike required coincident stimulation of 15 synapses. With background activity the threshold fell to 6 synapses. These values closely match with the experimental range explored in this current study, and suggests that the differences we have identified between the 7-spine and 15-spine dynamics could provide novel molecular insights into the basis by which asynchronous activity interacts with active inputs to shape dendritic integration. Further work is required to bridge the gap between knowledge gained from in vitro slice and modelling studies that have high experimental flexibility, and the in vivo network activity that ultimately controls brain function.”

Additional references

Argunsah AÖ, Israely I. Homosynaptic plasticity induction causes heterosynaptic changes at the unstimulated neighbors in an induction pattern and location-specific manner. *Front Cell Neurosci.* 2023 Sep 27;17:1253446. doi: 10.3389/fncel.2023.1253446. PMID: 37829671; PMCID: PMC10564986.

Barnes SJ, Franzoni E, Jacobsen RI, Erdelyi F, Szabo G, Clopath C, Keller GB, Keck T. Deprivation-Induced Homeostatic Spine Scaling In Vivo Is Localized to Dendritic Branches that Have Undergone Recent Spine Loss. *Neuron.* 2017 Nov 15;96(4):871-882.e5. doi: 10.1016/j.neuron.2017.09.052. Epub 2017 Nov 5. PMID: 29107520; PMCID: PMC5697914.

Farinella M, Ruedt DT, Gleeson P, Lanore F, Silver RA. Glutamate-bound NMDARs arising from in vivo-like network activity extend spatio-temporal integration in a L5 cortical pyramidal cell model. *PLoS Comput Biol.* 2014 Apr 24;10(4):e1003590. doi: 10.1371/journal.pcbi.1003590. PMID: 24763087; PMCID: PMC3998913.

Frank AC, Huang S, Zhou M, Gdalyahu A, Kastellakis G, Silva TK, Lu E, Wen X, Poirazi P, Trachtenberg JT, Silva AJ. Hotspots of dendritic spine turnover facilitate clustered spine addition and learning and memory. *Nat Commun.* 2018 Jan 29;9(1):422. doi: 10.1038/s41467-017-02751-2. PMID: 29379017; PMCID: PMC5789055.

Jungenitz T, Beining M, Radic T, Deller T, Cuntz H, Jedlicka P, Schwarzacher SW. Structural homo- and heterosynaptic plasticity in mature and adult newborn rat hippocampal granule cells. *Proc Natl Acad Sci U S A.* 2018 May 15;115(20):E4670-E4679. doi: 10.1073/pnas.1801889115. Epub 2018 Apr 30. PMID: 29712871; PMCID: PMC5960324.

Reviewer #2 (Remarks to the Author):

The manuscript by Chater et al. attempted to clarify the underlying mechanisms of spine structural plasticity elicited by simultaneous stimulation of multiple spines along dendritic segments. The quantitative time-lapse imaging of spine structural dynamics before and after plasticity induction by glutamate uncaging was combined with the mathematical modeling of temporal and spatial propagation of signaling molecules, which induce bidirectional changes in spine structure. The proposed model can explain differential responses of spines stimulated individually or as clusters with different numbers or relative distances, indicating the usefulness of the model based on the idea of competition among spines for the resources of signaling molecules with slow temporal kinetics.

Previous studies reported a variety of spine structural plasticity in response to multiple spine stimulation. This variation, either spine shrinkage or expansion in response to clustered stimulation of nearby spines, should be explained by a model integrating both spatiotemporal dynamics of signaling events and subsequent regulation of spine shape. As a first step toward this goal, the manuscript by Chater et al. is highly valuable for providing the quantitative model that explains the magnitude and temporal pattern of spine responses. This reviewer positively evaluates this study linking the mathematical modeling and the response of the spine population along dendritic segments. However, several points should be corrected and improved in a future manuscript.

Major points.

1. The mathematical model contains two components, C and P, which have different mobility in the cytoplasm. If the model is consistent with the spatiotemporal pattern of signal propagations, the mobility of C and P should be comparable to the experimentally measured molecular diffusion within dendrites. This point should be confirmed.

We thank the reviewer for the insightful feedback and the opportunity to enhance our manuscript by establishing a more concrete connection between the model components, C and P, and their biological counterparts. Their suggestion to align the mobility of these components with the experimentally measured molecular diffusion within dendrites is indeed pivotal for the consistency and validity of our model.

Regarding component C, which is inspired by the highly local and fast moving dynamics of calcium and calcium-binding proteins, our model posits that its behaviour is primarily governed by the degradation term rather than the diffusion term. This approach is in concordance with the experimentally observed characteristics of calcium signalling, particularly the highly localised calcium transients. In our experimental observations, the response of C is intensely localised around the stimulated location, mirroring the spatial specificity seen in calcium-mediated processes. To illustrate this principle, we have added two additional panels to supplemental figure 1, which show the spatial distribution of C and P at timepoints up to 2 min. We see that the amount of C is highly localised around the stimulated spines and degrades rapidly while P diffuses and does not degrade as fast (which is consistent with our experimental data we now show in the supplement):

Caption: Given the 3 spine model simulation, we can gain insight into the spatial dynamics of the C and P components in (g) and (h), respectively. We note that the C dynamics are highly localised and dominated by the degradation term, while P primarily diffuses, thus being found at locations further from the stimulation site as well as accumulating around the stimulation sites.

Many previous reports have imaged calcium dynamics in spines and dendrites in response to various stimuli, both in vitro and in vivo (for example, Losonczy & Magee, Scholl et al., 2021, reviewed in Higley & Sabatini 2012), and have shown that following synaptic activity spine calcium quickly returns to baseline within seconds, matching our modelling predictions. However, to make sure that the localised nature of the C component aligns well with experimental data on Ca²⁺ dynamics in the specific stimulation paradigms we use, we decided to measure the spatial and temporal footprint of Ca²⁺ in the dendrites and spines during the uncaging stimulus, (see below, supplementary figure 2 in the revised manuscript). This figure illustrates the calcium transients for the specific stimulation paradigms we use in our manuscript.

Figure S2: Imaging of GCaMP6s in spines and dendrites during uncaging. Dendrites from neurons expressing GCaMP6s and DsRed were targeted for simultaneous imaging and glutamate uncaging. a) Example images illustrating spatial spread of calcium during uncaging. Green, GCaMP6s, magenta, DsRed. Scale bar = 5 μm . b) Spine and c) dendritic calcium dynamics during uncaging. Repetitive glutamate uncaging (yellow bar) elicits a sustained spine and dendritic calcium elevation during stimulation, which returns to baseline levels within seconds of the cessation of uncaging. Measuring calcium elevations along the dendritic shaft demonstrates that calcium rises are spatially confined close to targeted spines, and that peak calcium occurs in the centre of the stimulated spine cluster. d) Measured average spatial profile of calcium during uncaging stimulus. All individual experiments were aligned to the stimulus spine (for the 1x spine condition) or to the centre of the stimulated cluster (for the 3x, 7x, 15x, and the distributed 7x conditions). N's (cells); 1x = 8, 3x = 9, 7x = 11, 15x = 5, 7x distributed = 7.

To better highlight this aspect in the text, we have added the following sentence to the updated manuscript:

“The mechanics of C and P, as a function of space and the corresponding effects on heterosynaptic spines in and outside clusters of stimulated spines is shown in Supplemental Figure 1. We note that C is highly localised in comparison to P and degrades at a much faster rate which is in line with our experimentally recorded Ca^{2+} dynamics (supplemental Fig. S2).”

Now, let us comment on the spatial footprint of the component P in our model. In our model, P has a longer action range and a relatively slow diffusion rate compared to the degradation rate of C. This characteristic enables P to exert a measurable effect at more distant locations within the dendrite, a feature that is pivotal for its role in the model. A critical question to ask in this context is whether the diffusion coefficient of P is compatible with experimental reports of many dendritic proteins. To this end, we have included an additional supplemental figure in our manuscript, which can be seen below. It illustrates that the diffusion coefficient we obtained from our model fit (black dot and its arrow bar) falls within the experimentally reported range of dendritic and synaptic proteins, including those linked to synaptic plasticity (e.g. CamKII):

Figure S3: The diffusion coefficient we obtained in our model fit of the 3-spine stimulation experiment (black dot with error bars) is consistent with many experimentally observed diffusion coefficients of possible biological candidates for the component *P*. Abbreviations rec. and SU stand for receptor and sub-unit, respectively. ¹(Thillaiappan et al., 2017), ²Datapoints refer to a slow, medium-fast and fast CamKII α population as well as CamKII δ populations from Lu et al. (2014), ³(Hannezo et al., 2015), ⁴(Muir and Kittler, 2014), ⁵(Hausrat et al., 2015), ⁶(Mikasova et al., 2012), ⁷(Renner et al., 2012), ⁸(Pantazaka and Taylor, 2011), ⁹(Neupert et al., 2015), ¹⁰(Boiko et al., 2007), ¹¹(Peng et al., 1989), ¹²(Murakoshi et al., 2004).

We have added the following to the discussion on possible biological candidates for C and P respectively:

“Therefore, C can be interpreted to be a fast protein, a small molecular compound that reacts quickly to spine stimulation, or potentially a combination of both. Since our model simulations of C in supplemental Figure S1g and the highly localised nature of Ca²⁺ in our experiments in supplemental Figure S2 are compatible with each other, C can also represent Ca²⁺ directly.”
[...]

“In supplemental figure 3, we show that the diffusion coefficients of the possible biological correlates of P are consistent with the range of diffusion coefficients we obtained for P. Additionally, it is conceivable that our model variable P represents the combined action of multiple proteins. In this case a mix of the different CamKII proteins could also be a potential biological correlate of P.”

In summary, we believe that our model, through these distinct characteristics of C and P, can be a good representation for the spatiotemporal signal integration as observed experimentally in multiple spines.

References

- Higley, M. J., & Sabatini, B. L. (2012). Calcium signaling in dendritic spines. *Cold Spring Harbor perspectives in biology*, 4(4), a005686. <https://doi.org/10.1101/cshperspect.a005686>
- Losonczy, A., & Magee, J. C. (2006). Integrative properties of radial oblique dendrites in hippocampal CA1 pyramidal neurons. *Neuron*, 50(2), 291–307. <https://doi.org/10.1016/j.neuron.2006.03.016>
- Scholl B, Thomas CI, Ryan MA, Kamasawa N, Fitzpatrick D. Cortical response selectivity derives from strength in numbers of synapses. *Nature*. 2021 Feb;590(7844):111-114. doi: 10.1038/s41586-020-03044-3. Epub 2020 Dec 16. Erratum in: *Nature*. 2021 Feb 1;; PMID: 33328635; PMCID: PMC7872059.

2. In the Abstract, the authors wrote “our model can reconcile disparate experimental reports of sLTP and sLTD at stimulated and non-stimulated spines.” However, the model presented in this study does not explain the spine volume change reaching a value less than the initial volume, namely sLTD. The model contains the component of F(S), which can reduce the spine volume, but this implementation of the negative regulation component (a component of depotentiation) is not directly related to the manifestation of sLTD, which requires a net decrease of spine volume after stimulation.

We thank the reviewer for raising this point, as they are correct in noting that we do not directly induce or measure sLTD. To this end, we have removed the sentence from the abstract: “Moreover, our model can reconcile disparate experimental reports of sLTP and sLTD at stimulated (homosynaptic) and non-stimulated (heterosynaptic) spines.” and replaced it with “our model can reconcile seemingly conflicting plasticity reports obtained across varying numbers and positions of stimulated spines”. We also took care to differentiate between sLTD as an outcome and the depressive action of a molecular agent inside the spines.

3. The authors should clarify the experimental data that supports the title of Figure 2, “different spine dynamics for spines on the edge or middle of the clusters.” The differences in volume change between the middle and edge spines shown in Figure 2g and 2h are not large. Are these differences statistically significant? Judging from the experimental and modeling data shown in Figure 2e and f, the largest difference may exist in the slope of volume decay from the peak ($t = 2$ min) to $t = 10$ min, probably reflecting the difference in the amount of available C. Therefore, the critical evaluation of the experimental data should be the difference in the extent of spine volume decay in this temporal window. This point should be evaluated.

We thank the reviewer for suggesting to clarify this key point to increase the clarity about when edge and middle spines exhibit different dynamics. In the previous version of the manuscript, we generated model predictions using the average size of the clusters across each experimental paradigm. Then, we used this distance of average clusters to generate an equidistant set of spines corresponding to the average spine-to-spine distance in our data. While this assumption leads to a reasonable prediction of the average stimulated spine dynamics, upon re-analysis, we discovered that it was not precise enough to accurately describe the dynamics of the edge and middle spines individually. To understand how the different cluster sizes and spine-to-spine distances may affect the dynamics, we generated the set of spine-to-spine distances and then calculated the predicted model dynamics for the minimal and the maximal experimentally observed spine-to-spine distance. We now display the model prediction for the mean spine-to-spine distance as a bold line and the model predictions for the minimal and maximal distances as a shaded area of the same colour (green for edge and blue for middle). This provided us with a model-based way of quantifying the variability in our spine dynamics. The new corresponding model figure is included below; the inset shows the predicted dynamics between 0-10 min at expanded timescale.

Caption: Using the maximal and the minimal spine-to-spine distances in our experimental data allows us to delineate the expected variability in the dynamics of the edge (green) and middle (blue) spines as well as their average predicted behaviour. The bold line represents the respective mean response. The model predicted overlap in the edge and middle spine dynamics indicates that no significant differences between these two spine types can be expected in this experimental scenario. Inset represents the boxed section of the plot at an expanded timescale.

We note that the overlap in the model variability predicted for the edge and middle spines as calculated for the spine-to-spine distance statistics implies that little statistical differences can be expected from these two groups for this particular 3-spine stimulation paradigm.

Next, we performed the same re-analysis for our 7-spine experiments and predicted the range of plasticity responses for the middle and edge spines using the maximal and minimal spine-to-spine distance within this experiment. Here, our model predicted significant overlap in the early stages of the plasticity dynamics (2 and 10 min) and starting from 20 min the model indicated that the plasticity response for the edge spines would be above the responses of the middle spines. This is in line with the data. We found that the response curve for the edge spines (green) lies above that of the middle spines (blue) and these responses differ significantly at +20 min (see figure below).

Caption: When considering edge spines (green) and middle spines (blue), our model predicts a significant difference, given our experimental spine-to-spine distance, for timepoints larger than 10 min. The experimental data reveals significantly different dynamics

at $t = 20$ min between the edge (light grey) and middle (dark grey) spines. $*p = 0.03$, two-sided t -test. Error bars represent the standard error of the mean.

We extended the same re-analysis for our 15-spine experiments and predicted the range of plasticity responses for the middle (7 spines) and edge spines (4 spines flanking middle spines) using the maximal and minimal spine-to-spine distance within this stimulation scenario. See Figure below.

*Caption: When distinguishing between 8 edge (green) and 7 middle spines (blue) and considering the variability in their pairwise distances, our model predicts increasing differences for later time points. This difference between the edge (light grey) and middle (dark grey) spines is statistically significant for $t=30$ min and $t=40$ min. $**p=0.005$ and $*p = 0.04$, two-sided t -test. Error bars represent the standard error of the mean.*

Therefore, supported by the model, we hypothesise that the competition in the three-spine scenario is not sufficient to cause significant differences between spine types, but as more spines are stimulated, a growing difference between the edge and middle spines is observed, whereby the response of the edge spines is above that of the middle spines.

To summarise, we are grateful for this comment and have now added the explanation above to the description of the 3-spine stimulation experiment in our manuscript as follows:

“To understand how the different cluster sizes and the distances between spines may affect the plasticity dynamics, we studied the statistics of the spine-to-spine distances in our 3-spine stimulation paradigm and calculated the predicted model dynamics for the minimal ($2.2 \mu\text{m}$) and the maximal ($4.2 \mu\text{m}$) experimentally observed cluster size. The model prediction for the mean cluster size ($3.2 \mu\text{m}$) is shown as a bold line, and the space between the model predictions for the minimal and maximal distances is represented by a shaded area of the same colour (green for edge and blue for middle) in Fig. 2e. This provides a model-based way of quantifying the variability in our spine dynamics. We note that the model dynamics we obtain from our 3-spine stimulations exhibits a substantial overlap between the middle and edge spines. Given this overlap, the model predicts that edge and middle spines would exhibit a similar plasticity response. This matches our experimental data as shown in Fig 2e. Moreover, when comparing sham-stimulated spines, while both types of spines grew significantly at 2 min, only the edge spines at 40 min were significantly enlarged (Fig. 2g and Fig. 2h, respectively).”

Given these results, we hypothesise that when 3 spines are stimulated the spine-spine competition for the available resources is not sufficient to cause significant differences between edge and middle spines. “

To describe the 7 and 15 spine stimulation experiments we have added the following text:

“Following the steps we used to model the 3-spine stimulation experiment in Fig. 2, we studied the plasticity response of the edge and middle spines in the experiment where 7 spines were simultaneously stimulated. The distribution of minimal spine-to-spine distance in the experiment with 7 stimulated spines is shown in Fig. 4g. Fig. 4f displays the plasticity response of the three inner spines (middle spines in blue) alongside the plasticity response of the four outermost spines (edge spines in green). To compare model predictions with recorded data, we first calculated the response dynamics given the largest ($13.1 \mu\text{m}$) and smallest ($6 \mu\text{m}$) cluster size, with the central line defining the average cluster size ($\approx 10.6 \mu\text{m}$). We find significant overlap in the early stages of the predicted dynamics (2 and 10 min). At later time-points, the edge and middle spines become more distinct, which suggests that there may be significant differences in the dynamics between edge and middle spines of the experimental data. As predicted by the model, we find that edge spines are significantly larger at +20 min than the middle spines (see Fig. 4f). Interestingly, this difference was

larger than that observed previously in the three-spine experiment (cf. Fig. 2e-f). This implies that a high number of stimulated spines, which compete for resources, not only has global effects but also impacts local resources creating an abundance of P and a lack of C which are more pronounced at certain spines compared to others (in line with the first model prediction)."

[...]

"Finally, our goal was to investigate whether increasing the competition further by stimulating more spines (15 simultaneously stimulated spines) would change the plasticity dynamics as predicted by our model. To this end, we generated a set of model predictions using the same model parameters as above and stimulating 15 spines in total whose smallest, mean, and largest cluster sizes were 20 μm , 28.3 μm and 30.6 μm , respectively. Next, we compared the predicted dynamics (Fig. 5f) with the experimentally observed results. We note that the model has good NMSE values in both edge and middle cases, but the R2w value shows that the experimental variability is missed by the model (Reply comment: please see below for an explanation of NMSE and R2w values). Additionally, we found that the edge spines (4 outermost spines on each side) responded stronger than the middle spines (7 inner spines flanked by 4 edge spines on either side) at +30 and +40 min post stimulation, in line with the model predictions. These observations are consistent with C and P dynamics of the model (Fig. 5g-i). As defined by the initial condition, the edge spines start with a higher C (because they have slightly more neighbouring resources; see grey horizontal dashed lines). C then diffuses away rapidly, making them comparable to the spines in the middle of the cluster after 2 min. In contrast, the behaviour of the protein variable P is slightly different because it evolves on longer time scales (see Fig. 5h). As more C is initially available at the edge spines (due to less competition), more P is subsequently generated at those edge spines. However, for time points past 2 min, P in the middle spines accumulates due to the diffusion of P from those spines on either side of the middle spine that are closer to the edge (in addition to the local activation of P) and is coupled with the lower degradation of P. This pooling of P amongst middle spines affects their longer time spine plasticity dynamics, which leads to lower potentiation and more rapid depression back to the baseline compared plasticity of the edge spines (see Fig. 5f). Therefore, supported by the model, we hypothesise that the increased competition in the 15-spine paradigm leads to larger differences in the plasticity responses within the stimulated cluster compared to plasticity responses in the three-spine and seven-spine cases."

For completeness, let us state the definitions of the quality of fit measures we use:

Our first fit quality measure is the Normalised Mean Squared Error (NMSE). The NMSE is a normalised version of the Mean Squared Error (MSE), a common metric for evaluating regression models by quantifying the difference between a model's predicted values and the values observed from experiments. The normalisation of the MSE is done by dividing by the variance of the observed data and making the NMSE a relative measure rather than an absolute one. To extend this to temporal data, we calculate the NMSE at each available snapshot and then average across those snapshots to get a mean NMSE value across the full dynamics. Mathematically, we define this average NMSE as:

$$\text{NMSE} = \frac{1}{N} \sum_{j=1}^N \frac{\sum_{i=1}^n (S_{\text{pred},j} - S_{\text{expt},j,i})^2}{\sum_{i=1}^n (S_{\text{expt},j,i} - \bar{S}_{\text{expt},j})^2}$$

where N is the number of snapshots, n is the number of data points for snapshot j, $S_{\text{pred},j}$ is the prediction of the model for the j snapshot, $S_{\text{expt},j,i}$ is the observed value of the i data point of snapshot j and $\bar{S}_{\text{expt},j}$ is the average of the observed data points of snapshot j. The numerator represents the standard MSE and the denominator represents the normalisation by the variance at each point. We note that the minimum value is achieved when $S_{\text{pred},j} = \bar{S}_{\text{expt},j}$, which leads to NMSE = 1. Therefore, the closer the value is to 1, the better the fit to the mean of the data at each snapshot, i.e., the fit follows the mean dynamics.

Additionally, we introduce a second fit quality measure: a variation of the commonly used R² metric, which measures the fitness of a linear fit. Here, we adapted this by using a weighted R² value (R_w²), where we compare the model predictions against the mean of the data at each of the snapshots, weighted by the inverse of the variance at that point. This means that points with higher variance are less critical to the metric. Finally, we normalise this value against the total variation of the dataset, i.e., the mean of each observed snapshot against the mean of all data points. Mathematically, we describe this as:

$$R_w^2 = 1 - \frac{\sum_{i=1}^N w_i (S_{\text{pred},i} - \bar{S}_{\text{expt},i})^2}{\sum_{i=1}^N w_i (\bar{S}_{\text{expt},i} - \bar{S}_{\text{expt}})^2}$$

By incorporating variance as a weighting factor, this aims to account for the heteroscedasticity (varying spread) of the data across different time points. A higher value of R_w² (closer to 1) indicates a better fit, considering both the model's predictive accuracy and the inherent variability of the data. More concretely, the R_w² measures how much better our model performs over a model with a constant horizontal line which takes the mean of the experimental data as its height.

Finally, we were particularly intrigued by the reviewer's suggestion (similar to the comment of the first reviewer) to study the time scales of spine volume decay in the temporal window of 2-10 minutes after induction and test whether there are significant differences across stimulation paradigms. Therefore, we decided to fit an exponential model ($f(x) = a \cdot \exp(-b \cdot x + c)$) to the different experimental paradigms (1 spine, 3 Spine, 7 Spine and 15 spine control settings) to quantify the decay time scales. The result are shown in the new figure 6d and e and below:

Caption excerpt: d) Performance of our model vs a simple exponential $f(x) = a \exp(-bx) + c$ with varying parameters across experimental paradigms along with NMSE and R^2_{wp} values. e) A change in the exponent b of the exponential model across simulation scenarios indicates a non-linear dependence of the temporal plasticity scales.

In panel d of the figure above, we show the exponential fits (fit parameters a , b , and c) to the average experimentally recorded dynamics (considering error bars) for each of the experimental stimulation paradigms. In all cases, the NMSE is close to 1, which follows intuitively as this quantity is correlated with the value we minimise when performing the fitting procedure. We note that this simpler model also reproduces the general features of the dataset (e.g., ordering at 2 min, decay rates, and final saturation value at 40 min). Additionally, the exponential decay describes the 1, 3, and 7 spine experiments well. However, these good results are obtained by directly fitting the exponential form to the data in comparison to the predictive capabilities of the more complex model. However, qualitative differences exist between the experiments and the simple exponential model when comparing the 15-spine paradigm. In this setting, there is very little decay (the dynamics are almost linear) over the observed time, meaning that the exponential model nearly instantaneously decays to the saturation point and remains there. In comparison, our more complex model can reproduce this slow decay, leading to better quantitative goodness-of-fit measures and providing a possible explanation of the underlying mechanisms.

The exponential model can provide valuable insights into some aspects that the more complicated model cannot. Notably, as suggested by the reviewer, the decay rate (parameter b) is a parameter that we can compare between experiments to determine quantitative differences. To this end, we took the individual spines of each experiment and generated the exponential fit for each of these. We omitted those spines where the resulting decay rate was larger than 10, as the sparse temporal resolution coupled with the noisy biological nature of the spines led to some spines having erroneous fit results. This then led to a set of decay parameters we could compare in Fig 6e. We note differences between the seven spine and other experiments, where the spines in the seven spine experiment show a significantly slower decay from the initial transient potentiation than the other two multi-spine stimulation paradigms. We would initially assume that the 15-spine experiment would have the slowest decay. However, given the semi-linear nature of the dynamics, the fits are not so easily obtained in this scenario, a problem that our model, which predicted rather than was fit to the dynamics, did not face. For our model, the predictive power remained valid across different stimulation paradigms showing that the internal model parameters obtained in one stimulation paradigm generalise to varying numbers of stimulated spines. For example, we found that the smaller peak potentiation and slower plasticity decay observed in the 15-spine experiment could be explained by the increased competition among the spines, leading to the lower amount of P per stimulated spine. In our model, different amounts of P give rise to the different speeds at which the spines potentiate and then depress after stimulation.

We have added this discussion to the updated manuscript as follows:

“To gain insight into the temporal decay of the spines following the initial potentiation, we have attempted to fit an exponential decay model of the form $f(x) = a \exp(-bx + c)$ to each experimental data set. While the simple exponential fits will not provide us with the mechanistic insight into the plasticity dynamics that our original model does, it can help quantify the decay differences via the changes in b .

Firstly, we fit the exponential model (fit parameters a , b , and c) to the mean dynamics of each experimental paradigm in Fig. 6d. In all cases, the NMSE is close to 1, which follows intuitively as this quantity is correlated with the value we minimise when performing the fitting procedure. We note that this simpler model also reproduces the general features of the dataset (e.g., order of efficacy with which spine size increase is observed at 2 min, decay rates of the initial transient potentiation, and final level of spine size increase at 40 min). We emphasise that these results are directly obtained from fitting the simple model and do not provide a prediction like the more complex model. We note that qualitative differences exist between the experimental data and the simple exponential model when comparing the 15-spine paradigm. In this setting, the stimulated spines do not show a decline in size after initial potentiation (or do so very slowly), which leads to the insufficiency of the exponential model in describing this data (instantaneous decay to reach the saturation point, i.e. stable potentiation). In comparison, the more complex model can reproduce this slow decay, leading to better quantitative goodness-of-fit measures, possibly as a result of more aptly representing the underlying mechanisms.

To address the differences in decay rates of the initial potentiation, we took the individual spines of each experiment and generated the exponential fit to each of these. We omitted those spines where the resulting decay rate was larger than 10, as the sparse temporal resolution coupled with the noisy biological nature of the spines led to some spines having erroneous fit results. This procedure led to a set of decay parameters we could compare in Fig. 6e. We note differences between the seven spine and other experiments, where the seven spine experiment decays significantly slower from the initial transient potentiation than the other two paradigms. We initially assumed that the 15-spine experiment would have the slowest decay. However, given the semi-linear nature of the dynamics, the fits are not easily obtained in this scenario (as seen in the previous fit of the average dynamics). For our model, the predictive power remained valid across different stimulation paradigms showing that the internal model parameters obtained in one stimulation paradigm generalise to varying numbers of stimulated spines. For example, we found that the smaller peak potentiation and slower plasticity decay observed in the 15-spine experiment could be explained by the increased competition among the spines, leading to the lower amount of P . In our model, the different amounts of P were the reason behind the different speeds at which the spines were depressing after stimulation.

4. The distances between the three spines shown in Figure 2a-c are not similar, but the two spines are in close vicinity. If two spines are closer, the effect on edge spines on this side may be weak. It is desirable to incorporate this point into the analysis.

We are grateful to the reviewer for carefully reading our manuscript and mentioning an excellent point we had not initially considered. The original three-spine example displayed a great disparity in the distance between the three spines, with two being clustered and one on its own. As the reviewer correctly points out, the effect on the single-edge spine might be minimal, which is indeed confirmed by the model. However, we note that our example in the original manuscript did not accurately reflect the distribution of our "standard" three spine paradigm. To this end, we have replaced it with a more typical example that better represents our experimental set displaying a more homogenous spatial distribution (see below):

In order to fully address the reviewer's suggestion, we also tested whether our equidistant assumption (and thus treating the symmetrical edge spines on either side as the same) was critical to any model outcomes. We calculated the average spatial percentage of the middle spine (i.e., the fraction of the full cluster size where the middle spine lies) and found that this value for our control was $0.73\% \pm 0.05\%$ (s.e.m.). As this does represent a certain amount of asymmetry, we next simulated this setting using the model to see how the two types of edge spines would behave:

As shown in the plot above, we found very subtle differences between the cluster (the edge spine close to the middle) and the single-edge spine (red lines). However, we also find that the equidistant edge spine (blue) falls between these two spines, which implies that an average of the edge spines in the asymmetric case would be comparable to the equidistant scenario. Wishing to keep the model and its set-up as simple as possible, we have continued with the equidistant approach in the updated manuscript. Nonetheless, the reviewer's point adds an interesting dimension: Is there a particular arrangement of active spines that maximally drives clustered synaptic plasticity, and does this arrangement differ for different dendrites/neuronal cell types? In our current study, we cannot answer this question given that the focus was not on the spatial distribution per se but more on the number of spines stimulated. We thank the reviewer once more for this interesting point and have added these points to the discussion as follows:

“Let us note that our model predictions are derived for equidistantly distributed spines; however, our model and the code we provide can be used to explore generalisations. For example, our results indicate that small clusters of 3-5 spines could lead to a larger plasticity amplitude compared to scenarios where ten or more spines are clustered together or if spines are spatially further apart from each other. Future studies could clarify the precise number and cluster size along a dendrite to maximise plasticity outcomes.”

5. Information about the optimization of parameters should be provided. For example, the ratio between Cs and Cd may influence the initial spine enlargement, but how critical is this value in mathematical modeling? Similarly, does the larger parameter phi, which removes the transition phase, disrupt the pattern of a slow decrease in spine volume after stimulation? These points may help understand the contribution of each parameter in the temporal pattern of spine responses.

We thank the reviewer for this important comment. We optimised the parameters using a non-linear adjoint looping approach, a gradient-based method particularly suited to optimising multiple interrelated parameters within a dynamical systems setting. Nonetheless, as we are optimising a non-linear system using an approach relying on a gradient, it is important to consider the impact of global vs. local solutions. To account for this, we initiated the optimisation routine ten times with several randomly chosen starting points. Within each case, we arrived at similar solutions that could generalise to the other experimental datasets by altering the amount of available P and C, as demonstrated in our manuscript. We have expanded on this point in the manuscript as follows:

“Given that our approach is gradient-based, and we are employing it in a non-linear setting, it is important to consider the impact of global vs. local solutions. To account for this, we initiated our parameter optimization algorithms ten times with multiple randomly chosen starting points. In each case, the algorithm proposed similar solutions that generalised to the other test scenarios with different numbers of stimulated spines equivalent to the parameter set we used in our manuscript.”

Additionally, we are grateful to the reviewer for mentioning that the ratio of Cs and Cd in our model may be important to consider in detail. To study the effect of having more or less Cs/Cd, we took advantage of the model fits that we obtained from the 3-spine paradigm and transformed all available C (which in the optimal fits is made up of a mix of C_s) to be entirely composed of the somatic component, Cs, or dendritic component, Cd. We display the result of this analysis, as well as the experimental data and optimal model predictions, in the figure below. Since the superposition of the data and model predictions per experiment can be found in the manuscript, we do not display them below.

Top left: Comparative summary of experimentally measured synaptic plasticity responses across stimulation paradigms with 1, 3, 7 or 15 spines. **Top right:** Using model parameters obtained from data in Fig. 2 (fit of the 3-spine experiment) allowed to predict the temporal dynamics in experiments with varying numbers of stimulated spines. **Bottom left:** Using the same model parameters, but setting the amount of shared dendritic to zero C (Cd=0) removed synaptic competition (only the spine component Cs remained) lead to erroneous model predictions (for example: increasing potentiation as more spines are added). **Bottom right:** Using the same model parameters, but setting the amount of spine specific C to zero C (Cs=0) increases synaptic competition, once more leading to erroneous model predictions (such as progressively smaller potentiation response).

The two upper panels of the above figure are taken from a new figure (figure 6 in the updated manuscript, requested by reviewer 1) that shows a comparative analysis between stimulation paradigms. We note that the model (with optimal fits from the 3-spine

experiment) provides good agreement with the experimental data from the 7-spine experiment, recreating all the major trends (such as the growth at +2 minutes and final saturation at +40 minutes) without the need for parameter re-fitting.

However, when we alter the nature of C, by removing the competitive component C_d , (but still maintaining the same net amount of C throughout the system), the predictive power of the model is lost. Because C is no longer subject to competition it means that as more spines are added, inherently more C is added to our system which leads to a stronger initial potentiation response. We do not observe this feature in the experimental data, where the maximum potentiation is achieved for the 3 spine example followed by a decline for the 7 and 15 spine example. Additionally, as all competition is now driven by P, we note that the difference between the edge and middle spines in the experiment with 15 stimulated spines happens later, at 30 and 40 mins.

On the other hand, defining C as exclusively dendritic and removing the uncontested somatic component ($C_s=0$) seems to generate model predictions that closer match the optimal model predictions (albeit with lower potentiation for all experiments with more than 1 spine at +2 minutes in comparison to the fit model) but, in particular, the prediction for the 15 spine experiment is not correct. As we stimulate 15 spines, the model without C_s predicts that the potentiation response stays close to 1 as the competition has almost completely removed C from the system. When compared to the experimental results, we note, it is in fact the 15 spine paradigm that has the highest normalised volume at +40 minutes. Therefore, it seems a compromise involving both components, C_s and C_d , allows spines to potentiate even in the face of very strong competition. Comparing all three scenarios, it seems that C_s and C_d become important for different mechanisms: C_d seems important when small numbers of spines are stimulated while C_s plays an important role when large numbers of stimulated spines are involved. This is also reflected in the ratio of optimal fits for C_s and C_d (which is approximately 10:1).

Let us note that the exact C_s/C_d ratio may be hard to pin down precisely because of our temporal resolution in the data, but each value, C_s and C_d , seems to be important for the ability of our model predictions to be valid across different stimulation scenarios. It is plausible that the ratio between C_s and C_d has an impact on the difference of the temporal dynamics of the edge and middle spines. If the competition is driven primarily by the C dynamics (which is a fast component), we should expect to see a very rapid differentiation between these spines, while if the competition is driven primarily by the P dynamics (which acts on slower scales), we would expect to see differentiation between edge and middle spines later, at 30 mins and beyond (see difference in initial C distributions below):

Left: In our original model fit we started with $C_s/C_d= 1/10$. Right: Removing C_d while keeping the total C (an equivalent amount of C_s) leads to a large C_s/C_d . As C_s is not subject to competitive elements, the difference between the edge and middle spines disappears. Instead we have an equal amount of C at each spine (compare with left, where a noticeable dip in C_d leads to differences in C at each spine).

The reviewer also mentions the ϕ parameter as a possible critical component in the temporal dynamics of the spines. Because we kept this parameter constant between experimental paradigms (i.e., once fit to the 3-spine experiment, we kept all biophysical model parameters constant), we didn't consider the consequences of changing this parameter. Nonetheless, we find the suggestion to study this parameter particularly intriguing. To this end, we took the ϕ parameter and either multiplied or divided it by 10 to get an insight into the resulting spine dynamics (here, we took the 15-spine experiment as an example and studied the mean of the stimulated spines). The resulting figures are shown below (left is decision variable F, and the right is the resulting synaptic dynamics).

We note significant differences in the temporal dynamics and even the final saturation result (the strength and timing of the decay are affected due to the altered effect of P). When using the smaller ϕ value ($0.1 * \phi$), we have a very shallow slope, which indicates that there is little difference when S is above or below the threshold ν , which is reflected in the example spine dynamics (very small decay after growth). On the other hand, using a ten times larger ϕ leads to a very abrupt change from the potentiating to the depressing regime. Additionally, as we increase ϕ , we note that the strength of F(S) is maximised, and small deviations from the equilibrium point already achieve the maximum growing or shrinking potential. Therefore, it seems that this parameter, beyond altering the temporal dynamics of P, ϕ , also has a significant impact on the plasticity outcome.

We have an added text to the revised manuscript discussing this effect while focusing on the ratio of the C components and the parameter phi) to the supplemental material as follows:

“Let us note that the effects of certain biological parameters, such as the degradation rates or location of the P threshold, can be intuitive to understand. However, interpreting the effects of other model parameters such as the steepness of the function $\phi(S)$ or the ratio C_s/C_d may be harder. To build intuition, we studied the effects these parameters have on the plasticity outcomes. Removing the spine-specific component of C_s we observed less competition as more spines are stimulated and the shared dendritic store is depleted. On the other hand, when we removed the competitive element of C (C_d) this resulted in stronger potentiation among stimulated spines since more C was now activated per stimulation site. Let us note that the exact C_s/C_d ratio may be hard to pin down precisely because of our temporal resolution in the data, but each value, C_s and C_d , seems to be important for the ability of our model predictions to be valid across different stimulation scenarios. In addition, the C_s/C_d ratio had an impact on the difference between edge and middle spines. In stimulation scenarios where the spine competition is driven primarily by the fast C dynamics, our model predicts a rapid differentiation between edge and middle spines, whereas if the spine competition is driven primarily by the P dynamics (which acts on slower scales), the model predicted edge-to-middle difference later, after 20 minutes.

Studying the effect of ϕ on the plasticity outcome we found that it mediates how strongly P is able to contribute to potentiation and whether its contribution is of equal strength as that of C . For example, when using the smaller ϕ value, $F(S)$ exhibits a shallow slope and the potentiation effect of P will be severely dampened close to the threshold. In contrast, using a large ϕ will lead to an instantaneous shift from potentiation to depression. Similarly, if ϕ is small, the size-dependent feedback mechanism is not able to suppress growth of large spines which means that potentiation amplitude will be stronger for small ϕ .

Minor points,

1. In the Results section, the authors wrote “At its core, the model depends on two distinct proteins: C, which acts on faster timescales”. It is confusing where C is postulated as a protein or a small molecule such as calcium.

We agree with the reviewer that a more precise treatment of C would be beneficial. We now have updated the text as follows:

“We emphasise that our model does not rely on a particular molecular identity of the parameter C; instead, we believe it exhibits the characteristics consistent with the above mentioned candidate proteins/molecules. Therefore, C can be interpreted to be a fast protein, a small molecular compound that reacts quickly to spine stimulation, or potentially a combination of both. Since our model simulations of C in supplemental Figure S1g and the highly localised nature of Ca^{2+} in our experiments in supplemental Figure S2 are compatible with each other, C can also represent Ca^{2+} directly.

2. The decay curve of C in Figure 2i is difficult to evaluate. Individual curves should be presented.

We thank the reviewer for their suggestion to improve the figure. Our re-analysis considering the distance variations between spines found that in the three-spine stimulation paradigm the model predicted that there was no observable difference between the edge and middle spines, which was confirmed by the experimental data. Therefore, we revised Figure 2 by removing the original panel i. In a new set of analyses comparing plasticity dynamics of edge and middle spines, we found that the 15-spine example displayed significant differences between edge and middle spines. New plots describing the dynamics of C and the amount of P associated with 15-spine experiments have been added to Fig 5 (panels g,h). In particular, taking the reviewer’s suggestion concerning the difficulty of evaluating the decay curve of C, new Fig 5g depicts the difference between C at the edge and the middle spines, which can help evaluate the temporal dynamics of C according to location. New versions of figure 5g and h, and the associated caption, is below.

g) Difference in the amount of C at the edge vs middle spines reveals that the edge gets initially more C (less competition) and thus potentiates more strongly. h) Temporal dynamics of P for the edge (darker) and middle spines (lighter). We note that middle spines accumulate more P due to spillover compared to edge spines .

3. In the text related to Figure 3g, the authors wrote “single spines rapidly grow (as there is no competition for C) but then equally rapidly return to the baseline.” But Figure 3g shows the normalized spine volume at t = 40 min is about 1.3, not returning to the baseline.

The reviewer suggests a very good clarification that we have now included in the updated manuscript. The new sentence is follows: “[...] but then equally rapidly shrinks back to a lower value of approximately 1.3 (as there is no competition for P)”

4. The spine volume data is normalized against the three observed data points immediately preceding the glutamate uncaging. This strategy is adequate for comparing single spine responses, but the data shown in Figure 5g-i calculate the total spine responses, which may require incorporating differences in the initial spine volume before stimulation.

We thank the reviewer for the insightful comment and allowing us to clarify our analysis. To study the total growth in the spine size, the reviewer correctly points out that if the unnormalised spine size is significantly different pre-stimulation, that the total growth post stimulation may be impacted unintentionally. To make sure that this was not the case, we analysed the full distribution of unnormalised spine sizes before stimulation as follows. During the experiments, we attempted to select spines of similar sizes (medium sizes spines). Below we see the distribution of spine sizes for each of the control experiments:

We note that the 1, 3 and 7 spines all have similar averages, while the variances are larger for experiments involving more spines. In the 15 spine experiments do exhibit a wider range of sizes and also a slightly higher mean size. This is a natural result of the design of the experiment: a need to select 15 relatively clustered spines considerably limits the freedom in choosing the spine size as we require a dendritic branch carrying 15 spines showing a confined spatial distribution and that are also in the same z plane of the microscope.

Despite the limitations, we note that the percentage of spines in the 7 and 15 spine experiments whose sizes fall in the range of the three spine experiment (i.e., the number of spines that are bigger than the smallest spine in the 3 spine experiment and smaller than the largest spine in the 3 spine experiment) is 86.7% percent and 88.3% percent, respectively.

We hope that this demonstrates that the unnormalised spine sizes are within similar ranges allowing us to then compare the total growth post stimulation.

5. In the legend of Figure 3 a-c), the words “bottom row sham” should be removed. This point should be discussed.

We are grateful to the reviewer for spotting this. The words “bottom row sham” originated from an early version of the figure and have therefore removed the words “bottom row sham” in the updated manuscript.

6. In the legend of Figure 4 a-c), “Example image of the seven-spine experiment for min before” should be “for 5 min before”.

We thank the reviewer for the careful reading of our manuscript. We have now corrected the legend.

Reviewer #3 (Remarks to the Author):

The authors explore how plasticity of dendritic spines depends on the number and spatial proximity of other nearby active synapses, focusing on the impact of local competition for resources. They use a computational model incorporating putative plasticity-regulating proteins with various spatial and temporal dynamics to predict differences in magnitude and time course of structural LTP as a function of cluster dimensions and the position of the spines in the cluster. They compare the model predictions with experimental measurements of spine head size after inducing LTP by glutamate uncaging in pyramidal neurons in organotypic culture.

The activity-dependent factors that shape the strength of individual synapses in the local dendritic environment are far from being completely elucidated, and more experimental and theoretical work is clearly needed to better understand local synaptic plasticity rules. However, in my opinion the conclusions of the current paper have not been convincingly supported by the results provided. I have several concerns regarding the analysis and interpretation of the data and the limitations of the applied approach.

Major comments:

1) The authors make several claims that the experimental results confirm the model's predictions, which are not sufficiently supported by the presented data and analysis.

- For the 3-spine cluster (Fig 2), the model predicts that the increase of spine head volume is larger in the first 10 minutes in spines at the cluster edge than those in the middle, followed by a decline to similar levels by 20 minutes (Fig 2f). However, the data (Fig 2e, g, h) do not demonstrate a significant difference between the middle and edge spines stimulated in the same clusters in the same experiments. The only difference is reported when the 'edge' spine group is compared to independent sham experiments at 40 minutes, however this cannot substitute for the lack of significant effect in the relevant direct comparison of middle and edge spines. Overall I do not see that the measured results would agree with the model's prediction.

We thank the reviewer for mentioning this key point and for suggesting to provide more clarity about when significant differences can be expected in the data and when the model indicates no difference between edge and middle spines. We followed the reviewer's suggestion, along with the suggestions of reviewers 1 and 2, and now provide additional analysis to quantify the effect size of our model prediction and check whether our experimental data for the 3-spine stimulation protocol is in line with our theory. We believe interpreting the 3-, 7- and 15-spine stimulation experiments need to be considered together to understand when significant differences between the edge and middle spines can be expected and we now took several measures to clarify this point in the revised manuscript, which we detail below.

In the previous version of the manuscript, we generated model predictions using the average size of the clusters across each experimental paradigm. Then, we used this distance of average clusters to generate an equidistant set of spines corresponding to the average spine-to-spine distance in our data. While this assumption leads to a reasonable prediction of the average stimulated spine dynamics, upon re-analysis, we discovered that it was not precise enough to accurately describe the dynamics of the edge and middle spines individually. To understand how the different cluster sizes and spine-to-spine distances may affect the dynamics, we generated the set of spine-to-spine distances and then calculated the predicted model dynamics for the minimal and the maximal experimentally observed spine-to-spine distance. We now display the model prediction for the mean spine-to-spine distance as a bold line and the model predictions for the minimal and maximal distances as a shaded area of the same colour (green for edge and blue for middle). This provided us with a model-based way of quantifying the variability in our spine dynamics. The new corresponding model figure is included below; the inset shows the predicted dynamics between 0-10 min at expanded timescale.

Caption: Using the maximal and the minimal spine-to-spine distances in our experimental data allows us to delineate the expected variability in the dynamics of the edge (green) and middle (blue) spines as well as their average predicted behaviour. The bold line represents the respective mean response. The model predicted overlap in the edge and middle spine dynamics indicates that no significant differences between these two spine types can be expected in this experimental scenario. Inset represents the boxed section of the plot at an expanded timescale.

We note that the overlap in the model variability predicted for the edge and middle spines as calculated for the spine-to-spine distance statistics implies that little statistical differences can be expected from these two groups for this particular 3-spine stimulation paradigm. Therefore, supported by the model, we hypothesise that the competition in the three-spine scenario is not sufficient to cause significant differences between spine types, but as more spines are stimulated, a growing difference between the edge and middle spines is observed, whereby the response of the edge spines is above that of the middle spines.

To summarise, we are grateful for this comment and have now added the explanation above to the description of the 3-spine stimulation experiment in our manuscript as follows:

"To understand how the different cluster sizes and the distances between spines may affect the plasticity dynamics, we studied the statistics of the spine-to-spine distances in our 3-spine stimulation paradigm and calculated the predicted model dynamics for the minimal (2.2 μm) and the maximal (4.2 μm) experimentally observed cluster size. The model prediction for the mean cluster size (3.2 μm) is shown as a bold line, and the space between the model predictions for the minimal and maximal distances is represented by a shaded area of the same colour (green for edge and blue for middle) in Fig. 2e. This provides a model-based way of quantifying the variability in our spine dynamics. We note that the model dynamics we obtain from our 3-spine stimulations exhibits a substantial

overlap between the middle and edge spines. Given this overlap, the model predicts that edge and middle spines would exhibit a similar plasticity response. We could confirm this finding in our experimental data, see Fig. 2e. Moreover, when comparing sham-stimulated spines, while both types of spines grew significantly at 2 min, only the edge spines at 40 min were significantly enlarged (Fig. 2g and Fig. 2h, respectively).”

- The authors say that they demonstrated that sLTP of single stimulated spines decays faster than that of clustered potentiated spines. However, no statistical analysis is provided, and no evident difference in kinetics can be seen when comparing the data in Fig 2d and Fig 3d. In fact, the differences predicted by the model should be observed within the first 10 minutes, but such short-term changes cannot be evaluated with the low temporal resolution of the measurements.

We thank the reviewer for mentioning this important point. We agree that adding more statistical quantification would provide more clarity on the plasticity time constants. At the suggestion of another reviewer, we decided to fit a simple exponential model ($f(x) = a \cdot \exp(-b \cdot x + c)$) to the different experimental paradigms (1 spine, 3 spine, 7 spine and 15 spine scenarios). The resulting exponential fits are now superimposed for direct comparison and are now shown in the new figure 6d and e below. For visual clarity we avoided placing the experimental data on top of each fit (and provide instead quality of fit measures), however if necessary we are happy to place individual fits and the corresponding data into an additional supplemental figure.

Caption excerpt: d) Performance of our model vs a simple exponential $f(x) = a \exp(-bx) + c$ with varying parameters across experimental paradigms along with NMSE and R^2_w values. e) A change in the exponent b of the exponential model across simulation scenarios indicates a non-linear dependence of the temporal plasticity scales.

In panel d of the figure above, we show the exponential fits (fit parameters a , b , and c) to the average experimentally recorded dynamics (considering error bars) for each of the experimental stimulation paradigms. In all cases, the NMSE is close to 1, which follows intuitively as this quantity is correlated with the value we minimise when performing the fitting procedure. We note that this simpler model also reproduces the general features of the dataset (e.g., ordering at 2 min, decay rates, and final saturation value at 40 min). Additionally, the exponential decay describes the 1, 3, and 7 spine experiments well. However, these good results are obtained by directly fitting the exponential form to the data in comparison to the predictive capabilities of the more complex model. However, qualitative differences exist between the experiments and the simple exponential model when comparing the 15-spine paradigm. In this setting, there is very little decay (the dynamics are almost linear) over the observed time, meaning that the exponential model nearly instantaneously decays to the saturation point and remains there. In comparison, our more complex model can reproduce this slow decay, leading to better quantitative goodness-of-fit measures and providing a possible explanation of the underlying mechanisms.

The exponential model can provide valuable insights into some aspects that the more complicated model cannot. Notably, as suggested by the reviewer, the decay rate (parameter b) is a parameter that we can compare between experiments to determine quantitative differences. To this end, we took the individual spines of each experiment and generated the exponential fit for each of these. We omitted those spines where the resulting decay rate was larger than 10, as the sparse temporal resolution coupled with the noisy biological nature of the spines led to some spines having erroneous fit results. This then led to a set of decay parameters we could compare in Fig 6e. We note differences between the seven spine and other experiments, where the spines in the seven spine experiment show a significantly slower decay from the initial transient potentiation than the other two multi-spine stimulation paradigms. We would initially assume that the 15-spine experiment would have the slowest decay. However, given the semi-linear nature of the dynamics, the fits are not so easily obtained in this scenario, a problem that our model, which predicted rather than was fit to the dynamics, did not face. For our model, the predictive power remained valid across different stimulation paradigms showing that the internal model parameters obtained in one stimulation paradigm generalise to varying numbers of stimulated spines. For example, we found that the smaller peak potentiation and slower plasticity decay observed in the 15-spine experiment could be explained by the increased competition among the spines, leading to the lower amount of P per stimulated spine. In our model, different amounts of P give rise to the different speeds at which the spines potentiate and then depress after stimulation.

We have added this discussion to the updated manuscript as follows:

“To gain insight into the temporal decay of the spines following the initial potentiation, we have attempted to fit an exponential decay model of the form $f(x) = a \exp(-bx + c)$ to each experimental data set. While the simple exponential fits will not provide us with the mechanistic insight into the plasticity dynamics that our original model does, it can help quantify the decay differences via the changes in b .

Firstly, we fit the exponential model (fit parameters a , b , and c) to the mean dynamics of each experimental paradigm in Fig. 6d. In all cases, the NMSE is close to 1, which follows intuitively as this quantity is correlated with the value we minimise when performing the fitting procedure. We note that this simpler model also reproduces the general features of the dataset (e.g., order of efficacy with which spine size increase is observed at 2 min, decay rates of the initial transient potentiation, and final level of spine size increase at 40 min). We emphasise that these results are directly obtained from fitting the simple model and do not provide a prediction like the more complex model. We note that qualitative differences exist between the experimental data and the simple exponential model when comparing the 15-spine paradigm. In this setting, the stimulated spines do not show a decline in size after initial potentiation (or do so very slowly), which leads to the insufficiency of the exponential model in describing this data (instantaneous decay to reach the saturation point, i.e. stable potentiation). In comparison, the more complex model can reproduce this slow decay, leading to better quantitative goodness-of-fit measures, possibly as a result of more aptly representing the underlying mechanisms.

To address the differences in decay rates of the initial potentiation, we took the individual spines of each experiment and generated the exponential fit to each of these. We omitted those spines where the resulting decay rate was larger than 10, as the sparse temporal resolution coupled with the noisy biological nature of the spines led to some spines having erroneous fit results. This procedure led to a set of decay parameters we could compare in Fig. 6e. We note differences between the seven spine and other experiments, where the seven spine experiment decays significantly slower from the initial transient potentiation than the other two paradigms. We initially assumed that the 15-spine experiment would have the slowest decay. However, given the semi-linear nature of the dynamics, the fits are not easily obtained in this scenario (as seen in the previous fit of the average dynamics). For our model, the predictive power remained valid across different stimulation paradigms showing that the internal model parameters obtained in one stimulation paradigm generalise to varying numbers of stimulated spines. For example, we found that the smaller peak potentiation and slower plasticity decay observed in the 15-spine experiment could be explained by the increased competition among the spines, leading to the lower amount of P . In our model, the different amounts of P were the reason behind the different speeds at which the spines were depressing after stimulation.“

- In 7-spine clusters (Fig 4), the authors claim that spines at the edge of the cluster potentiate more than spines in the middle. My understanding is that this is based on a single data point being different at 20 minutes, whereas no differences are indicated at other time points (including 2 and 40 minutes, which are analysed in the rest of the study). The authors also say in the text that the difference between middle and edge spines is larger in 7-spine clusters than in 3-spine clusters, but this is not confirmed by quantitative analysis.

We are grateful to the reviewer for raising this point. Following the suggestions of the reviewer, along with the suggestions of reviewers 1 and 2, we incorporated the analysis previously discussed in the 3 spine section in reply to point 1 of the reviewer, which aimed to study the inherent variability of our model to the seven and 15 spine example. More concretely, by generating the plasticity dynamics for the largest and smallest experimentally observed cluster size, note that at later stages of the model-predicted temporal dynamics, the edge and middle spines do not overlap. Therefore, only in this regime can we hope to get significantly different dynamics between these different types of spines. Indeed, we observe that the edge spines significantly differ at +20 min (see figure below). Importantly, the model predicts that the plasticity curve for the edge spines should be above that of the middle spines, which is also confirmed experimentally. Importantly, we obtained model parameters from the 3-spine stimulation experiments and generated our model prediction solely based on these parameters while adapting only the number of stimulation sites in the model and the spine-spine distances. This emphasises the generalisation ability of our model from 3- spine to 7-spines and their distinct edge and middle sub-classes.

Caption: When considering edge spines (green) and middle spines (blue), our model predicts a significant difference, given our experimental spine-to-spine distance, for timepoints larger than 10 min. The experimental data reveals significantly different dynamics at $t = 20$ min between the edge (light grey) and middle (dark grey) spines. * $p = 0.03$, two-sided t -test. Error bars represent the standard error of the mean.

We extended the same re-analysis for our 15-spine experiments and predicted the range of plasticity responses for the middle (7 spines) and edge spines (4 spines flanking middle spines) using the maximal and minimal spine-to-spine distance within this stimulation scenario. See Figure below.

Caption: When distinguishing between 8 edge (green) and 7 middle spines (blue) and considering the variability in their pairwise distances, our model predicts increasing differences for later time points. This difference between the edge (light grey) and middle (dark grey) spines is statistically significant for $t=30$ min and $t=40$ min. $**p=0.005$ and $*p=0.04$, two-sided t -test. Error bars represent the standard error of the mean.

To summarise, we are grateful for this comment and have now added the explanation above to the description of the 7-spine and 15-spine stimulation experiment in our manuscript as follows:

“Following the steps we used to model the 3-spine stimulation experiment in Fig. 2, we studied the plasticity response of the edge and middle spines in the experiment where 7 spines were simultaneously stimulated. The distribution of minimal spine-to-spine distance in the experiment with 7 stimulated spines is shown in Fig. 4g. Fig. 4f displays the plasticity response of the three inner spines (middle spines in blue) alongside the plasticity response of the four outermost spines (edge spines in green). To compare model predictions with recorded data, we first calculated the response dynamics given the largest ($13.1 \mu\text{m}$) and smallest ($6 \mu\text{m}$) cluster size, with the central line defining the average cluster size ($\approx 10.6 \mu\text{m}$). We find significant overlap in the early stages of the predicted dynamics (2 and 10 min). At later time-points, the edge and middle spines become more distinct, which suggests that there may be significant differences in the dynamics between edge and middle spines of the experimental data. As predicted by the model, we find that edge spines are significantly larger at +20 min than the middle spines (see Fig. 4f). Interestingly, this difference was larger than that observed previously in the three-spine experiment (cf. Fig. 2e-f). This implies that a high number of stimulated spines, which compete for resources, not only has global effects but also impacts local resources creating an abundance of P and a lack of C which are more pronounced at certain spines compared to others (in line with the first model prediction).”

[...]

“Finally, our goal was to investigate whether increasing the competition further by stimulating more spines (15 simultaneously stimulated spines) would change the plasticity dynamics as predicted by our model. To this end, we generated a set of model predictions using the same model parameters as above and stimulating 15 spines in total whose smallest, mean, and largest cluster sizes were $20 \mu\text{m}$, $28.3 \mu\text{m}$ and $30.6 \mu\text{m}$, respectively. Next, we compared the predicted dynamics (Fig. 5f) with the experimentally observed results. We note that the model has good NMSE values in both edge and middle cases, but the $R2w$ value shows that the experimental variability is missed by the model (Reply comment: please see below for an explanation of NMSE and $R2w$ values). Additionally, we found that the edge spines (4 outermost spines on each side) responded stronger than the middle spines (7 inner spines flanked by 4 edge spines on either side) at +30 and +40 min post stimulation, in line with the model predictions. These observations are consistent with C and P dynamics of the model (Fig. 5g-i). As defined by the initial condition, the edge spines start with a higher C (because they have slightly more neighbouring resources; see grey horizontal dashed lines). C then diffuses away rapidly, making them comparable to the spines in the middle of the cluster after 2 min. In contrast, the behaviour of the protein variable P is slightly different because it evolves on longer time scales (see Fig. 5h). As more C is initially available at the edge spines (due to less competition), more P is subsequently generated at those edge spines. However, for time points past 2 min, P in the middle spines accumulates due to the diffusion of P from those spines on either side of the middle spine that are closer to the edge (in addition to the local activation of P) and is coupled with the lower degradation of P. This pooling of P amongst middle spines affects their longer time spine plasticity dynamics, which leads to lower potentiation and more rapid depression back to the baseline compared plasticity of the edge spines (see Fig. 5f). Therefore, supported by the model, we hypothesise that the increased

competition in the 15-spine paradigm leads to larger differences in the plasticity responses within the stimulated cluster compared to plasticity responses in the three-spine and seven-spine cases.”

For completeness, let us state the definitions of the quality of fit measures we use:

Our first fit quality measure is the Normalised Mean Squared Error (NMSE). The NMSE is a normalised version of the Mean Squared Error (MSE), a common metric for evaluating regression models by quantifying the difference between a model's predicted values and the values observed from experiments. The normalisation of the MSE is done by dividing by the variance of the observed data and making the NMSE a relative measure rather than an absolute one. To extend this to temporal data, we calculate the NMSE at each available snapshot and then average across those snapshots to get a mean NMSE value across the full dynamics.

Mathematically, we define this average NMSE as:

$$NMSE = \frac{1}{N} \sum_{j=1}^N \frac{\sum_{i=1}^n (S_{pred,j} - S_{expt,j,i})^2}{\sum_{i=1}^n (S_{expt,j,i} - \bar{S}_{expt,j})^2}$$

where N is the number of snapshots, n is the number of data points for snapshot j, $S_{pred,j}$ is the prediction of the model for the j snapshot, $S_{expt,j,i}$ is the observed value of the i data point of snapshot j and $\bar{S}_{expt,j}$ is the average of the observed data points of snapshot j. The numerator represents the standard MSE and the denominator represents the normalisation by the variance at each point. We note that the minimum value is achieved when $S_{pred,j} = \bar{S}_{expt,j}$, which leads to $NMSE = 1$. Therefore, the closer the value is to 1, the better the fit to the mean of the data at each snapshot, i.e., the fit follows the mean dynamics.

Additionally, we introduce a second fit quality measure: a variation of the commonly used R^2 metric, which measures the fitness of a linear fit. Here, we adapted this by using a weighted R^2 value (R_w^2), where we compare the model predictions against the mean of the data at each of the snapshots, weighted by the inverse of the variance at that point. This means that points with higher variance are less critical to the metric. Finally, we normalise this value against the total variation of the dataset, i.e., the mean of each observed snapshot against the mean of all data points. Mathematically, we describe this as:

$$R_w^2 = 1 - \frac{\sum_{i=1}^N w_i (S_{pred,i} - \bar{S}_{expt,i})^2}{\sum_{i=1}^N w_i (\bar{S}_{expt,i} - \bar{S}_{expt})^2}$$

By incorporating variance as a weighting factor, this aims to account for the heteroscedasticity (varying spread) of the data across different time points. A higher value of R_w^2 (closer to 1) indicates a better fit, considering both the model's predictive accuracy and the inherent variability of the data. More concretely, the R_w^2 measures how much better our model performs over a model with a constant horizontal line which takes the mean of the experimental data as its height.

- In the 7-spines experiment, the authors say that they predict and find differences in plasticity when the synapses are more distributed spatially than in the control clustered condition. However, no statistical comparison was made between the control and distributed 7-spine conditions. The only analysis presented is the lack of a significant difference between distributed 7 spines and single spine data, which does not directly address the original question.

We thank the reviewer for this comment suggesting adding more statistical evaluation to describe the experimental scenario with 7 spatially distributed spines. We have addressed this by revising the original figure to include the clustered seven spine condition and performed statistical tests to determine whether a significant difference in the dynamics was present. We found that four time points post-stimulation show significance, as predicted by the model. In addition to revising the corresponding figure we have added sentences describing this comparative analysis to the results section.

Caption excerpt: h) Normalised spine changes of stimulated spines for the seven spine (dark blue), distributed seven spines (orange) single spine (light blue, dashed) and experiments. As predicted by the model, the increased spatial distance between spines leads to a recovery of the single-spine dynamics. $p < 0.05$, two-sided t-test between clustered and distributed seven spine dynamics. No significant difference between the distributed and single spine dynamics was found. Error bars represent the standard error of the mean. $N = 68$ distributed spines.

“As predicted by our model, no significant difference in the potentiation for seven distributed spines was observed when compared against the single spine stimulation. Additionally, we found that four time points post-stimulation showed significance when comparing the distributed and clustered seven spine dynamics, which is in line with our model predictions. This emphasises that the spatial arrangement is critical for the plasticity outcome.”

- In several cases paired t-tests were performed across multiple data groups (for example Fig 3e-f, Fig 5g). Appropriate statistical tests or adjustment need to be applied to correct for the multiple comparisons.

We are grateful to the reviewer for pointing out this potentially confusing point. In the figures, we performed a binary comparison, i.e., a studentized t-test for each pair of experiments. We agree that our notation (a bar starting and ending at the two experiments to be compared) may be confusing. Therefore, we have updated the annotation to include two small lines that extend downward to indicate the scenarios compared. We hope that this clarifies the nature of our statistical analysis. Examples (with the new annotative lines) can be seen below:

- The numbers of experiments should be indicated throughout the paper, and exact p values reported for all statistics.

We thank the reviewer for this comment, which is in line with the other reviewers' suggestions. We have now added the N number to each experiment in Supplemental Table 1 and reported p values for each paired t-test in each figure. Additionally, we have added a sentence indicating these changes:

“Details on the N numbers of animals, experiments, stimulated spines for each experimental paradigm are shown in table 1.”

2) The experiments were done in 0 mM Mg²⁺ and in most cases multiple spines were stimulated quasi-simultaneously. It is reasonable to expect that such stimulation produces strong summated dendritic depolarisation, activates voltage dependent Ca²⁺ and other channels and possibly release from intracellular stores, and overall induces large dendritic Ca²⁺ accumulation. This biological state differs from that in the model, where synapses are individual plasticity points that do not interact other than through biochemical mechanisms. Have the authors measured the electrical and Ca²⁺ responses to the LTP stimulation, and considered to take into account the impact of excess dendritic Ca²⁺?

We thank the reviewer for these important comments. We did not directly measure electrical responses to multi-spine uncaging, but instead performed calcium imaging experiments. To address the question of calcium dynamics during stimulation we overexpressed GCaMP6s, along with DsRED as a cell-fill, and imaged calcium dynamics during the uncaging pulse train. Below shows summary plots of the spine and dendritic calcium transients for the 1, 3, 7, and 15 spine experiments during uncaging (yellow bar). The magnitude of spine and dendritic transients increases as more spines are added to the cluster, and calcium levels quickly return back to baseline within seconds following the cessation of uncaging, reminiscent of the rapid degradation of component C in the model. Collectively, it is likely that the effects of excess intracellular dendritic calcium observed during stimulation would have translated into calcium-dependent biochemical signalling processes by 2 min post stimulation when the peak potentiation occurs. The present work aims to provide an initial framework for explaining heterosynaptic interactions through a simple mathematical model that is in alignment with experimental data. We acknowledge that we cannot rule out the contribution of response amplification by the engagement of voltage signalling in the present work. Furthermore, there may also be a contribution of extracellular signalling. These will be key points to explore in a future study. Given that spine structural plasticity is manifested as changes involving reorganisation of postsynaptic actin cytoskeleton, scaffolds and AMPA receptors, which in turn rely on biochemical signalling, we believe that a model based on abstract P and C terms using experiments in 0 mM Mg²⁺ is nonetheless beneficial over the slow, minutes to tens of minutes time scales that are addressed in this study. As a general point, using 0 mM Mg²⁺ solution is a common approach to allow the induction of spine structural plasticity via glutamate uncaging (for example, Matsuzaki et al., 2004, Govindarajan et al., 2006, Oh et al., 2015), and has been used to induce structural plasticity in vivo where the surface of the cortex was perfused with a solution containing caged glutamate in 0 mM Mg²⁺ (Noguchi et al. 2019), demonstrating the usefulness of this approach to the community. We have added to the revised manuscript the following supplemental figure that illustrates the calcium transients in our dataset along with an additional paragraph in the methods section that describes how these images were acquired and analysed.

Figure S2: Imaging of GCaMP6s in spines and dendrites during uncaging. Dendrites from neurons expressing GCaMP6s and DsRed were targeted for simultaneous imaging and glutamate uncaging. **a)** Example images illustrating the spatial spread of calcium during uncaging. Green, GCaMP6s, magenta, DsRed. Scale bar = 5 μm . **b)** Spine and **c)** dendritic calcium dynamics during uncaging. Repetitive glutamate uncaging (yellow bar) elicits a sustained spine and dendritic calcium elevation during stimulation, which returns to baseline levels within seconds of the cessation of uncaging. Measuring calcium elevations along the dendritic shaft demonstrates that calcium rises are spatially confined close to targeted spines, and that peak calcium occurs in the centre of the stimulated spine cluster. **d)** Measured average spatial profile of calcium during uncaging stimulus. All individual experiments were aligned to the stimulus spine (for the 1x spine condition) or to the centre of the stimulated cluster (for the 3x, 7x, 15x, and the distributed 7x conditions). N's (cells); 1x = 8, 3x = 9, 7x = 11, 15x = 5, 7x distributed = 7.

Additionally, we have added the following methods section to the supplement:

"Calcium imaging

To obtain the results of Fig. S2 neurons were biolistically transfected with GCaMP6s and DsRed, and used for experiments 48-72 hours later. Both GCaMP6s and DsRed were excited at 910 nm. Uncaging conditions were identical to the other experiments in the manuscript (i.e. 60 pulses at 1 Hz), except dendrites were simultaneously imaged at 1 Hz throughout the stimulus, including a short (5 second) baseline and chase period. To measure spine and dendritic calcium dynamics during glutamate uncaging, ROIs were positioned over the targeted spines, and the dendrite directly below the targeted spine. For clusters of spines, the dendritic ROI was positioned in the centre of the cluster. GCaMP6s intensity was measured using Fiji (Schindelin et al., 2012). To measure dendritic spatial calcium dynamics, a line profile was drawn along the dendrite and expanded to fill the internal width of the dendrite. This was used to measure both the GCaMP and the DsRed signal intensity along the dendrite. The GCaMP signal was normalised to the DsRed signal for the entire recording, then the average during the stimulus was calculated (excluding baseline and chase period), and finally this was normalised to the baseline. All individual dendritic profiles were then aligned to the stimulated spine, or in the case of a cluster, the centre of the cluster."

3) The model predicts mostly modest differences in sLTP (for example, ~10-15% difference for middle and edge spines), which requires that the experimental method is able to reliably reveal small effects. However, glutamate uncaging is difficult to standardise to such extent across spines and experiments (location of the uncaging point around and away from the spine, laser power at different depths), and it is unclear from the manuscript what attempts have been made to do so. Another important point is that the relative volume change from baseline during sLTP depends on the initial spine head size (Matsuzaki et al 2004 Nature, and others). Therefore the only way to appropriately determine the effects of cluster size or spine position on sLTP strength would be to select spines with very similar initial head sizes, which does not seem to have been the case here.

Thank you for this comment. In fact, we performed experiments to validate the specificity of our uncaging, and it was our oversight in not having included this in the original manuscript.

To elicit plasticity the stimulating laser was positioned 0.5 μm from the spine head, taking care to only choose spines that were in focus in the z plane. Dendrites used for experiments were consistently near the slice surface. In an initial round of experiments, we combined whole-cell patch clamp recordings with glutamate uncaging, and set the uncaging laser power to elicit uncaging evoked post-synaptic currents (uEPSCs) that matched the amplitude of spontaneous miniature EPSCs (mEPSCs). With regards to the specificity of the uncaging stimulus, we did not see structural plasticity if a) we moved the uncaging laser away from the spine head (to 2 μm), b) changed the wavelength of the uncaging laser (720 nm to 880 nm), or c) uncaged in the presence of APV in the aCSF. Moreover, our single-spine stimulation elicited structural plasticity only in the targeted spine, while leaving nearby neighbouring spines unchanged, demonstrating the spatial specificity of our stimulation. This is comparable to other published studies using this technique (for example, Matsuzaki et al., 2004). Please see below for example figures. We have added additional text to the methods section to better explain our approach.

Figure - Controls for the specificity of glutamate uncaging. Neurons expressing EGFP were targeted for uncaging. a) glutamate uncaging 0.5 μm from the spine head results in robust sLTP (control, black trace, $n = 7$), but not if NMDARs are blocked (APV, red trace, $n = 7$), the uncaging laser wavelength is changed (880 nm laser, blue trace, $n = 5$), or if the uncaging spot is moved away from the spine head (laser shift, green trace, $n = 6$). b), example images from an experiment where the uncaging laser was positioned 2 μm away from the spine head (white arrowhead, target spine; red dot, uncaging location). Under these conditions, no sLTP is observed. Scale bar = 5 μm . c) Changes in neighbouring spine size following sLTP induction of a single target spine (Data from 10 neurons, 299 neighbouring spines).

Additional methods

“Uncaging laser power was set so that uncaging evoked excitatory post-synaptic currents (uEPSCs) matched endogenous spontaneous miniature EPSCs (mEPSCs), as measured by whole-cell patch clamp. We did not see structural plasticity if we moved the uncaging laser away from the spine head (to 2 μm), changed the wavelength of the uncaging laser (720 nm to 880 nm), or uncaged in the presence of APV in the aCSF.”

We thank the reviewer for the insightful comment about the initial spine head size that allowed us to clarify our analysis. To study the total growth in the spine size, the reviewer correctly points out that if the unnormalised spine size can be significantly different pre-stimulation, that the total growth post stimulation may be impacted unintentionally. To make sure that this was not the case, we analysed the full distribution of unnormalised spine sizes before stimulation as follows. During the experiments, we attempted to select spines of similar sizes (medium sizes spines). Below we see the distribution of spine sizes for each of the control experiments:

We note that the 1, 3 and 7 spines do not differ significantly in their mean. However, the 15 spine experiments do exhibit a wider range of sizes and also a slightly higher mean size (albeit within the error bar of the 1 spine experiment). The variability in the variance is a natural result of the design of the experiment: a need to select 15 relatively clustered spines considerably limits the

freedom in choosing the spine size as we require a dendritic branch carrying 15 spines showing a confined spatial distribution and that are also in the same z plane of the microscope.

Despite the limitations, we note that the percentage of spines in the 7 and 15 spine experiments whose sizes fall in the range of the three spine experiment (i.e., the number of spines that are bigger than the smallest spine in the 3 spine experiment and smaller than the largest spine in the 3 spine experiment) is 86.7% percent and 88.3% percent, respectively.

We hope that this demonstrates that the unnormalised spine sizes are within similar ranges allowing us to then compare the total growth post stimulation.

4) The configuration of the experimentally stimulated spines does not correspond well to that used in the model.

- Fig 2a: two of the three stimulated spines emanate from the same dendritic location. Which of these spines would be 'middle' and 'edge'?

- In Fig 5 the stimulated spines are unevenly distributed forming two large groups of several small clusters of very close spines. In this scenario, can the spines be considered as a single cluster of spines competing for the same resources?

Thank you for this helpful comment that Fig 2a in its previous form may be confusing. In the example image of original Fig 2a, although the two spines appear to originate very close to one another on the dendrite, the raw z-stack data shows that the lower spine is sprouting from the dendrite further along than the upper spine, and is therefore the edge spine. Therefore, we have replaced the image with a clearer example of a spine cluster bounded with edge spines.

Fig 5: This is an interesting point, and we thank the reviewer for raising it. It is experimentally challenging to find and stimulate precisely the same arrangement of spines (starting size, inter-spine distance etc.) across multiple experiments, especially in the 15 spine condition. Throughout this current study the spines are stimulated quasi-simultaneously, and thus form a single group in a temporal sense. Spatially, they share the same dendrite, and are within a few microns of one another. Examining the entire dataset demonstrates that this "double" cluster is not the typical spatial arrangement, and to reduce confusion we have replaced the example image in the manuscript with a more representative one. Furthermore, in the revised manuscript, we include a new analysis that takes into account minimal and maximal cluster size within a group and show that the significant differences beyond the cluster size variations (attributed to spine-to-spine distance variations) emerge when comparing edge and middle spines within a cluster for 15 spine experiments that is consistent with competition for resources (new Fig 5f).

To show the overall distribution of all the stimulated spines, in all the clusters, we measured the interspine distance between every pair of spines within a cluster. This is shown below, with the pooled data for all clusters plotted under "all" on the far right in grey. We hope this illustrates that our stimulated clusters tend to be distributed along the dendrite, and do not form two or more "sub-clusters".

5) The image in Fig 2b shows severe swelling and distortion of the stimulated spines and the dendrite at 2 min after uncaging, which seems out of the normal. Other than the independent sham experiments, can the authors exclude that their stimulation conditions caused unphysiological side effects affecting spine size in some experiments?

Thank you for this comment, and for bringing up the extremely important point of potential damage/phototoxicity to the cells during imaging and uncaging. Any experiments that did show prolonged dendritic swelling/blebbing were not included in the analysis, and were quite rare (less than 5%) in the total dataset.

With regards to the referenced image, please see the figure below of individual image slices of the z-stack shown in Fig 2b. What appears to be dendritic swelling in the maximum intensity projection is actually a pair of spines (a and b, indicated by arrows in the image) that are projecting in the z direction. These spines underwent a transient enlargement, but returned to their starting size shortly after. In order to avoid potential confusion in interpreting the strong fluorescence signals we have replaced the example image with another.

6) The model seems to be quite complicated to the details, which stands in contrast with the lack of clearly plausible biological counterparts with properties similar to those of the proteins designed in the model. One cannot help but wonder that other combinations of mechanisms and features, not based on competition for resources, might be just as suitable to explain the experimental observations.

We thank the reviewer for this comment, which gave us the opportunity to articulate our thoughts with respect to the balance between the complex biological reality and the need for abstraction when constructing models to gain insights into biological phenomena. The mechanisms underlying the coordination of synaptic plasticity across multiple synapses have been and continue to be a hotly debated topic with many potential candidate processes and molecules. Even a single spine alone is capable of supporting a multitude of biochemical signalling cascades (e.g. Nakahata and Yasuda, Front Synaptic Neurosci 2018), and thus considerations of interactions between groups of synapses, along with active dendritic mechanisms add additional complexity. We agree that there may be other “combinations of mechanisms and features” that can explain our results besides competition for resources represented by P and C. Additionally, we acknowledge the reviewer’s concerns regarding the seeming complexity of our model and questions about whether and how direct biological counterparts for the two components could be identified. In fact, it is these challenges posed by the vast possibilities of mechanisms underlying multi-spine plasticity that have prompted us to take a reductionist approach in considering a conceptually simple model that relies on just two dynamic components, P and C. Crucially, we have generated

experimental data by varying conditions in a highly controlled manner by eliciting multi-spine plasticity along single dendritic branches at a variable number of spines using the same induction protocol. Therefore, whereas all individual stimulated spines would have experienced the same initial triggering activity, depending on the number of stimulated spines, the effective change in dendritic activity will differ and, in turn, likely engage signalling events to different degrees. Despite the variable conditions, the finding that our simple model can account for multi-spine plasticity from 3 spine paradigm to 15 spine paradigm is striking, and this gives us a framework for further dissecting the underlying mechanisms represented by P and C.

Sharing or competing for resources is a general biological concept motivated by the finite amount of resources available. Although we have posited P and C to be calcium-dependent factors that can quickly dissipate and/or diffuse, such as small molecules or proteins that can span multiple spines, the idea of competition for resources can involve much larger intracellular organelles and polyribosomes, as supported by ultrastructural studies (cf. Harris, *Curr Opin Neurobiol* 2020). Moreover, we do not exclude other mechanisms, such as various forms of synaptic tagging (e.g. Okuno et al., *Cell* 2012) that differentiate certain spines from others according to activity history or the contribution of extracellular sources such as synapse modulation by glial cells. Nevertheless, NMDA receptor-dependent triggering of structural spine plasticity is prompted by a rapid intracellular rise in calcium. A model inspired by calcium-dependent signalling cascades that could be either shared by or competed for amongst neighbouring spines offers a base for future studies to build upon and explore additional mechanisms beyond those we have proposed rather than intending to provide definitive answers to all questions surrounding interactions governing spine dynamics.

Additional references

Harris KM. Structural LTP: from synaptogenesis to regulated synapse enlargement and clustering. *Curr Opin Neurobiol*. 2020 Aug;63:189-197. doi: 10.1016/j.conb.2020.04.009. Epub 2020 Jul 10. PMID: 32659458; PMCID: PMC7484443.

Nakahata Y, Yasuda R. Plasticity of Spine Structure: Local Signaling, Translation and Cytoskeletal Reorganization. *Front Synaptic Neurosci*. 2018 Aug 29;10:29. doi: 10.3389/fnsyn.2018.00029. PMID: 30210329; PMCID: PMC6123351.

Okuno, H., Akashi, K., Ishii, Y., Yagishita-Kyo, N., Suzuki, K., Nonaka, M., Kawashima, T., Fujii, H., Takemoto-Kimura, S., Abe, M., Natsume, R., Chowdhury, S., Sakimura, K., Worley, P. F., & Bito, H. (2012). Inverse synaptic tagging of inactive synapses via dynamic interaction of Arc/Arg3.1 with CaMKII β . *Cell*, 149(4), 886–898.

Noguchi, J., Nagaoka, A., Hayama, T. and Ucar, H., Yagishita, S. and Takahashi, N. and Kasai, H. (2019), Stringent structural plasticity of dendritic spines revealed by two-photon glutamate uncaging in adult mouse neocortex in vivo, *bioRxiv*, 577742

Minor comment:

7) Fig 3a-c caption says to show top and bottom rows with control and sham experiments, but only one row is displayed.

We are grateful to the reviewer for spotting this. The words “bottom row sham” originated from an early version of the figure and have therefore removed the words “bottom row sham” in the updated manuscript.

REVIEWER COMMENTS

Reviewer #1 (Remarks to the Author):

In the revised manuscript, the authors addressed my primary concerns by providing more substantial statistical support for their conclusions. Additionally, the revised manuscript presents the results with greater clarity and discusses their relevance to in vivo studies, which will allow the work to reach a broader audience. The manuscript is ready for publication.

Reviewer #1 (Remarks on code availability):

I did not have time to install and run the code, but from a brief skim of the code it looks very straightforward. All code creating figures and necessary dependent functions appear to be provided.

Reviewer #2 (Remarks to the Author):

The authors have made appropriate reanalysis and modifications in response to my comments. I would favor the publication of the manuscript after correcting several minor points listed below.

1. Figure 3e and f. For comparisons of four groups, it is advisable to use ANOVA and post hoc comparisons as a statistical test instead of repeated t-tests.
2. Main text page 8, Figure 2 legend, last line. “p-values for edge and middle spines are ~ 0.33 and 0.017, respectively,” does not match the data. The order of p-values may be reversed.
3. Main text page 9, line 16. “This is analogous to lessening the contribution of P in our model” may not be correct. The FK506 effect should be the opposite of the CaMKII inhibitor.
4. Main text page 13, line 2. “Thus far, the experimental results the experimental results” may be a typing mistake.
5. Main text page 13, line 23. “(Fig. 5g-i)” should be Fig. 5g-h.
6. Main text, page 13, line 24, “see grey horizontal dashed line,” is not clear. Maybe this is related to Figure 1g.
7. Supplemental Information Figure S3. Symbols are too small to recognize. Change the size and use more colors.

Reviewer #3 (Remarks to the Author):

While I do appreciate the work the authors have put into the revision, several of my major concerns remained incompletely or not addressed. The new data in the revised ms/rebuttal raise additional questions.

To be clear, I think that both the modelling and the experimental part of the paper are valuable on their own. The problem is that the claimed match between the results of the two approaches has not been convincingly demonstrated. Therefore, most of the main conclusions do not stand on sufficiently solid grounds.

General critical points:

- the correspondence between the measured vs modelled differences in spine dynamics is overall weak. This is not surprising due to the low sampling rate and large variability in the experiments compared to the predicted kinetics and effect sizes. Particularly, the temporal resolution of the experiments is not sufficient to accurately monitor the kinetics and magnitude of spine size changes, especially in the first 10 minutes after the stimulation, when the largest differences are predicted.
- since there is no difference between middle and edge spines in the 3-spine cluster scenario and the evidence for such an effect is statistically obscure even for the 7-spine clusters, the relevance of the proposed middle-edge competition seems to be limited to very dense, physiologically implausible clusters (15 coactive spines on 30 microns).

Detailed comments, according to the original points:

1)

- 3-spine experiments: In contrast to the previous version, the revised manuscript finds no difference between middle and edge spine volume changes in 3-spine clusters. I should say I was perplexed that, although the modelling result has been flipped to the opposite direction, the authors continue to claim that the experiments match the model. Namely, the current prediction (no difference) is supposed to be validated by the very same data that, in the first version of the paper, have been claimed to confirm a significant difference. (The legend of the figure describing the experiment has been simply changed from “a difference can be observed” to “no difference can

be observed”, still without any direct statistical comparison reported). This is hard to accept as rigorous fact-based evaluation of the data.

In addition, the text is confusing about these results. It begins with: “Our model predicts that spines on the edge of a cluster should potentiate more strongly than those in the middle.”, then “the model predicts that edge and middle spines would exhibit a similar plasticity response. This matches our experimental data. ...no significant differences between these two spine types can be expected in this experimental scenario.” Nonetheless, then it proceeds with: “So far, we have demonstrated, both experimentally and within our model, that i) spines within a cluster compete, as measured by their ability to enlarge following stimulation...”

Has the text perhaps not been completely updated?

- Single-spine experiments: my original concern remains, i.e. that the experiments do not have sufficient temporal resolution to convincingly evaluate the model’s prediction about the fast initial kinetics or the maximum of the spine volume change in the first 10 minutes. (This also holds for the 7-spine distributed data, and affects the single-spine growth data in Fig. 6g.)

The authors rather tried to gain more insight into the decay by new exponential decay fits (shown in Fig.6), but this approach appears limited. From what is presented it seems that no statistical evidence was found for a different decay of single spines from that in the other groups. The individual spine decay parameters are not significantly different between any of the groups according to the p values in Fig.6e (contrary to the description in the text). It is mentioned that spines with erroneous fits were omitted, but how many spines remained in the different groups is not specified.

- 7-spine experiments: I still find it problematic that the claimed difference between middle and edge spines is based on an odd single data point at 20 minutes. By 30 minutes we don’t see difference in the data, although the model predicts that the two spine groups should be distinct. In addition, it is not clear whether a difference at 20 minutes remains significant after correction for multiple comparisons (see below).

There is a relatively wide range of different cluster lengths reported in this experiment, from 6 to 13 micrometers. Is there a correlation between predicted and experimentally measured differences for middle and edge spines?

Overall, as the comparative summary plots in Fig 6a-b highlight, the match between the measured vs modelled differences in spine dynamics is generally rather low, except for the 15-spine scenario.

- Statistics: The cosmetic modifications in the revision did not address the problem of multiple uncorrected comparisons (also raised by Reviewer 1). Multiple paired comparisons accumulate statistical errors, which need to be adjusted by appropriate corrections (e.g. Bonferroni or other). The authors need to clarify how they handle this, if they do. This concerns not only the comparisons of different cluster arrangements, but also the multiple comparisons between middle and edge spines at different time points, where weak significant differences are reported by paired tests (e.g. Fig. 4f, Fig. 5f).

- Some of the rows in Table 1 contain lower number of stimulated spines per experiment than the expected set (e.g 7-spine, 7-spine distributed, 15-spine, 15-spine sham). What is the explanation for this discrepancy?

2) The new Ca^{2+} measurements show large, homogeneous, nonlinear increases in the spine and dendrite Ca^{2+} signals by clustered multi-spine stimulations, indicating a qualitatively different response and activation of additional Ca^{2+} sources compared to single-spine stimulation, and drastically changing the dynamics and dendritic distribution of Ca^{2+} from that assumed in the model (Fig. 1g). These results therefore warn against simplifying the mechanism that determines plasticity to competition for two components originating from individual spine activations.

3) The controls for glutamate uncaging are a useful addition. It would be nice to see whether the uncaging stimulus generates similar spine responses at the middle vs the edge of the the field of view.

However, my concern about variable initial spine sizes affecting the plasticity outcome has not been completely resolved. The distributions of initial spine sizes in the multi-spine experiments are very wide and are non-normal (the mean values are not very informative). Importantly, the larger initial spine size in the 15-spine experiments may contribute to why the normalized peak spine increase was numerically lower than in the other experiments, questioning whether this difference is biological.

5) The z-stack of the original 3-spine experiment indeed hints that spines projecting in the z-plane could be responsible for the enormous swelling. However, this illustrates an additional issue. The enlarged spines were not targeted for uncaging, but they were very close to the targeted spines, likely within the 3D volume of the glutamate stimulus. This contradicts the authors' argument in the rebuttal that "the single-spine stimulation elicited structural plasticity only in the targeted spine, while leaving nearby neighbouring spines unchanged". It rather suggests that the stimulus is often not restricted to single synapses, meaning that the real number of stimulated spines in the experiments is not known.

Given the above issues, the manuscript contains many exaggerated statements about the results (examples: “we confirm one of the model’s prediction, that single spines rapidly grow (as there is no competition for C) but then equally rapidly shrink back to a lower value”, “...our model... could accurately describe the experimentally generated plasticity and its time course across different numbers of stimulated spines.”; “the model captures the final saturation rate of each experiment, underscoring the idea that the number of spines stimulated fundamentally affects the increase in the size of the spines reached over the long-term”, etc) The authors need to tone down their interpretations and make only well supported conclusions.

The Abstract includes some inaccuracies.

“Potentiation of clusters of dendritic spines lead to changes that are dependent on NMDAR, CaMKII and calcineurin.” This has only been shown for single spines.

‘our results provide a quantitative description of the multi-spine stimulation footprint over minutes and hours post-stimulation’. Hours have not been investigated.

Reviewer #2 (Remarks to the Author):

The authors have made appropriate reanalysis and modifications in response to my comments. I would favor the publication of the manuscript after correcting several minor points listed below.

1. Figure 3e and f. For comparisons of four groups, it is advisable to use ANOVA and post hoc comparisons as a statistical test instead of repeated t-tests.

We thank the reviewer for this comment. As the reviewer suggested, we tested for significance by performing a Kruskal-Wallis test followed by Dunn's multiple comparisons with the false discovery rate (FDR) correction factor following the Benjamini-Hochberg procedure. We find that spine volume increase in control, and FK506 remains significant compared to sham, whereas the increase in the presence of AIP, a CaMKII inhibitor, is largely attenuated and not significant compared to sham, highlighting the involvement of CaMKII in spine potentiation. We have added the following clarification to the statistical methods section:

To test for significance, first, a Kruskal-Wallis test was performed, followed by Dunn's multiple comparison with the false discovery rate (FDR) correction factor.

2. Main text page 8, Figure 2 legend, last line. "p-values for edge and middle spines are ~ 0.33 and 0.017, respectively," does not match the data. The order of p-values may be reversed.

We have now updated the p-values to refer to the correct comparisons.

3. Main text page 9, line 16. “This is analogous to lessening the contribution of P in our model” may not be correct. The FK506 effect should be the opposite of the CaMKII inhibitor.

We are grateful to the reviewer for the careful reading of our manuscript. We apologise for any confusion caused by an error in our text. To clarify, FK506 (calcineurin inhibition), which promotes more growth and weaker depression, is modelled by a lower effect of P in the model, while AIP (which blocks CaMKII) is instead responsible for an enhanced effect of P. We have now amended this paragraph as follows:

Our model suggests that single spines, in the absence of competition imposed by near simultaneous activation of other nearby spines, will exhibit dynamics that are dominated by rapid growth followed by a similarly rapid decline, distinct from the dynamics of multi-spine plasticity. To explore this point, we next elicited sLTP by targeting only single spines. Glutamate uncaging of single spines resulted in robust sLTP (Fig. 3a, top row, pooled data plotted in Fig. 3d, blue line). To gain further insight into the underlying mechanisms, we turned to calcium and calmodulin-dependent enzymes, CaMKII and calcineurin, which previously have been shown to regulate LTP and LTD, respectively. In one set of experiments, single spine glutamate uncaging was performed in the presence of a specific competitive inhibitor of CaMKII, myristoylated autocamtide-inhibitory peptide (AIP, 5 μ M; Ishida et al., 1995) to test if inhibiting CaMKII produced an effect that was similar to enhancing the effect of P. In a separate set of experiments, we inhibited calcineurin using FK506 (2 μ M; Dumont, 2000). As calcineurin activity was previously reported to promote sLTD, inhibiting the action of calcineurin could mimic the effect of lessening the contribution of P in our model.

4. Main text page 13, line 2. “Thus far, the experimental results the experimental results” may be a typing mistake.

We thank the reviewer for noticing this error. The second “experimental results” has been removed.

5. Main text page 13, line 23. “(Fig. 5g-i)” should be Fig. 5g-h.

This has now been corrected.

6. Main text, page 13, line 24, “see grey horizontal dashed line,” is not clear. Maybe this is related to Figure 1g.

The grey horizontal line was a reference to a previous version of this figure. We have amended the sentence to now read:

[...] see positive values for the difference between C at edge and middle in Fig 5g.

7. Supplemental Information Figure S3. Symbols are too small to recognize. Change the size and use more colors.

We agree with the reviewer that the previous version of the figure was not clear enough due to the similarities of colours and size of the markers. We have updated this and hope this addresses the reviewers concerns:

Reviewer #3 (Remarks to the Author):

While I do appreciate the work the authors have put into the revision, several of my major concerns remained incompletely or not addressed. The new data in the revised ms/rebuttal raise additional questions.

To be clear, I think that both the modelling and the experimental part of the paper are valuable on their own. The problem is that the claimed match between the results of the two approaches has not been convincingly demonstrated. Therefore, most of the main conclusions do not stand on sufficiently solid grounds.

General critical points:

- the correspondence between the measured vs modelled differences in spine dynamics is overall weak. This is not surprising due to the low sampling rate and large variability in the experiments compared to the predicted kinetics and effect sizes. Particularly, the temporal resolution of the experiments is not sufficient to accurately monitor the kinetics and magnitude of spine size changes, especially in the first 10 minutes after the stimulation, when the largest differences are predicted.

Thank you for raising the point about the temporal resolution. Our experimental measurements were taken at 2, 10, 20, 30 and 40 min since our goal was to measure and quantify the experimental spine dynamics with the model prediction over a relatively slower time scale. This is motivated by the fact that less attention has been given to the later phase of plasticity in

previous models. Moreover, there is a practical limitation for imaging the initial fast spine dynamics in the same experiment at high temporal resolution while still aiming to capture slow spine dynamics over a longer time frame. Excessive sampling leads to photobleaching and compromised cell health, and therefore, reliable long-term recordings have been obtained at the expense of high temporal resolution recording of earlier time points. In line with the reviewer’s comment, we have reworded the text throughout the manuscript to reflect that the presented analyses focus on the 2-40 min time window post-stimulation. To address the reviewer’s concerns regarding the low sampling rate and the large variability, we have analysed the following additional data sets:

(i) short-term, higher-frequency recordings of spines for 1-spine (N = 21 experiments) and 7-spine (N = 10 experiments) paradigms during and after stimulation up to 2 min following stimulation onset.

(ii) 1-spine (N = 6) and 7-spine stimulation experiments (N = 13) that we have performed for a separate but related project.

For (i), we present the results below to the reviewer. For (ii) on 7-spine stimulation experiments, we have included the data in the present manuscript, which has helped reduce the variability of the 7-spine dataset (please see below; cf. Figure 4).

To monitor spine dynamics during stimulation and immediately after stimulation, in the 1-spine paradigm, images were captured every 5 sec during uncaging, followed by imaging at 5 sec and at 1 min time points post-uncaging. In the 7-spine paradigm, stimulated spines were imaged during and after stimulation every 5 sec until 1 min after stimulation. We then took the model prediction from Figure 4 (solid blue line) and compared it to the experimentally observed spine dynamics (no fitting or other changes in the model parameters).

Caption: Stimulation and early post-stimulation dynamics of the single spine example

Caption: Stimulation and early post-stimulation dynamics of the 7 spine example

With respect to the model prediction for the post-stimulation phase (green region), we find that the model dynamics underestimate the spine size increase immediately after stimulation for the 1-spine paradigm, but can capture the general rise and fall dynamics of the data. Interestingly, we find that in the 7-spine paradigm, the predicted and measured spine dynamics are in agreement with one another in the green region. Here, we emphasise that the parameters for the model were derived from the 3-spine paradigm obtained in the time frame of 2-40 mins, and no additional parameter fitting of any kind was performed (this holds throughout the manuscript for 1-spine, 7-spine, and 15-spine stimulation scenarios). This suggests that the model could suitably capture the plasticity dynamics, at least in the multi-spine paradigms from tens of seconds to tens of minutes.

Regarding the plots of the model prediction, in the previous version of the manuscript, a continuous theory line was shown throughout, including the stimulation time frame (0-1 min; corresponding to the grey region containing the blue dotted line in the figures above). Our model was designed to account for spine dynamics after completion of the stimulation and not during the stimulation. Therefore, to reflect this point and avoid confusion, we have updated all the figures that include model dynamics so that the evolution of the spine volume in this first minute is depicted with a thin dashed line. Additionally, we state that the model in this part of the experiment is presented for completeness but should not be taken as a prediction. The two example figures below (Fig. 4i and Fig. 5d) illustrate how the updated figures in the manuscript appear now:

Caption: The predicted dynamics of the distributed seven-spine case (orange line) and the single spine case (dashed blue line) are similar. Error bars represent the standard error of the mean. Thinner dashed gray lines at $t \approx 0$ represent the stimulation region where the model predictions have limited validity.

Caption: Changes in normalised spine volume relative to the baseline of the stimulated spines are shown (grey points). By reducing the amount of initially available C and P according to the model equations (no refitting), the model can capture the experimentally measured temporal dynamics. Further analysis of the data at time points in the grey boxes marked with (†, ‡) is shown in e). Thin dashed lines refer to the stimulation region which is outside of the temporal range of model validity.

With these experimental cross checks on the fast time scales (0-2 min) in mind, we stress that our model is primarily designed to capture the time scales starting from 2 min post stimulation to 40 min post-stimulation. Following the reviewer's advice, we now emphasise this temporal validity (2-40 min) of the model and do not make statements about model validity for faster timescales in the re-revised manuscript.

- since there is no difference between middle and edge spines in the 3-spine cluster scenario and the evidence for such an effect is statistically obscure even for the 7-spine clusters, the relevance of the proposed middle-edge competition seems to be limited to very dense, physiologically implausible clusters (15 coactive spines on 30 microns).

We are grateful to the reviewer for giving us the opportunity to strengthen the data vs model comparison for the 7-spine experiment with respect to the edge and middle difference. While our model could capture the average dynamics of the stimulated spines (Fig. 4d), the predicted distinction between edge vs middle spines showed considerable variability. To address this point, we included additional experimental data in the 7-spine paradigm and reinvestigated the edge vs middle differences. We discuss this point below (in the reply text to the more detailed referee comment starting with “- 7-spine experiments:”) and present updated figures where, with additional data, the edge vs middle spine dynamics display significant differences at 30 min and 40 min post-stimulation (albeit losing the significance at the +20 min time point) and that the edge spine response is larger than the middle spine response, as predicted by the model.

Concerning the second part of the reviewer's remark questioning the physiological likelihood of the occurrence of 7 to 15 co-active spines over short stretches of dendrite, there are reports in the literature that lend support to such a scenario. Combined glutamate uncaging and electrophysiological recording suggest that it takes 20 co-active synapses to trigger a local dendritic spike on CA1 radial oblique dendrites (Losonczy and Magee, 2006), the same cell type and compartment we performed our experiments on. Given the physiological nature of local dendritic spikes, the finding suggests that the numbers in our study are not entirely implausible. There is also evidence that for some connections, a single axon can make two or more synapses locally along the same dendritic stretch in CA1 distal inputs (Bloss et al., 2018) and in human cortical neurons (Shapson-Coe et al., 2024). Notably, in the latter study, 39% of highly innervated neurons reconstructed in the EM volume had at least one axonal input that made seven or more synapses along a short dendritic segment, representing a powerful connection. In addition, spine clusters during decision-making in vivo (Kerlin et al., 2019) show a spatial activity correlation of $13 \pm 6 \mu\text{m}$ for cortical pyramidal cells. Based on spine densities that typically range from 1 to 2 spine per μm dendritic length for cortical neurons (Parajuli et al., 2020), the synaptic cluster sizes of 7 and 15 spines which we chose in our study are within the range of co-activated spine clusters reported in vivo. We appreciate that under healthy physiological conditions, cases in which 15 spines on a single dendrite are all simultaneously active may be rare, while 15-spine stimulation experiments have been useful in exploring the validity of our model predictions.

Bloss EB, Cembrowski MS, Karsh B, Colonell J, Fetter RD, Spruston N. Single excitatory axons form clustered synapses onto CA1 pyramidal cell dendrites. *Nat Neurosci.* 2018 Mar;21(3):353-363. doi: 10.1038/s41593-018-0084-6. Epub 2018 Feb 19. PMID: 29459763.

Kerlin A, Mohar B, Flickinger D, MacLennan BJ, Dean MB, Davis C, Spruston N, Svoboda K. Functional clustering of dendritic activity during decision-making. *Elife.* 2019 Oct 30;8:e46966. doi: 10.7554/eLife.46966. PMID: 31663507; PMCID: PMC6821494.

Losonczy A, Magee JC. Integrative properties of radial oblique dendrites in hippocampal CA1 pyramidal neurons. *Neuron.* 2006 Apr 20;50(2):291-307. doi: 10.1016/j.neuron.2006.03.016. PMID: 16630839.

Parajuli LK, Urakubo H, Takahashi-Nakazato A, Ogelman R, Iwasaki H, Koike M, Kwon H-B, Ishii S, Oh W C, Fukazawa Y, Okabe S, Geometry and the Organizational Principle of Spine Synapses along a Dendrite, eNeuro 27 October 2020, 7 (6) DOI: 10.1523/ENEURO.0248-20.2020

Shapson-Coe A, Januszewski M, Berger DR, Pope A, Wu Y, Blakely T, Schalek RL, Li PH, Wang S, Maitin-Shepard J, Karlupia N, Dorkenwald S, Sjostedt E, Leavitt L, Lee D, Troidl J, Collman F, Bailey L, Fitzmaurice A, Kar R, Field B, Wu H, Wagner-Carena J, Aley D, Lau J, Lin Z, Wei D, Pfister H, Peleg A, Jain V, Lichtman JW. A petavoxel fragment of human cerebral cortex reconstructed at nanoscale resolution. Science. 2024 May 10;384(6696):eadk4858. doi: 10.1126/science.adk4858. Epub 2024 May 10. PMID: 38723085.

Detailed comments, according to the original points:

1)

- 3-spine experiments: In contrast to the previous version, the revised manuscript finds no difference between middle and edge spine volume changes in 3-spine clusters. I should say I was perplexed that, although the modelling result has been flipped to the opposite direction, the authors continue to claim that the experiments match the model. Namely, the current prediction (no difference) is supposed to be validated by the very same data that, in the first version of the paper, have been claimed to confirm a significant difference. (The legend of the figure describing the experiment has been simply changed from “a difference can be observed” to “no difference can be observed”, still without any direct statistical comparison reported). This is hard to accept as rigorous fact-based evaluation of the data.

We thank the reviewer for this comment, and we would like to clarify potentially confusing statements with respect to the data analysis. In the original manuscript, the differences we commented on were based on model simulations where the model spines were placed on a grid whose width corresponded to the average spatial spine distribution. In this setting, the model predicted a slight difference between the edge and middle spines. Additionally, when we displayed the experimentally measured temporal plasticity trajectory, we observed that the dynamics (while not significant) displayed the distinguishing trend (edge above middle) during the 10/20 minute mark, which seemed consistent with the model simulations. It was our oversight to remark on the trend as a difference in the original manuscript. In addressing the reviewer comments on this point and as part of further analysis for the previous revision, we observed that the inherent variability in the spatial positioning of spines (experimentally measured spines exhibited a range of cluster sizes that were not sufficiently described by an average cluster size) rendered it unlikely that the small theoretically predicted difference between edge and middle, which we saw in the model, would be detectable in our experiments. Therefore, we updated the legend of Figure 2 to reflect this new knowledge. We are very grateful for the reviewers' suggestions that led to this extra insight, as we believe this has substantially strengthened the paper.

We agree that our original statement was unclear and did not adequately describe our process used to obtain it. Instead, we now write:

“Experimentally obtained normalised spine volume changes for edge (green) and middle (blue) spines are shown separately.”

We hope that this explanation clarifies our thought process, which inadvertently resulted in the seemingly contradictory statements, and hope that it alleviates the reviewer’s concern.

In addition, the text is confusing about these results. It begins with: “Our model predicts that spines on the edge of a cluster should potentiate more strongly than those in the middle.”, then “the model predicts that edge and middle spines would exhibit a similar plasticity response. This matches our experimental data. ...no significant differences between these two spine types can be expected in this experimental scenario.” Nonetheless, then it proceeds with: “So far, we have demonstrated, both experimentally and within our model, that i) spines within a cluster compete, as measured by their ability to enlarge following stimulation...”
Has the text perhaps not been completely updated?

We are grateful to the reviewer for the careful reading of our manuscript. We have now edited the text in detail to clarify our findings. Additionally, we have toned down these statements as follows:

Using this simple 3-spine stimulation scenario, we investigated if spines on the edge and middle of clusters interact as they compete for plasticity components, specifically examining whether spines on the edge of a cluster potentiate more strongly than those in the middle, as suggested by the model.

To understand how the different cluster sizes and the distances between spines may affect the plasticity dynamics, we studied the statistics of the spine-to-spine distances in our 3-spine stimulation paradigm and calculated the predicted model dynamics for the minimal (2.2 μm) and the maximal (4.2 μm) experimentally observed cluster size. Fig. 2e shows the model prediction for the edge (green) and middle (blue) spines for the mean cluster size (3.2 μm ; bold line), in which cluster sizes were obtained experimentally, along with the model predictions for the minimal and the maximal cluster sizes for edge (green) and middle (blue) spines, represented by a shaded area. The plot illustrates a way of setting limits to the variability expected in our modelled spine dynamics. Notably, there is a substantial overlap between the middle and edge spines, which suggests that our present experimental conditions for the 3-spine stimulation are not likely to promote differences between the plasticity behaviour amongst the stimulated spines. Consistently, our experimental data do not show substantial differences between edge and middle spines (Fig. 2f-h). Altogether, these results indicate that when 3 spines are stimulated, the spine-spine competition for the available resources is not sufficient to cause significant differences between the plasticity of edge and middle spines. This prompted us to

study the effects of further increasing the number of stimulated spines on spine plasticity dynamics, to which we will return later.

- Single-spine experiments: my original concern remains, i.e. that the experiments do not have sufficient temporal resolution to convincingly evaluate the model’s prediction about the fast initial kinetics or the maximum of the spine volume change in the first 10 minutes. (This also holds for the 7-spine distributed data, and affects the single-spine growth data in Fig. 6g.)

We thank the reviewer for this comment and for making us aware that the way we discussed the limitations of our temporal resolution was not clear enough. We have now toned down or removed statements referencing the fast dynamics in the initial plasticity phase of 0-2 mins (see above).

We also performed additional analysis, which showed that our model could capture the trajectory of the 7 spine dynamics after the stimulation phase and before the 2 min point. This analysis implies that our model (whose parameters we fitted using the 3-spine stimulation paradigm) may be equipped to describe the early post-stimulation spine dynamics in other multi-spine stimulation paradigms, such as the 7-spine experiment. Future work can use our current model as a platform to address the spine dynamics during the stimulation phase in more detail and study how the initial dynamics morph into the response dynamics across the longer time scales (2-40 mins), which were the focus of our study.

With regards to Fig. 6g, we have clarified the y-axis caption by replacing the potentially misleading phrase “total growth”, which refers to the total growth not over time but instead over the stimulated spines at the +2 min time point, to “Summed norm. Vol. (stim. spines, +2 min)”. We hope the new caption conveys the intended message more precisely. This is shown below (Figure 6i in the new version of the manuscript).

The authors rather tried to gain more insight into the decay by new exponential decay fits (shown in Fig.6), but this approach appears limited. From what is presented it seems that no statistical evidence was found for a different decay of single spines from that in the other groups. The individual spine decay parameters are not significantly different

between any of the groups according to the p values in Fig.6e (contrary to the description in the text). It is mentioned that spines with erroneous fits were omitted, but how many spines remained in the different groups is not specified.

We thank the reviewer for mentioning this point and agree that our approach, as presented in the previous version of the manuscript, may not have been optimal. Following the previous advice of the reviewers, we originally had attempted to find the statistical distribution of the exponential decay parameters by fitting each of the individual spines and collecting those parameters as an estimate for each of the experimental paradigms. These collected fits were then compared against each other. As noted by the reviewer, this approach was limited as no statistical difference was found, and certain spines needed to be omitted.

We thus altered the approach for generating the statistical distributions of the parameters for the exponential models. As described in the statistical methods section of the updated manuscript, we now opted for a bootstrapping approach, where for each experimental paradigm, a subset of experiments are sampled (with replacement), and the exponential model was fit to this subset. This procedure was repeated 1000 times (as this was found to lead to stable statistics for the parameters). The part in the statistical methods section reads:

“To obtain the fits for the parameters of the exponential model, a bootstrapping approach was utilised. Half of the experiments were randomly selected for each spine stimulation paradigm (with replacement) and the exponential model was fit. This procedure was repeated 1000 times, as this value was found to lead to stable statistics for the a, b and c parameters. To test for significance, first a Kruskal-Wallis test was performed followed by Dunn’s multiple comparison with the false discovery rate (FDR) correction factor.”

We have added the results of this bootstrapping as Figures 6e, f and g (all parameters of the exponential decay model and not just the decay itself are now included) in the revised manuscript and show them below:

Caption: Statistical distributions for the exponential model parameters (“a”, “b”, and “c” in panels e, f, and g, respectively) were obtained using the bootstrapping approach described in the statistical definitions section. The star in e indicates that, according to the Kruskal-Wallis test followed by a pairwise Dunn’s test, each of the experiments differs significantly ($p < 0.05$). In f and g, identical symbols denote experiments that do not differ significantly from each other; that is,

when two experiments share the same symbol, they are not statistically different. Differences in the parameters across the different stimulation paradigms indicate a non-linear dependence of the temporal plasticity scales. Boxes represent the interquartile range of the bootstrapped fits; the whiskers represent the 95% confidence interval, the black triangles are the mean of the data and the orange lines are the median.

We found that these fits revealed differences in the dynamics of the different experiments that align well with the conclusions of the more complex model. We have added this discussion to the updated manuscript as follows

To statistically quantify the differences in the exponential decay model parameters, we performed a bootstrapping procedure, where the exponential was fit on a randomly selected subset within a stimulation paradigm and this procedure was repeated (see the methods section for more details). This procedure led to a distribution of the model parameters that we could compare in Fig 6e-g. Figure 6e shows the different fits of a (or the scaling factor of the exponential model) with a larger a implying a greater initial amplitude of the dynamics. Significant differences between the fits of this parameter for the stimulation paradigms reflect the ordering of the more complicated model we presented, albeit with the single-spine example having a larger a than the 3-spine example. Next, in Figure 6f, which depicts the degradation rate b , we see that the single spine decays significantly faster than the other 3 and 7 spine experiments, which agrees with our model dynamics in Figure 6b. We note that the 15-spine example also exhibits high degradation rates; however, given the aforementioned incompatibility of the 15-spine dynamics with the exponential model, we attribute their large values to a relative difficulty fitting the dynamics to the exponential decay. Finally, we consider the saturation value c , which is the value that the exponential model tends towards as time advances and is shown in Figure 6g. The 15-spine example has the highest value (significantly above the other experiments), while the single-spine experiment saturates at the lowest value. These observations are consistent with the experimental data, and the results of the more complicated model fits. In line with the findings of the more complex model, the exponential decay model suggests that the 3 and 7 spine experiments fall within similar regimes, having similar decay rates and final saturation values, while the single and 15 spine experiments could represent fundamentally different dynamics.

- 7-spine experiments: I still find it problematic that the claimed difference between middle and edge spines is based on an odd single data point at 20 minutes. By 30 minutes we don't see difference in the data, although the model predicts that the two spine groups should be distinct. In addition, it is not clear whether a difference at 20 minutes remains significant after correction for multiple comparisons (see below).

We agree with the reviewer that the 7 spine experiments, as presented in the first revision, were not as compelling due to variability in the data. Given that the results of 7-spine experiments constitute a key component of our findings, we were determined to include more data in the analysis with the aim of reducing the variability and increasing the power of our statistical tests. To this end, we added 13 new experiments of 7-spine stimulations, which we had initially performed for a related unpublished project, and now included them in the present manuscript.

The result of these extra datasets, which doubled the number of experiments from our original dataset, are shown in the new figure 4 (shown below):

The model prediction is in agreement with the mean dynamics of the seven stimulated spines (Fig 4d); a robust potentiation is observed at +2 min, and following a gradual decline, potentiation still remains at +40 min. Importantly, the additional experiments have reduced the data variability, and the new analysis reveals a significant difference between the middle and the edge spines at +30 min and +40 min (albeit losing the significance at the +20 min time point). This is similar to what we see for the 15 spine condition. The significant differences at 30 min and 40 min remain when correcting for multiple comparisons, and this result is in agreement with the model prediction. Altogether, the addition of new experiments has considerably strengthened our results, and we are grateful to the reviewer for raising this point.

There is a relatively wide range of different cluster lengths reported in this experiment, from 6 to 13 micrometers. Is there a correlation between predicted and experimentally measured differences for middle and edge spines?

The reviewer has made an important point regarding our 7-spine experiments, namely that there is some variation in the distances between spines and hence the cluster size, and asks if our model prediction shows differences between edge and middle spines that correlate to

experimental variations, an approach we previously hadn't considered. We are happy to implement the comparison in this revision, which is helped by the strengthened dataset with the inclusion of extra 7-spine stimulation experiments.

Below, we show the differences in edge and middle coming from the model and experiment against each other, alongside a linear regression (with a 95% confidence interval) and the calculated Pearson correlation coefficient. This analysis indicates that there is a measure of positive correlation (+0.39), at a p-level of 0.056.

Overall, as the comparative summary plots in Fig 6a-b highlight, the match between the measured vs modelled differences in spine dynamics is generally rather low, except for the 15-spine scenario.

We thank the reviewer for this comment. For the post-stimulation period, the correlation (r -squared) between the experimental data and the model is 0.743 for the 1-spine condition (Figure 3g), 0.947 for the 3-spine condition (Figure 2d), 0.967 for the 7-spine condition (Figure 4d), and 0.823 for the 15-spine condition (Figure 5d).

We hope that with the new strengthened 7-spine dataset, the close similarity between the experimental data and the model prediction for the differences in the behaviour of edge vs. middle spines is convincing not only for the 15-spine case but also for the 7-spine case. In

addition, as mentioned above, the model prediction for the first minute is simply an extrapolation, and it is now depicted in all relevant figure plots as a thin dashed line.

- Statistics: The cosmetic modifications in the revision did not address the problem of multiple uncorrected comparisons (also raised by Reviewer 1). Multiple paired comparisons accumulate statistical errors, which need to be adjusted by appropriate corrections (e.g. Bonferroni or other). The authors need to clarify how they handle this, if they do. This concerns not only the comparisons of different cluster arrangements, but also the multiple comparisons between middle and edge spines at different time points, where weak significant differences are reported by paired tests (e.g. Fig. 4f, Fig. 5f).

We thank the reviewer for raising this issue. In order to implement the necessary corrections to ensure robust statistical tests, we performed the Kruskal-Wallis test, followed by Dunn's multiple comparisons with the false discovery rate (FDR) correction factor following the Benjamini-Hochberg procedure. We find that spine volume increase in control and FK506 remain significant compared to sham, whereas the increase in the presence of AIP, a CaMKII inhibitor, is largely attenuated and not significant compared to sham, highlighting the involvement of CaMKII in spine potentiation. We have updated all figures and p-values in the new manuscript, as well as added the following clarification to the statistical methods section:

"To adjust for multiple comparisons, we employed the false discovery rate (FDR) method proposed by Benjamini and Hochberg (1995)."

- Some of the rows in Table 1 contain lower number of stimulated spines per experiment than the expected set (e.g 7-spine, 7-spine distributed, 15-spine, 15-spine sham). What is the explanation for this discrepancy?

We are grateful to the reviewer for the careful reading of our manuscript, particularly noting that the number of spines reported in Table 1 does not align with what would be expected (i.e. Total number of spines = Number of stimulated spines per experiment * Number of experiments). This discrepancy arises due to a spine occasionally being obscured in the z-axis by another structure (typically a dendrite) and, therefore, not being analysed by our pipeline. Nonetheless, we are able to see where these spines have fallen within the ordering of the cluster, and thus, we can retain the consistency in the classification of the spines (edge vs middle).

An example of such a scenario can be seen below, where we see a spine (S1 in figure) covered by another dendrite in the field of view (scale bar = 5 μ m).

We apologise for the oversight in having not mentioned this in the original manuscript. We have added a sentence to the updated manuscript that is as follows:

Spines that were obscured by a dendrite or other spines were omitted from the analysis.

2) The new Ca²⁺ measurements show large, homogeneous, nonlinear increases in the spine and dendrite Ca²⁺ signals by clustered multi-spine stimulations, indicating a qualitatively different response and activation of additional Ca²⁺ sources compared to single-spine stimulation, and drastically changing the dynamics and dendritic distribution of Ca²⁺ from that assumed in the model (Fig. 1g). These results therefore warn against simplifying the mechanism that determines plasticity to competition for two components originating from individual spine activations.

The reviewer suggests that multi-spine stimulations likely engage Ca²⁺-dependent responses that are distinct and more complex from single-spine stimulation due to differences in the spatial and temporal dynamics of Ca²⁺ measurements. The reviewer then raises questions about the validity of the model that is based on competition for two Ca²⁺-dependent components. While we acknowledge the concern with respect to simplifying the model mechanisms, we would like to emphasise the following points concerning our study.

First and foremost, our model parameter fits have been performed using only the 3-spine stimulation paradigm, not the single-spine stimulation, and subsequent model predictions are based on the initially fixed parameters (cf. Methods under “Model Fitting Algorithm”). In basing the model parameters on the simplest multispine stimulation scenario rather than the single spine stimulation case, we aimed to incorporate the cooperative nature of Ca²⁺-triggered responses and to help generalise across paradigms using different stimulation numbers (1, 7 and 15).

Second, it is important to mention that in this study, Ca²⁺-dynamics have been monitored using GCaMP6s, which has relatively slow kinetics (Chen et al., Nature, 2013). Overall, the measurements of Ca²⁺ concentration changes are highly dependent on the sensor kinetics, and any nonlinearities would be further skewed by a slow Ca²⁺ sensor. Consequently, in the present experimental setup, it is challenging to assess the relative Ca²⁺ activation ratios across different stimulation paradigms, and moreover, we cannot use the GCaMP6s signals as a proxy for the initial $C(0)$ values in our model.

Third, one effect our measured GCaMP6s signals show reliably is that in single spines, intracellular Ca^{2+} levels decay to baseline levels within ~ 20 sec after stimulation, with the true decline expected to be even faster. As mentioned above, our model is designed to describe the effects of Ca^{2+} *after* the stimulation. The degradation coefficient of C in our model, after fitting, was 1.05 1/s. Therefore, in consideration of the fast-acting Ca^{2+} kinetics triggered by plasticity induction, our C(t) variable likely represents the collective action of Ca^{2+} -activated molecules rather than the level of intracellular Ca^{2+} *per se*. In other words, C accounts for the influence of Ca^{2+} -binding molecules rather than Ca^{2+} directly.

Altogether, we acknowledge that the model involves simplification of complex biological processes. Nevertheless, sharing or competing for resources is a general biological concept prompted by finite resources. Given the multitude of biochemical signalling cascades and downstream processes that can be engaged in spines upon their activation, a reductionist approach in considering a simple model that relies on competition for resources represented by just two dynamic components can be beneficial. In fact, that the model, whose parameter fits have been based on the 3-spine stimulation, can predict to some degree the behaviour of spine plasticity dynamics in 7-spine and 15-spine stimulation paradigms where competitive processes are plausible, supports the utility of our modelling approach.

3) The controls for glutamate uncaging are a useful addition. It would be nice to see whether the uncaging stimulus generates similar spine responses at the middle vs the edge of the the field of view.

To address the point of uniformity of uncaging stimulus, we performed glutamate uncaging coupled with calcium imaging to assess the variability across the imaging field of view. We stimulated single spines and plotted the peak calcium transient against the distance from the edge of the imaging field of view for each stimulated spine. This is shown below. The flat distribution of the data demonstrates that there is no apparent effect of the position of uncaging – i.e. middle vs. edge of the field of view - on the neuronal response.

Caption: Uncaging-triggered spine calcium levels are stable across the imaging field of view. Neurons expressing GCaMP6s, along with a red volume marker (DsRed) were targeted for single-spine uncaging. Single 1 millisecond pulses of uncaging laser light were delivered $0.5 \mu\text{m}$ from

individual spine heads, and the resultant calcium transient recorded. 50 spines, from 10 dendrites, from 2 neurons.

However, my concern about variable initial spine sizes affecting the plasticity outcome has not been completely resolved. The distributions of initial spine sizes in the multi-spine experiments are very wide and are non-normal (the mean values are not very informative). Importantly, the larger initial spine size in the 15-spine experiments may contribute to why the normalized peak spine increase was numerically lower than in the other experiments, questioning whether this difference is biological.

We thank the reviewer for this comment, which we agree we did not address sufficiently in the previous reply. We concur with the point that because large spines can only grow so large if they represent a considerable proportion of stimulated spines in the 15-spine experiment, their presence could limit the size of the subsequent *normalised* evolution. We noted that 88.3% of the spines in the 15-spine experiment fell within the range of the 3-spine experiment (see below, dotted red lines). Therefore, we believe that the primary dynamics we observe are driven by spines that fall within this range rather than being skewed by the behaviour of a small proportion of large spines. To ensure that this was actually the case, we performed the following analysis.

First, we trimmed the outliers from the 15 spine example so that the average initial size of each spine in the 15 spine experiment fell within the limits of the 1 spine example (green dotted lines in the left-hand side figure below). We then plotted the dynamics of this and compared it to the full 15 spine dynamics as well as the single spine dynamics (below, right-hand side). We note that despite being in the same initial size range, the dynamics of the 15 spine - 1 range line (lightest blue) shares the dynamics of the full 15 spine example and does not show a significant difference, whereas when compared to the single spine dynamics, it significantly differs at +10 min and +30 min timepoints ($p = 0.022, 0.037$, respectively. t-test with multiple pairwise correction).

Next, we trimmed the outliers from the 15 spine example so that the average initial size of each spine in the 15 spine experiment fell within the limits of the 3 spine example (red line in the figure above). The spine size distribution of this trimmed 15 spine (i.e. “15 Spine – 3 range”)

and the original 3 spine experiment was not significantly different (Kolmogorov-Smirnov two-sample test led to a p-value of 0.718). We then compared the dynamics of the 15 spine and 15 spine - 3 range experiments (dark and light blue, respectively, in the plot below) and saw no significant differences (albeit a slightly bigger growth in the trimmed set, as expected) between the two dynamics.

Altogether, the analysis points to the fact that the initial spine size distribution is less impactful than the stimulation paradigm. We hope this alleviates the reviewer’s concern regarding the variability of initial spine sizes and their effects on the outcome of multispine stimulation.

5) The z-stack of the original 3-spine experiment indeed hints that spines projecting in the z-plane could be responsible for the enormous swelling. However, this illustrates an additional issue. The enlarged spines were not targeted for uncaging, but they were very close to the targeted spines, likely within the 3D volume of the glutamate stimulus. This contradicts the authors’ argument in the rebuttal that “the single-spine stimulation elicited structural plasticity only in the targeted spine, while leaving nearby neighbouring spines unchanged”. It rather suggests that the stimulus is often not restricted to single synapses, meaning that the real number of stimulated spines in the experiments is not known.

Thank you for this comment. Let us clarify that our comment on “single-spine stimulation” was intended to refer to the single-spine condition and not the 3-, 7-, or 15-spine conditions. As demonstrated by our calcium imaging data, stimulating *clusters* of spines elicits dendritic calcium transients, which then may back-propagate into nearby neighbouring spines and transiently potentiate them. Moreover, quantification of both spine volume (included in the previous rebuttal comments and reproduced below) and calcium transients show little to no changes in nearby spines during single-spine stimulation.

Caption: Single-spine stimulation is specific to the targeted spine.

- a) *Changes in neighbouring spine size following sLTP induction of a single target spine. Change in heterosynaptic, non-stimulated spine size is plotted against distance from stimulated spine sharing the same dendrite. (Data from 10 neurons, 299 neighbouring spines)*
- b) *Calcium responses in targeted spines and neighbouring spines. Single 1 msec pulses of glutamate were delivered 0.5 μm from a target spine head (yellow arrow), and GCaMP6s fluorescence was measured in it and nearby neighbouring spines in the same z imaging plane. Calcium transients are largely restricted to the target spine. (Data from 3 neurons, 10 target spines, 29 neighbouring spines)*

We agree with the reviewer that glutamate spillover may be a factor during multispine stimulation and could contribute to the emerging plasticity. There is evidence from acute slice experiments that highly synchronous presynaptic activity leads to glutamate spillover across nearby inputs (Biró and Nusser, 2005) and, as such, may act to coordinate plasticity across inputs close in space. A recent model of glutamate spillover in extrasynaptic space that took into account the presence of perisynaptic astrocytes and glutamate transporter binding, estimated that millimolar concentrations of glutamate released would rapidly decay by 1000 to 3000 fold within 3 ms of release, yet simulations predicted a long tail of free glutamate of >10 μM that remained at ~300 nm from the release site at 3 ms (Savtchenko and Rusakov, 2022).

Therefore, MNI-glutamate uncaged at 500 nm from the spine head is likely to contribute low levels of glutamate to a nearby spine, which by itself may not be sufficient to trigger a spine response in the same way as the presynaptically released glutamate or glutamate uncaging at the targeted spine. Nonetheless, in the case of multi-spine plasticity that stimulates nearby spines (where the typical interspine distance between all spines on the dendrite is ~0.5-1 μm), within 3 ms of uncaging of the previous spine, the small spillover could add to the effects of uncaged glutamate. As mentioned above, our model parameters have been fit using not the single spine stimulation case but the 3-spine stimulation experiments. Therefore, the model

already takes into account the residual effect of glutamate spillover from stimulating the previous spine, should such a spillover effect occur. A comment on the potential effects of spillover is now included in the discussion in the revised manuscript: “Notably, our multi-spine stimulation paradigm could potentially promote glutamate spillover amongst spines of the targeted cluster and contribute to the spine plasticity outcome. Although the model parameters were fit using the 3-spine stimulation experiments where compound effects of spillover, if present, would have been included, future studies into the nature of C and P should take into consideration the effects of glutamate spillover.”

Biró AA, Nusser Z. Synapse independence breaks down during highly synchronous network activity in the rat hippocampus. *Eur J Neurosci*. 2005 Sep;22(5):1257-62. doi: 10.1111/j.1460-9568.2005.04304.x. PMID: 16176369; PMCID: PMC1560105.

Savtchenko LP, Rusakov DA. Increased Extrasynaptic Glutamate Escape in Stochastically Shaped Probabilistic Synaptic Environment. *Biomedicines*. 2022 Sep 26;10(10):2406.

Given the above issues, the manuscript contains many exaggerated statements about the results (examples: “we confirm one of the model’s prediction, that single spines rapidly grow (as there is no competition for C) but then equally rapidly shrink back to a lower value”, “...our model... could accurately describe the experimentally generated plasticity and its time course across different numbers of stimulated spines.”; “the model captures the final saturation rate of each experiment, underscoring the idea that the number of spines stimulated fundamentally affects the increase in the size of the spines reached over the long-term”, etc) The authors need to tone down their interpretations and make only well supported conclusions.

We are grateful to the reviewer for making this point and agree that our conclusions were stated too strongly in comparison to the results. We have now toned down the text throughout the manuscript to reflect the results appropriately and limit our conclusions to those that are directly substantiated by the results. For example, the passages highlighted by the reviewer now read:

- *we confirm one of the model’s prediction, that single spines rapidly grow (as there is no competition for C) but then equally rapidly shrink back to a lower value:*

→

Nonetheless, the dynamics align well with one of the model's prediction, that single spines rapidly grow (as there is no competition for C)

- *our model... could accurately describe the experimentally generated plasticity and its time course across different numbers of stimulated spines.*

→

our model ... could describe in reasonable agreement the experimentally generated plasticity and its time course across different numbers of stimulated spines.

- *the model captures the final saturation rate of each experiment, underscoring the idea that the number of spines stimulated fundamentally affects the increase in the size of the spines reached over the long-term → this has been completely removed.*

The Abstract includes some inaccuracies.

“Potentiation of clusters of dendritic spines lead to changes that are dependent on NMDAR, CaMKII and calcineurin.” This has only been shown for single spines.

We thank the reviewer for this comment and apologise for the confusion. This phrase was not meant to refer to the findings in this work but instead indicate the results found in the literature, such as Ouyang et al. 1997, Rose et al. 2009, and Tong et al. 2021. We have changed the sentence to clarify this point as follows:

We used glutamate uncaging to induce sLTP in clusters of synapses sharing the same dendritic branch, whilst systematically varying the number of stimulated inputs, which has been shown previously to produce changes that are dependent on NMDAR, CaMKII and calcineurin.

‘our results provide a quantitative description of the multi-spine stimulation footprint over minutes and hours post-stimulation’. Hours have not been investigated.

We agree with the reviewer that the previous sentence in the abstract did not accurately reflect the work performed in our study. We have corrected the relevant section of the abstract so that it now reads:

Comparison of our experimental observations with model predictions suggests that (i) competition among spines for molecular resources is a key driver of multi-spine plasticity and (ii) the spatial distance between simultaneously stimulated spines impacts the resulting spine dynamics. Moreover, our model can account for the seemingly conflicting plasticity reports obtained across varying numbers and positions of stimulated spines. Altogether, our results provide a quantitative description of the multi-spine stimulation footprint over minutes to tens of minutes post-stimulation across several of microns of dendritic space.

REVIEWERS' COMMENTS

Reviewer #2 (Remarks to the Author):

The authors have made appropriate modifications to the manuscript according to this reviewer's suggestions. The manuscript is an important experimental and theoretical contribution to the mechanism underlying the expression of multi-synapse plasticity regulated by multiple local synaptic inputs.

Dear reviewers,

Thank you very much for the time you took to give us feedback. We have revised our manuscript to incorporate all comments and to adhere to editorial policies.

Reviewer #2 (Remarks to the Author):

The authors have made appropriate modifications to the manuscript according to this reviewer's suggestions. The manuscript is an important experimental and theoretical contribution to the mechanism underlying the expression of multi-synapse plasticity regulated by multiple local synaptic inputs.

Thank you very much for the appreciation of our manuscript, which we have now finalized for publication.